

# Microstructure and composition of marine aggregates as co-determinants for vertical particulate organic carbon transfer in the global ocean

Joeran Maerz[1], Katharina D. Six[1], Irene Stemmler[1], Soeren Ahmerkamp[2], and Tatiana Ilyina[1]

[1]Max Planck Institute for Meteorology (MPI-M), Hamburg, Germany
[2]Max Planck Institute for Marine Microbiology (MPI-MM), Bremen, Germany

**Correspondence:** Joeran Maerz (joeran.maerz@mpimet.mpg.de)

**Abstract.**

Marine aggregates are the vector for biogenically bound carbon and nutrients from the euphotic zone to the interior of the oceans. To improve the representation of this biological carbon pump in the global biogeochemical HAMburg Ocean Carbon Cycle (HAMOCC) model, we implemented a novel *Microstructure, Multiscale, Mechanistic, Marine Aggregates in the Global Ocean* ($M^4AGO$) sinking scheme. $M^4AGO$ explicitly represents the size, microstructure, heterogeneous composition, density, and porosity of aggregates, and ties ballasting mineral and particulate organic carbon (POC) fluxes together. Additionally, we incorporated temperature-dependent remineralization of POC. We compare $M^4AGO$ with the standard HAMOCC version, where POC fluxes follow a Martin curve approach with linearly increasing sinking velocity with depth, and temperature-independent remineralization. Minerals descend separately with a constant speed. In contrast to the standard HAMOCC, $M^4AGO$ reproduces the latitudinal pattern of POC transfer efficiency which has been recently constrained by Weber et al. (2016). High latitudes show transfer efficiencies of $\approx 0.25 \pm 0.04$ and the subtropical gyres show lower values of about $0.10 \pm 0.03$. In addition to temperature as a driving factor, diatom frustule size co-determines POC fluxes in silicifiers-dominated ocean regions while calcium carbonate enhances the aggregate excess density, and thus sinking velocity in subtropical gyres. In ocean standalone runs and rising carbon dioxide ($CO_2$) without $CO_2$ climate feedback, $M^4AGO$ alters the regional ocean-atmosphere $CO_2$ fluxes compared to the standard model. $M^4AGO$ exhibits higher $CO_2$ uptake in the Southern Ocean compared to the standard run while in subtropical gyres, less $CO_2$ is taken up. Overall, the global oceanic $CO_2$ uptake remains the same. With the explicit representation of measurable aggregate properties, $M^4AGO$ can serve as a testbed for evaluating the impact of aggregate-associated processes on global biogeochemical cycles, and, in particular, on the biological carbon pump.





## 1 Introduction

Marine aggregates transfer biologically bound carbon and nutrients from the sunlit surface waters, the euphotic zone, to the interior of the oceans. While uncertainty with respect to primary production estimates exists, about $4.0\,\mathrm{Gt\,C\,yr^{-1}}$ to $11.2\,\mathrm{Gt\,C\,yr^{-1}}$ biologically bound carbon are annually exported out of the euphotic zone of the global ocean (Laws et al., 2000; Najjar et al., 2007; Henson et al., 2012). The net-withdrawal of carbon dioxide ($CO_2$) from the ocean surface through export of carbon bound in particulate organic matter (POM) and biogenic minerals and subsequent release through microbial remineralization and dissolution during aggregates descent determines the strength of the so-called biological carbon pump. The biological carbon pump critically depends on phytoplankton growth, the replenishment of the euphotic zone with nutrients through mixing and upwelling processes, and the efficiency of biologically bound carbon transfer from surface waters to the interior of the oceans. The region and depth of carbon sequestration eventually determines the residence time of the biologically bound carbon upon recurrence at the oceans surface. Representing transport and fate of marine aggregates in Earth System Models (ESMs) is therefore key to quantify the future evolution of biogeochemical cycles, and particularly the biological carbon pump and its feedback on the Earth system under climate change (Ilyina and Friedlingstein, 2016). In the present study, we thus aim at advancing the representation of marine aggregates in an ESM framework.

Marine aggregates are porous entities which are heterogeneously composed of POM, biogenic and inorganic minerals. The sinking velocity of marine aggregates, their microbial remineralization and zooplankton grazing governs the attenuation of vertical particulate organic carbon (POC) fluxes. The sinking velocity of aggregates is primarily determined by their size. In addition, the internal microstructure, defined by the porosity and heterogeneous composition, entail high variability of excess density and thus sinking speed of aggregates. Biogenic calcium carbonate ($CaCO_3$) and opal structures, primarily formed by coccolithophores and diatoms, act as ballasting minerals in organic aggregates (De La Rocha and Passow, 2007; Armstrong et al., 2002). On the contrary, the available amount of POC, acting as glue, is suggested to limit the uptake capability for ballasting minerals before aggregates disintegrate (Passow, 2004; Passow and De La Rocha, 2006; De La Rocha et al., 2008). Ballasting increases the POC transfer efficiency (Klaas and Archer, 2002; Balch et al., 2010; Cram et al., 2018), defined as the fraction of POC exported out of the euphotic zone that reaches a particular depth, e. g. 1000 m (Francois et al., 2002). As $CaCO_3$ is significantly denser than opal, $CaCO_3$ is suggested to be a more effective ballasting material for marine aggregates (Balch et al., 2010) implying higher POC transfer efficiency in $CaCO_3$ production-dominated regions. Phytoplankton communities possess spatio-temporally varying patterns and prime the sinking flux ratios of detritus to ballasting minerals, i. e. the rain ratios. High $CaCO_3$ to opal ratios are found in oligotrophic regions of the mid-latitude subtropical gyres, while opal is the prevalent ballasting mineral in high latitudes and upwelling-influenced equatorial regions (Balch et al., 2010). However, simple ballasting relationships on aggregates are questioned and the prevailing plankton network is suggested as an additional driver for POC fluxes (Wilson et al., 2012; Henson et al., 2012; Guidi et al., 2016). For example, cell size and morphology present in the phytoplankton community are suggested as primer determining factor for sinking velocity of marine aggregates (Laurenceau-Cornec et al., 2015; Bach et al., 2016). In turn, the attenuation of POC fluxes is hypothesized to be modulated by microbial remineralization and by zooplankton grazing in oligotrophic and eutrophic regions, respectively (Guidi et al., 2009). Since





temperature controls enzymatic reaction kinetics, and thus microbial remineralization of POC, slower attenuation and thus higher transfer efficiency is suggested in cold high latitudes compared to warm oligotrophic regions (Marsay et al., 2015). For a long time, the aforementioned variable factors and processes, the limited understanding of aggregation and fragmentation processes that shape the aggregate size spectrum, and the sparse amount of data have retarded the emergence of a detailed

picture of global pattern of POC fluxes attenuation and thus transfer efficiency.

However, quantification of the regionally varying POC transfer efficiency and their variability is key to understand global biogeochemical cycles, in particular the carbon cycle (Falkowski et al., 1998). Recently, global POC fluxes have been constrained to possess high transfer efficiency in high latitudes and upwelling regions, and lower efficiency in the subtropical gyres (Weber et al., 2016). The underlying controls for the transfer efficiency pattern seem to exhibit a distinct latitudinal

variability (Cram et al., 2018). The simplified model study of Cram et al. (2018) suggests aggregate size, ballasting of particles by $CaCO_3$ and opal, temperature effects on microbial aerobic and anaerobic remineralization, water density, and molecular viscosity as major controls of the transfer efficiency pattern.

Processes of marine snow formation, ballasting and sinking are currently underrepresented in ESMs despite the relevance of aggregates for the transfer and sequestration of POC to the deep ocean. Only a few global models explicitly incorporate

aggregation of phytoplankton mechanistically (e. g. Gehlen et al., 2006; Schwinger et al., 2016) while ignoring ballasting effects or vice versa (Gehlen et al., 2006; Heinemann et al., 2019). POC sinking velocities in ESMs are typically formulated to reproduce the Martin curve (Martin et al., 1987) or heuristically describe ballasting of POC with opal and $CaCO_3$ (e. g. Gehlen et al., 2006; Heinemann et al., 2019), which limits the process-based adaptation of sinking velocities under changing environmental conditions associated with climate change.

As a first step, we develop the *Microstructure, Multiscale, Mechanistic, Marine Aggregates in the Global Ocean* (M⁴AGO) sinking scheme that explicitly represents composition, microstructure and related properties such as porosity and density of aggregates. We aim at consistently defining marine aggregates with their *in situ* measurable properties in an ESM framework. We implement M⁴AGO in the global HAMburg Ocean Carbon Cycle (HAMOCC) model which is part of the Max Planck Institute - Earth System Model (MPI-ESM), explicate the emerging pattern of aggregate properties, and examine their effect

on sinking velocity and the global pattern of POC transfer efficiency. We particularly aim at: i) representing the POC transfer efficiency pattern of Weber et al. (2016), ii) providing further understanding into the underlying driving factors for this pattern and iii) giving insights on the impact of M⁴AGO on the global $CO_2$ flux pattern. We focus on the transfer efficiency pattern identified by Weber et al. (2016) as it was derived by diagnosing phosphate fluxes from World Ocean Atlas 2009 phosphate concentration via inverse modeling. This approach benefits from order of magnitude more observations than direct flux obser-

vations (Usbeck et al., 2003; Weber et al., 2016) and can thus be regarded as, to date, more reliable than previous estimates (e. g. Henson et al., 2012; Marsay et al., 2015).

With M⁴AGO, we represent marine aggregates at global scale to provide a testbed for future investigations of aggregate-associated processes in ESMs, e. g. particle size-, microstructure- and composition-dependent remineralization rates.





## 2 Model description

The HAMburg Ocean Carbon Cycle (HAMOCC) model is a global biogeochemical model which features biology and resolves the carbon chemistry (Six and Maier-Reimer, 1996; Ilyina et al., 2013; Paulsen et al., 2017; Mauritsen et al., 2019). HAMOCC assumes a fixed stoichiometry for dead and living organic matter, and represents the nutrients phosphate, nitrate, silicate, and iron. HAMOCCs phytoplankton, namely bulk phytoplankton and diazotrophs, can thus experience nutrient co-limitation. Diazotrophs assimilate gaseous di-nitrogen under nitrate limitation and compete for phosphorus with bulk phytoplankton. Diazotrophs grow slower than bulk phytoplankton and have their optimal growth temperature at about $28\,°C$ (Paulsen et al., 2017, 2018). Zooplankton feeds on bulk phytoplankton and releases POM which enters the common detritus pool. During detritus formation through bulk phytoplankton or zooplankton, opal or $CaCO_3$ is produced depending on silicate availability. This treatment adequately depicts the spatial distribution of silicifying and calcifying plankton communities (Heinze et al., 1999). HAMOCC represents sediment processes (Heinze et al., 1999) and is coupled to the global three dimensional Max Planck Institute Ocean & Sea Ice Model (MPI-OM; Marsland et al., 2003; Jungclaus et al., 2013). HAMOCC is described and evaluated in previous studies, for details see e. g. Six and Maier-Reimer (1996); Ilyina et al. (2013); Paulsen et al. (2017); Mauritsen et al. (2019). In the following, we therefore focus on processes in the standard version, i. e. sinking and remineralization, which we modify with the $M^4AGO$ sinking scheme.

### 2.1 HAMOCCs standard representation of sinking fluxes & remineralization

The standard version of HAMOCC (Mauritsen et al., 2019) represents sinking fluxes of POC, $F_{POC}$, at depth $z > z_0$ according to the concept of the Martin curve (Martin et al., 1987; Kriest and Oschlies, 2008)

$$F_{POC}(z) = F_0 \left( \frac{z}{z_0} \right)^{-\beta} \tag{1}$$

where $F_0$ is the POC flux out of the euphotic zone at export depth $z_0$. For simplicity, the export depth is globally defined as being $z_0 = 100\,\mathrm{m}$ in HAMOCC. Above $z_0$, a constant sinking speed of $3.5\,\mathrm{m\,d^{-1}}$ is assumed. Below $z_0$, we assume a linearly increasing mean sinking velocity with depth. The ratio between the remineralization rate of POC, $R_{POC,remin}$ and the vertical gradient of the sinking velocity $\partial_z \bar{w}_s$ determines the POC flux slope $\beta = R_{POC,remin}/\partial_z \bar{w}_s$ (Kriest and Oschlies, 2008). In the standard version of HAMOCC, remineralization of POC is temperature-independent and comprehends oxygen concentration-dependent aerobic remineralization as well as sulphate reduction and denitrification under sub- and anoxic conditions.

The sinking tracers opal and $CaCO_3$ are treated separately from POC and sink with their own, constant sinking velocity. Aeolian dust is, apart from the release of bioavailable iron in surface waters, inert and sinks slowly through the water column. Opal dissolution rate in the standard model is linearly temperature-dependent. HAMOCC accounts for dissolution in carbonate ion under-saturated conditions below the dynamically emerging lysocline. In the following, we refer to this version of HAMOCC as 'standard'.





## 2.2 The novel M$^4$AGO sinking scheme in HAMOCC

Natural waters exhibit a size spectrum of aggregates whose diameter, $d$, composition and microstructure determine their terminal sinking velocity. In the M$^4$AGO approach, we explicitly represent microstructure and heterogeneous composition of aggregates. For the aggregate size spectrum, we limit the representation to a variable power law number distribution, $n(d)$,

with slope $b$ and power law factor $a$ (following e. g. Kriest and Evans, 1999; Gehlen et al., 2006; Schwinger et al., 2016)

$$n(d) = a\,d^{-b} \tag{2}$$

This way, we avoid the computational costs of size-class based model approaches (Jackson, 1990; Stemmann et al., 2004; Sherwood et al., 2018).

The local concentration-weighted mean sinking velocity, $\langle w_s \rangle$, in M$^4$AGO is eventually determined by a truncated number

distribution, Eq. (2), through the minimum and maximum aggregates sizes, $d_{\min}$ and $d_{\max}$, respectively, the aggregate mass, $m(d)$, and the sinking velocity of single aggregates, $w_s(d)$

$$\langle w_s \rangle = \frac{\int\limits_{d_{\min}}^{d_{\max}} n(d)\,m(d)\,w_s(d)\,\mathrm{d}d}{\int\limits_{d_{\min}}^{d_{\max}} n(d)\,m(d)\,\mathrm{d}d} \tag{3}$$

We only implicitly account for aggregation and fragmentation and explicitly represent the temporally and spatially variable heterogeneity and microstructure of aggregates and their effect on the mean sinking velocity. We refrain from representing

the potential heterogeneity of aggregate composition within the local size spectrum (see e. g. Jackson, 1998). Consequently, and in contrast to the standard configuration, the settling tracers in HAMOCC, opal, CaCO$_3$, detritus, and dust, are sinking in M$^4$AGO at the same mean sinking velocity of aggregates, Eq. (3). In contrast to Gehlen et al. (2006); Schwinger et al. (2016); Heinemann et al. (2019), we explicitly incorporate both, variable aggregate size and ballasting through heterogeneous composition. Under the above assumptions, we derive the terms for $b$, $m(d)$, $w_s(d)$, $d_{\min}$ and $d_{\max}$ in the following sections.

### 2.2.1 Representation of aggregate microstructure and heterogeneous composition

Marine aggregates are porous (Alldredge and Gotschalk, 1990) and feature a self-similar microstructure which can be described via a fractal dimension $d_f$ (Logan and Wilkinson, 1990; Kranenburg, 1994). A $d_f = 1$ would depict a chain of aggregate constituents and a $d_f = 3$ describes a solid sphere. Consequently, the mass of an aggregate, $m(d)$, grows disproportionately to the aggregates volume and can be expressed as

$$m(d) = m_f\,d^{d_f} \tag{4}$$

where $m_f$ is a mass-factor for the smallest entity. Thus, the density of an aggregate $\rho_f$ decreases with increasing diameter. Aggregates consist of e. g. phytoplankton cells or coccolithophore shells (Alldredge and Gotschalk, 1990) which we consider as spherical primary particles. Primary particles exhibit their own density $\rho_p$ and diameter $d_p$. Taking the fractal scaling of





aggregate mass into account, the excess density of an aggregate $\Delta\rho_f = \rho_f - \rho$ with respect to surrounding fluid density $\rho$ can be expressed as (Kranenburg, 1994)

$$\Delta\rho_f = (\rho_p - \rho)\left(\frac{d_p}{d}\right)^{3-d_f} \qquad \text{for: } d \geq d_p \tag{5}$$

Furthermore, the aggregate porosity, $\phi$, is defined as

$$\phi = 1 - \left(\frac{d_p}{d}\right)^{3-d_f} \qquad \text{for: } d \geq d_p \tag{6}$$

and hence, both, excess density and porosity, are regulated by the fractal dimension and primary particle size. The $\Delta\rho_f$ can be introduced to the terminal sinking velocity, $w_s$,

$$w_s = \sqrt{\frac{4}{3}\frac{\Delta\rho_f}{\rho}\frac{g\,d}{c_D}} \tag{7}$$

For small particle Reynolds numbers, $Re_{\mathrm{p}} = w_s\, d/\nu < 0.1$, the drag coefficient is $c_D = 24/Re_{\mathrm{p}}$ and the well known Stokes (1851) sinking velocity, $w_s$, becomes

$$w_s = \frac{1}{18\,\mu}(\rho_p - \rho)\,g\,d_p^{3-d_f}\,d^{d_f-1} \tag{8}$$

where $\mu$ and $\nu$ are molecular dynamic and kinematic viscosity, respectively, and $g$ is the gravitational acceleration constant. However, this approach assumes homogeneous, mono-sized primary particles while it displays the potential importance of primary particle size and density as well as aggregate microstructure for sinking velocity. To better represent aggregates in natural systems, the heterogeneity of primary particles was thus far considered either for size or density (Jackson, 1998; Maggi, 2009; Khelifa and Hill, 2006). With M[4]AGO, we represent aggregates composed of poly-dense, poly-sized primary particles under the assumption of a singled value fractal dimension throughout the aggregate size spectrum. This allows for representing heterogeneous primary particles such as diatom frustules, coccoliths, dust particles, and detritus as principal components of marine aggregates.

Bushell and Amal (1998) derived a representation of the mean primary particle size

$$\langle d_p \rangle = \left(\frac{\sum_i n_i\, d_{p,i}^3}{\sum_i n_i\, d_{p,i}^{d_f}}\right)^{\frac{1}{3-d_f}} \tag{9}$$

for an aggregate that is composed of $n_p = \sum_i n_i$ mono-dense spherical primary particles of different diameters $d_{p,i}$. Poly-sized formed aggregates disobey the traditional mass fractal relationship, but the fractal nature continues to emerge in a power law scaling for the mass present in a radial shell from an occupied point in the aggregate (Bushell and Amal, 1998). The approach of Bushell and Amal (1998) conserves the size of the aggregate and the encapsulated solid volume of the primary particles, while $n \cdot \langle d_p \rangle^3 = \sum_i n_i\, d_{p,i}^3$ with $n \neq \sum_i n_i$, and thus the porosity of the aggregate is unimpaired.

Under the assumption that aggregates feature the same composition and hence same heterogeneity in a size spectrum, the aggregate-composing primary particle types are always the same for any aggregate of diameter $d$ in a unit volume and thus




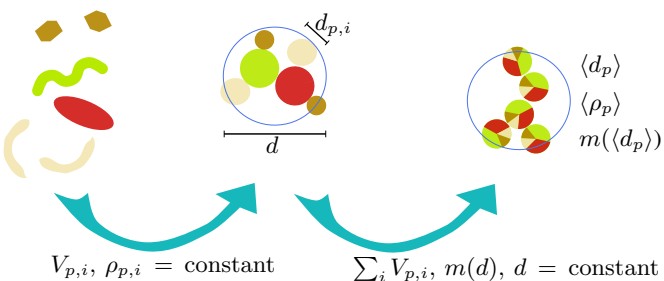

**Figure 1.** Underlying assumptions for the representation of aggregates composed of poly-dense, poly-sized primary particles. Primary particles, like dust particles, coccoliths, and diatom frustules (left) are assumed to be spherical and exhibit their characteristic density (middle). Once aggregated, we assume the diameter of the aggregate being constant and the total volume and mass of primary particles to be preserved (right).

$n_i/n_p = \text{constant}$. This further implies that the ratio $K_i$, between the total number of one primary particle type, $n_{i,\text{tot}}$, to the total number of primary particles, $\sum n_{i,\text{tot}}$, in a unit volume is equal to the ratio found in an individual aggregate

$$K_i = \frac{n_i}{\sum_i n_i} = \frac{n_{i,\text{tot}}}{\sum_i n_{i,\text{tot}}} \tag{10}$$

Rewriting, $n_i = K_i \sum_i n_i$ and inserting in Eq. (9) gives

$$\langle d_p \rangle = \left( \frac{\sum_i K_i \, d_{p,i}^3}{\sum_i K_i \, d_{p,i}^{d_f}} \right)^{\frac{1}{3-d_f}} \tag{11}$$

where the factors $K_i$ can be expressed in HAMOCC via the concentration of each aggregate-forming tracer $C_i$. Namely, we consider the HAMOCC tracers detritus, opal, calcite, and dust in taking part in the formation of heterogeneously composed aggregates. Calculating the number of primary particles from the tracer concentration requires the molecular concentration to mass factor, $R_i$, the tracer-related primary particle diameter, volume $V_{p,i}$, and density $\rho_{p,i}$

$$n_{i,\text{tot}} = \frac{C_i \, R_i}{\rho_{p,i} \, V_{p,i}} \tag{12}$$

The advantage of Eq. (11) is that it allows us to determine the mean primary particle diameter in HAMOCC while solid volume and density of primary particles are conserved. Ensuring mass conservation, we introduce the volume-weighted primary particle mean density

$$\langle \rho_p \rangle = \frac{\sum_i n_{i,\text{tot}} \, V_{p,i} \, \rho_{p,i}}{\sum_i n_{i,\text{tot}} \, V_{p,i}} \tag{13}$$

and hence, the mass of a mean primary particle can be written as (see also Fig. 1)

$$m(\langle d_p \rangle) = \frac{1}{6} \pi \langle d_p \rangle^3 \, \langle \rho_p \rangle \tag{14}$$

Substituting Eq. (14) into Eq. (4), we derive the mass-factor for heterogeneous aggregates $m_f = \frac{1}{6} \pi \langle d_p \rangle^{3-d_f} \langle \rho_p \rangle$. The derivation of the mean primary particle diameter (Eq. 11), density (Eq. 13), and mass (Eq. 14) allows for applying common fractal





laws for the calculation of aggregate mass, density, and thus sinking velocity. Hence, $w_s(d, \rho_p, d_p, \ldots)$ can be expressed as $w_s(d, \langle \rho_p \rangle, \langle d_p \rangle, \ldots)$. For a single type of primary particle, all underlying equations reduce to the traditional fractal scaling relationship.

### 2.2.2 Mean sinking velocity of marine aggregates

In the preceding section, we derived a formulation for the mean primary particle size, Eq. (11), which we apply as a lower integration bound in Eq. (3), and hence, $d_{\min} = \langle d_p \rangle$. The maximum aggregate diameter of the size spectrum, $d_{\max}$, is limited by fragmentation of particles. Several mechanisms can cause fragmentation of aggregates. Flow-induced turbulent shear has been suggested as the dominant process in the upper ocean, where turbulent shear reaches typical values of order $1\,\mathrm{s}^{-1}$ (Jackson, 1990). By contrast, Alldredge et al. (1990) showed that marine aggregates often withstand oceanic turbulence conditions and

suggested biological processes as mediating factor for shaping the size distribution. Zooplankton also generates turbulent shear that is strong enough to rupture aggregates (Dilling and Alldredge, 2000). Hill (1998) proposed an alternative control on aggregate size, namely the sinking of aggregates that produces shear of the same order of magnitude as ambient turbulent shear in the ocean (Bagster and Tomi, 1974; Adler, 1979; Alldredge et al., 1990). Sinking could thus cause fragmentation in deeper regions of the ocean, where turbulent shear is small ($O(0.01\,\mathrm{s}^{-1})$; McCave, 1984; Waterhouse et al., 2014). Since modeling

of particle-reactive thorium suggests continued fragmentation during particle descent in the deep ocean (Lam and Marchal, 2015), we adopt the hypothesis of sinking-induced fragmentation and limit the size distribution based on the particle Reynolds number, $Re_\mathrm{p}$,

$$Re_\mathrm{p}(d, \langle d_p \rangle, \langle \rho_p \rangle, d_f, \nu) = \frac{d\, w_s(d, \langle d_p \rangle, \langle \rho_p \rangle, d_f)}{\nu} \tag{15}$$

Kiørboe et al. (2001) suggested the particle Reynolds number being in a typical range up to $Re_p = 20$ while e. g. Alldredge

and Gotschalk (1988) measured particle Reynolds numbers up to $Re_p = 32$. Aggregates thus can exhibit larger $Re_\mathrm{p}$ than the laminar case ($Re_\mathrm{p} < 0.1$). The drag coefficient, $c_D$, in Eq. (7) can be represented by the expression for solid spheres of White (2005), valid up to $Re_\mathrm{p} < 10^5$

$$c_D = \frac{24}{Re_\mathrm{p}} + \frac{6}{1 + \sqrt{Re_\mathrm{p}}} + 0.4 \tag{16}$$

We approximate this representation by (Jiang and Logan, 1991)

$$c_D(Re_\mathrm{p}) = a_j\, Re_\mathrm{p}^{-b_j} \tag{17}$$

to avoid iteration and to allow for analytical solution of Eq. (3). Applying the parameter values of $a_{j=1} = 24.00$, $b_{j=1} = 1$ for : $Re_\mathrm{p} \leq 0.1$; $a_{j=2} = 29.03$, $b_{j=2} = 0.871$ for : $0.1 < Re_\mathrm{p} \leq 10$ and $a_{j=3} = 14.15$, $b_{j=3} = 0.547$ for : $10 < Re_\mathrm{p} \leq 100$ introduces maximum errors less than $10\,\%$ compared to Eq. (16) for $Re_\mathrm{p} < 100$ (Jiang and Logan, 1991).

By introducing Eq. (17) in Eq. (7), the approximation for the sinking velocity becomes

$$w_s = \left( \frac{4}{3} \frac{\rho_f(d, \langle \rho_p \rangle, \langle d_p \rangle, d_f) - \rho}{\rho} \langle d_p \rangle^{3 - d_f}\, g\, \frac{d^{b_j + d_f - 2}}{a_j\, \nu^{b_j}} \right)^{\frac{1}{2 - b_j}} \tag{18}$$



By substituting Eq. (18) into Eq. (15), the piece-wise integration boundaries, $d_j(Re_{\mathrm{p},j=0..3} = 0, 0.1, 10, Re_{\mathrm{crit}})$, according to the $c_D$ approximation for Eq. (3), become a function of $Re_{\mathrm{p}}$

$$d_j(Re_{\mathrm{p},j}) = \frac{(Re_{\mathrm{p},j}\,\nu)^{\frac{2-b_j}{d_f}}}{\left(\frac{4}{3}\frac{\langle\rho_p\rangle - \rho}{\rho}\langle d_p\rangle^{3-d_f}\,g\,\frac{1}{a_j\,\nu^{b_j}}\right)^{\frac{1}{d_f}}} \tag{19}$$

where $Re_{\mathrm{crit}}$ is the globally fixed critical $Re_{\mathrm{p}}$ for fragmentation. Consequently, the concentration-weighted mean sinking velocity, Eq. (3), can then be expressed as

$$\langle w_s \rangle = \frac{\sum\limits_{j=0}^{2}\left(\int\limits_{\max(\langle d_p\rangle, d_j(Re_{\mathrm{p},j}))}^{d_{j+1}(Re_{\mathrm{p},j+1})} n(d)\,m(d)\,w_s(d, a_{j+1}, b_{j+1})\,\mathrm{d}d\right)}{\int\limits_{\langle d_p\rangle}^{d_{\max}} n(d)\,m(d)\,\mathrm{d}d} \tag{20}$$

where $d_{\max} = d_j(Re_{\mathrm{crit}})$ is the maximum diameter of aggregates.

### 2.2.3 The particle distribution slope, $b$

Observed aggregate size spectra in the ocean exhibit a spatio-temporal dependent slope ranging between approximately 3.2 to 5.4 (DeVries et al., 2014) or even lower ($\approx 2$; Guidi et al., 2009). A smaller slope parameter, $b$, translates to more large aggregates relative to a larger $b$ and enhances mean sinking velocity. The evolution of the particle size spectra underlies the interacting processes of growth and decay of phytoplankton, aggregation, fragmentation and sinking of aggregates. Instead of modeling the processes of aggregation and fragmentation explicitly or prescribing $b$, we assume dynamic steady state for the slope of the number distribution. According to dimensional analysis, the slope of the number distribution in dynamic steady state depends on the fractal dimension of aggregates and the process of aggregation, aggregation due to shear, differential sinking and Brownian motion (Jiang and Logan, 1991). Aggregation due to Brownian motion is only relevant for particles smaller $\approx 1\,\mu\mathrm{m}$ (McCave, 1984) which we neglect here. We further assume that aggregation in the majority of the global ocean is dominated by differential settling and express the particle distribution slope, $b$, as (Jiang and Logan, 1991)

$$b = \frac{1}{2}\left(3 + d_f + \frac{2 + d_f - \min(2, df)}{2 - b_J}\right) \tag{21}$$

where $b_J$ is a fixed parameter for the sinking velocity dependency on the particle Reynolds number that we fix for simplicity to $b_J = b_{j=2}$. The assumption of differential settling-dominated aggregation is likely violated in the euphotic zone, where shear aggregation is probably more relevant and steady state assumption is questionable, which we will address in the discussion (Sec. 3.10).

### 2.2.4 Heuristic approach to variable aggregate stickiness and fractal dimension

Adhesion properties of particles affect the fractal structure of aggregates and the collision efficiency ('stickiness') of particles (Meakin, 1988; Liu et al., 1990). Theoretical studies show that the stronger the surface adhesive forces are, the higher is the





stickiness of particles and the smaller is the intrusion of particles and particle clusters into each other (Liu et al., 1990). As a result, this leads to a looser structure which translates to a small fractal dimension. Stickiness of phytoplankton is species-specific (Hansen and Kiørboe, 1997) and depends on the growth phase (Simon et al., 2014). Furthermore, phytoplankton releases extracellular polymeric substances (EPS; Decho, 1990) such as transparent exopolymer particles (TEPs) which are

suggested as aggregation-priming, sticky materials (Azam and Malfatti, 2007; Thornton, 2002; Passow, 2002). The resulting fractal dimension is typically determined as one value across a particle size spectra. We thus assign a single fractal dimension to an aggregate population and depict the linkage between stickiness and fractal dimension in a qualitative manner. We attribute a stickiness value, $\alpha_i$, to each of HAMOCCs sinking tracers and calculate a mean stickiness for the aggregates that is then mapped to a fractal dimension. Since adhesion, and thus stickiness, are a surface property, we calculate the mean stickiness

$$\langle \alpha \rangle = \frac{1}{A} \sum_i n_i A_i \alpha_i \quad \text{where}: A = \sum_i n_i A_i \tag{22}$$

weighted by the primary particles surfaces $A_i \propto d_{p,i}^2$. We map the mean stickiness to a range between zero and one

$$\langle \alpha \rangle_{\text{map}} = \frac{\langle \alpha \rangle - \alpha_{\min}}{\alpha_{\max} - \alpha_{\min}} \tag{23}$$

where $\alpha_{\min} = \min(\alpha_i)$ and $\alpha_{\max} = \max(\alpha_i)$.

Nicolás-Carlock et al. (2016) introduced a scaling parameter for the effective aggregation range in microscopic aggregation

models to stipulate a defined fractal dimension across aggregate sizes. The scaling parameter can be perceived as an indicator of stickiness that defines the effective aggregation range, i. e. higher stickiness results in a larger effective aggregation range. We introduce as an analogy for the dependency of the fractal dimension on the scaling parameter of Nicolás-Carlock et al. (2016) a transfer function for the mapped mean stickiness to fractal dimension, $d_f(\langle \alpha \rangle_{\text{map}})$,

$$d_f(\langle \alpha \rangle_{\text{map}}) = d_{f,\max} \exp(\beta_f \langle \alpha \rangle_{\text{map}}) \tag{24}$$

with $\beta_f = \log(d_{f,\min}/d_{f,\max})$, where $d_{f,\min}$ and $d_{f,\max}$ are parametrized minimum and maximum fractal dimension of aggregates. Modeled stickier aggregates thus exhibit lower fractal dimensions than non-sticky particles which is in qualitative agreement with previous studies (Meakin, 1988; Liu et al., 1990; Block et al., 1991; Nicolás-Carlock et al., 2016).

### 2.2.5 Diatoms as a special case of primary particles

Diatoms are silicifying phytoplankton that possesses a hollow opal skeleton, the diatom frustule, and are thus different from a

homogeneous, solid primary particle like coccoliths. Diatoms feature a wide range of sizes, with about a few microns to millimeters (Armbrust, 2009). Since sinking velocities of aggregates are proportional to their diameter, primary particle density and size (Eq. 8), aggregate-incorporated large diatom shells likely enhance sinking velocity of particles. Indeed, un-remineralized, intact diatom frustules were even found in deeper regions of the ocean (Assmy et al., 2013), which is in agreement with previously found high sinking speeds of large diatom aggregates (Alldredge and Gotschalk, 1988). We therefore explicitly account

for diatom shells by treating them as hollow opal spheres, filled i) with detritus, and ii) increasing water content with ongoing remineralization while sinking (see Fig. 2).





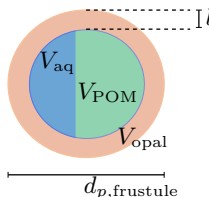

**Figure 2.** Diatom frustule and the remineralization state-dependent composition of the void.

The opal volume of a modeled diatom is

$$V_{\text{opal}} = \frac{1}{6}\pi \left(d_{p,\text{frustule}}^3 - (d_{p,\text{frustule}} - 2\,l)^3\right) \tag{25}$$

where $d_{p,\text{fustule}}$ is the diameter of the diatom, whose opal shell thickness $l$ is expressed in terms of the fixed opal-to-phosphorus formation ratio. The number of diatom frustules per unit volume

$$n_{\text{frustule}} = \frac{[\text{opal}]\,R_{\text{opal}}}{\rho_{\text{opal}}\,V_{\text{opal}}} \tag{26}$$

can therefore be deduced from the present opal concentration, $[\text{opal}]$, and the opal mol-to-weight factor $R_{\text{opal}}$ according to Eq. (12). We assume that the modeled detritus pool can be split into a free external, non-diatom and a diatom frustule-related, void-filling detritus part. We further assume that the external pool is remineralized before the intra-cellular pool of volume $V_{\text{POM}}$ and thus neglect cell lysis observed prior to aggregation (Armbrecht et al., 2014) and rather assume mineral protection of detritus (Hedges et al., 2001). If more detritus is remineralized than the frustules void would hold, it is replaced with the respective volume of water $V_{aq}$ of density $\rho$. The frustule density thus is

$$\rho_{\text{frustule}} = \frac{V_{\text{opal}}\,\rho_{\text{opal}} + V_{\text{POM}}\,\rho_{\text{POM}} + V_{\text{aq}}\,\rho}{V(d_{p,\text{frustule}})} \tag{27}$$

During growth and decay, diatoms excrete TEPs which are positively buoyant and possess a density of about $\rho_{\text{TEP}} = 700\,\text{kg}\,\text{m}^{-3}$ to $840\,\text{kg}\,\text{m}^{-3}$ (Azetsu-Scott and Passow, 2004; Mari et al., 2017). TEPs are suggested to play a prominent role in aggregation processes as they are probably sticky and thus enhance aggregation (Dam and Drapeau, 1995; Passow, 2002). In HAMOCC, phytoplankton excretion of TEPs is not resolved explicitly. We therefore treat TEPs virtually and assume a linear dependency of diatom stickiness and density on the freshness of detritus, defined as the mass ratio between the actual amount of detritus $m_e$ and the potential mass of detritus linked to opal production $m_{\text{potential}}$. An additional underlying assumption is that TEPs are remineralized with the same rates as normal detritus. Hence, we define the stickiness of diatoms as

$$\alpha_{\text{diatom}} = \frac{m_e}{m_{\text{potential}}}\alpha_{\text{TEP}} + \left(1 - \frac{m_e}{m_{\text{potential}}}\right)\alpha_{\text{opal}} \tag{28}$$

for $m_{\text{potential}} > 0$, where $\alpha_{\text{TEP}}$ and $\alpha_{\text{opal}}$ are the stickiness of TEPs and pure opal, respectively. The density of diatoms becomes

$$\rho_{\text{diatom}} = \frac{\rho_{\text{frustule}}\,m_{\text{potential}} + \rho_{\text{TEP}}\,m_e}{m_{\text{potential}} + m_e} \tag{29}$$



TEPs thus have a twofold effect on aggregates in our model: i) TEPs increase stickiness and loosen the aggregate structure, thus fractal dimension of aggregates is small, and ii) TEPs decrease the fresh diatom frustules density and thus add buoyancy without violating tracer mass conservation (see also model discussion Sec. 3.10).

### 2.3 Temperature-dependent opal dissolution & POC remineralization

Marine aggregates tie heterogeneous components together that are disparately remineralized or dissolved. By contrast, in the standard model, detritus, opal and $CaCO_3$ were sinking separately from each other and the global remineralization and dissolution rates are tuned independently because the processes are artificially decoupled. In $M^4AGO$, remineralization of detritus and dissolution of, in particular, opal are tightly linked through the same sinking velocity which let us to re-evaluate and revise the formulations for opal dissolution and remineralization.

Opal dissolution is temperature-dependent (Ragueneau et al., 2000, 2006) and is microbially mediated (Bidle et al., 2002). Intact diatom frustules are protected from dissolution by an organic matrix (Lewin, 1961). Once the organic protection surrounding the silicate frustule becomes utilized by temperature-dependent microbes, they initiate and mediate the dissolution of opal (Bidle et al., 2002). Hence, opal dissolution follows a sequential process: i) an initial temperature-dependent remineralization of the organic coating of the silicate frustule and ii) the microbially mediated dissolution of opal with a temperature dependency of $Q_{10} \approx 2.3$ (Bidle et al., 2002). We here focus on the temperature-dependent microbially mediated dissolution

and introduce a $Q_{10}$ temperature-dependent opal dissolution

$$\frac{\partial}{\partial t}[\text{opal}]\Big|_{\text{dissolution}} = -r_{\text{opal}} Q_{10,\text{opal}}^{\frac{T-T_{\text{ref,opal}}}{10}} [\text{opal}] \tag{30}$$

In the standard version, we remain with the former linearly temperature-dependent opal dissolution (Ragueneau et al., 2000; Segschneider and Bendtsen, 2013).

Analogously to opal, we incorporate a temperature-dependent $Q_{10}$ factor to aerobic POC remineralization (Dell et al., 2011; Mislan et al., 2014) which depends on oxygen concentration, $[O_2]$, (Mauritsen et al., 2019), where $K_{O_2}$ is the half saturation constant in Michaelis-Menten kinetics

$$R_{\text{POC,remin}} = -r_{\text{POC}} \frac{[O_2]}{K_{O_2} + [O_2]} Q_{10,\text{POC}}^{\frac{T-T_{\text{ref,POC}}}{10}} \tag{31}$$

We keep the anaerobic remineralization temperature-independent since we do not expect temperature shifts in ocean depths,

where oxygen minimum zones appear in HAMOCC. In the standard run, the remineralization rates are temperature-independent ($Q_{10,\text{POC}} = 1$).

### 2.4 Model setup, parametrization & evaluation

#### 2.4.1 General model setup

The $M^4AGO$ sinking scheme was implemented in HAMOCC which is coupled to MPI-OM (Jungclaus et al., 2013). For the

flow of calculations in the $M^4AGO$ sinking scheme, see Fig. 3. We run both, the standard and the $M^4AGO$ run, in a GR15/L40-OMIP setup. This translates to a horizontal resolution of about $1.5\,^\circ$, 40 uneven vertical layers with highest resolution in the first




**Figure 3.** Flow diagram of calculations for the $M^4AGO$ sinking scheme carried out at every ocean grid point and time step. Marine aggregates in $M^4AGO$ are composed of spherical primary particles derived from HAMOCC tracers. Primary particles featuring size, density, and stickiness are: detritus, diatom frustules, coccoliths ($CaCO_3$), and dust minerals. ❶ The number of diatom frustules, Eq. (26), related diatom density, Eq. (29), and stickiness, Eq. (28), are estimated from opal and detritus concentration. Diatoms are then considered as primary particles which feature particular characteristics. ❷ The remaining detritus is considered as detritus primary particles. ❸ The calculation of the fractal dimension, Eq. (24), and ❹, the calculations of mean primary particle size, Eq. (11), and density, Eq. (13) are carried out. ❺ The fractal dimension determines the number distribution slope, Eq. (21). ❻ The minimum, $d_{\min} = \langle d_p \rangle$, and maximum aggregate diameter, Eq. (19), are estimated. ❼ The mean sinking velocity, Eq. (20), can eventually be determined, with which the tracers sink.

few hundred meters of the ocean. OMIP is a climatological daily atmospheric forcing (Röske, 2005). The loss of POM, opal, and $CaCO_3$ due to sedimentation and subsequent burial was accounted for through homogeneously applied weathering rates which were adjusted accordingly, namely for the standard / $M^4AGO$ run: $CaCO_3 \approx 17.2 / 26.5$ T mol C yr$^{-1}$, dissolved organic phosphorus $\approx 99.6 / 101.5$ G mol P yr$^{-1}$, and silicate $\approx 3.2 / 2.3$ T mol Si yr$^{-1}$. We start the $M^4AGO$ run from the standard run in steady state and spin it up until steady state is reached in surface and mesopelagic waters, which translates to 1700 model





years. Through the long overturning times of the global ocean, we still see drifts of nutrient concentrations in deep, old North Pacific waters at this state (i. e. on average $\sim 7.1\,\mu\mathrm{mol}\,\mathrm{P}\,\mathrm{m}^{-3}\,\mathrm{century}^{-1}$ below 2000 m, which amounts to a centennial change of about 0.25 %). We neglect this drift as we focus on the aggregate properties and their effects on POC fluxes throughout the euphotic and mesopelagic zone.

### 2.4.2 Parameters of the M$^4$AGO scheme

The M$^4$AGO sinking scheme introduces a set of new parameters, in particular the primary particle characteristics, which require constraining and tuning (summarized in Tab. 1).

We applied HAMOCCs standard sediment densities for opal, CaCO$_3$, and dust to the densities of primary particles, namely $\rho_{\mathrm{opal}} = 2200\,\mathrm{kg}\,\mathrm{m}^{-3}$ , $\rho_{\mathrm{calc}} = 2600\,\mathrm{kg}\,\mathrm{m}^{-3}$ and $\rho_{\mathrm{dust}} = 2600\,\mathrm{kg}\,\mathrm{m}^{-3}$. For suspended detritus, we chose $\rho_{\mathrm{det}} = 1100\,\mathrm{kg}\,\mathrm{m}^{-3}$
(Fettweis, 2008). As density of TEPs, we applied $\rho_{\mathrm{TEP}} = 800\,\mathrm{kg}\,\mathrm{m}^{-3}$, which is within the measured range of $700\,\mathrm{kg}\,\mathrm{m}^{-3}$ to $840\,\mathrm{kg}\,\mathrm{m}^{-3}$ (Azetsu-Scott and Passow, 2004).

While the density of primary particles is comparably well constrained, the adhesion forces of primary particles, namely related stickiness and fractal dimension of aggregates, are less studied and are weakly constrained. There is yet no standardized way to investigate stickiness, fractal dimension, and their interdependence for aggregates in natural waters. Stickiness is
experimentally defined as the interparticle attachment rate divided by the interparticle collision rate. Uncertainties in any of the two rates, e. g. due to ignoring fractal structure of aggregates, aggregate permeability, etc., directly affects the calculated stickiness (Filella, 2007). In addition, stickiness is phytoplankton species-specific (Hansen and Kiørboe, 1997) and depends on the growth phase (Simon et al., 2014). The methodological limitations, the heterogeneity of marine aggregate constituents and their variable formation process lead to a wide spread of indirectly inferred values for stickiness and fractal dimension
(see e. g. Filella, 2007, for a broader overview). Diatom aggregates seem to feature a wide spread of fractal dimensions ranging from $d_f \approx 1.26$ to $d_f \approx 2.46$ (Alldredge and Gotschalk, 1988; Guidi et al., 2008) and $\langle \alpha \rangle$ of about 0.03 to 0.88 (Kiørboe et al., 1990; Dam and Drapeau, 1995; Alldredge and McGillivary, 1991). Indirectly inferred fractal dimension for re-worked aggregates exhibit values of $d_f \approx 2.26$ to $d_f \approx 2.46$ (Guidi et al., 2008) and mineral-dominated aggregates also feature high fractal dimensions of about $d_f \approx 2$ (Winterwerp, 1998) to $d_f \approx 2.6$ (Kranenburg, 1999) and low stickiness of $\langle \alpha \rangle \approx O(10^{-2})$
(Tambo and Hozumi, 1979). Since stickiness of *in situ* primary particles is seldom measured, we choose it to our best knowledge and order the stickiness for modeled primary particles according to the mean stickiness of observed aggregate types: $\alpha_{\mathrm{dust}} < \alpha_{\mathrm{opal}} \leq \alpha_{\mathrm{CaCO_3}} < \alpha_{\mathrm{det}} < \alpha_{\mathrm{TEP}}$ (see also Tab. 1). Hence, detritus and TEPs-rich aggregates are modelled with a loose structure and low fractal dimension, while degraded, mineral-rich aggregates are more compact and thus feature a higher fractal dimension (see Eq. 24) which is in congruence with the present conceptual understanding of aggregates becoming compacted
during their descent (Mari et al., 2017). For the minimum and maximum fractal dimension of marine aggregates, we chose conservative bounds of 1.6 (Logan and Alldredge, 1989; Li and Logan, 1995; Alldredge, 1998) and 2.4 (in the range of $d_f \approx 2.26$ to 2.46 for reworked aggregates Guidi et al., 2008) which is well within the observed range of 1.26 (Logan and Wilkinson, 1990) to 2.6 (Kranenburg, 1999) for marine particles.





**Table 1.** Model parameters for $M^4AGO^*$ and, if adjusted in comparison to the standard HAMOCC, the standard values. The value for the Martin curve of POC fluxes in the standard run, (Eq. 1), is $\beta = 1$.

| Parameter | Description | Value in HAMOCC standard/$M^4$AGO | Literature range | Unit | References |
|---|---|---|---|---|---|
| $\alpha_{det}$ | stickiness of detritus particles | 0.10 | – | – | Hansen and Kiørboe (1997); Simon et al. (2014); |
| $\alpha_{calc}$ | stickiness of CaCO₃ particles | 0.09 | for $\langle\alpha\rangle$ of | – | Filella (2007); Kiørboe et al. (1990); Dam and Drapeau (1995); Alldredge and McGillivary (1991); |
| $\alpha_{dust}$ | stickiness of dust mineral particles | 0.07 | aggregates, | – | |
| $\alpha_{opal}$ | stickiness of opal particles | 0.08 | see text | – | Tambo and Hozumi (1979) |
| $\alpha_{TEP}$ | stickiness of TEP particles | 0.19 | – | – | |
| $b_J$ | slope parameter | 0.871 | 0.547, 0.871, 1 | – | Jiang and Logan (1991) |
| $c_{max}$ | maximum fraction of CaCO₃ production | 0.20/0.18 | – | – | |
| *$d_{p,dust}$ | primary particle diameter of dust minerals | 2 | < 1 - ≈20 | μm | Maher et al. (2010); Mahowald et al. (2014) |
| *$d_{p,calc}$ | primary particle diameter of CaCO₃ | 3 | 1.5-15.5 | μm | Young and Ziveri (2000); Henderiks (2008) |
| *$d_{p,frustule}$ | primary particle diameter of diatom frustule | 20 | 6-29 /O(μm to mm) | μm | Litchman et al. (2009) |
| *$d_{p,det}$ | primary particle diameter of detritus | 4 | – | μm | |
| $d_{f,min}$ | minimum fractal dimension of aggregates | 1.6 | 1.26 - 2.60 | – | Logan and Wilkinson (1990); Kranenburg (1999) |
| $d_{f,max}$ | maximum fractal dimension of aggregates | 2.4 | – | – | Bidle et al. (2002); based on their values |
| *$Q_{10,opal}$ | Q10 factor for opal dissolution | 2.6 | 2.3 (-1.8-17 °C) - 2.9 | – | Mislan et al. (2014); Dell et al. (2011) |
| *$Q_{10,det}$ | Q10 factor for detritus remineralization | 1.0/2.1 | 2-3 | – | Bidle et al. (2002) |
| *$r_{opal}$ | dissolution rate opal | 0.010/0.023 | ᵃ0.001-0.138 (-1.8 - 33 °C) | d⁻¹ | Iversen and Ploug (2010) |
| *$r_{POC}$ | remineralization rate POC | 0.026/0.120 | ᵃ0.08-0.21 (at 15 °C) | d⁻¹ | Azetsu-Scott and Passow (2004) |
| $\rho_{TEP}$ | density of TEP | 800 | 700-840 | kg m⁻³ | Fettweis (2008) |
| $\rho_{det}$ | density of detritus | 1100 | 900-1300 | kg m⁻³ | Fettweis (2008) |
| $\rho_{opal}$ | density of opal | 2200 | – | kg m⁻³ | |
| $\rho_{calc}$ | density of CaCO₃ | 2600 | 2600-2800 | kg m⁻³ | clay, quartz; Fettweis (2008) |
| $\rho_{dust}$ | density of dust | 2600 | 2300-2800 | kg m⁻³ | |
| $R_{opal}$ | Silicon mol-to-opal-mass factor | 60 | – | kg SiO₂ (kmol Si)⁻¹ | |
| $R_{calc}$ | carbon mol-to-CaCO₃-mass factor | 100 | – | kg CaCO₃ (kmol C)⁻¹ | |
| $R_{det}$ | P mol-to-detritus-mass factor | 3166 | – | kg POM (kmol P)⁻¹ | Takahashi et al. (1985) |
| $R_{Si:P}$ | silicate to phosphate production ratio | 25 | – | mol Si (mol P)⁻¹ | |
| $Re_{crit}$ | critical particle Reynolds number | 20 | O(20-30) | – | Alldredge and Gotschalk (1988); Kiørboe et al. (2001) |
| $T_{ref,x=POC,opal}$ | reference temperature for $Q_{10}$ | 10 | – | °C | |

* : refers to manual tuning of parameter - all other parameters were fixed from beginning

ᵃ measured dissolution and remineralization rates include temperature-dependence, which is represented by the $Q_{10}$ factor in $M^4$AGO





For the primary particle sizes, we conceptually assume that the tracer characteristics of opal and $CaCO_3$ are primarily related to phytoplankton mineral structures such as diatom silicate frustules and the coccoliths of coccolithophores. This implies that modeled zooplankton egests biogenic mineral structures of algae while their own larger mineral body structures play, in numbers, a minor role in biogenic mineral fluxes. Coccolith diameters range from about 1.5 µm to 15.5 µm (Young and Ziveri,

2000; Henderiks and Pagani, 2008; Henderiks, 2008). The globally ubiquitous coccolithophore *Emiliana Huxleyi* (Read et al., 2013) exhibits coccoliths of about 3 µm to 4 µm in diameter (Young and Ziveri, 2000), and hence, we set $d_{p,\mathrm{calc}} = 3\,\mu m$, which is thus at the lower bound of the observed range while accounting for the volumetric density effect of non-spherical plate-like coccoliths. Diatoms frustules feature sizes of a few micrometers to millimeters (Armbrust, 2009) with a dominant size range of about 12 µm to 58 µm (Litchman et al., 2009, equivalent spherical diameter of body volumes $10^3\,\mu m^3$-$10^5\,\mu m^3$). We define the

diatom frustule size as $d_{p,\mathrm{frustule}} = 20\,\mu m$. The primary particle size of detritus is weakly constrained and likely ranges from sub-micrometer of microgels and bacteria to millimeter scales of zooplankton body structures (Verdugo et al., 2004). We here chose $d_{p,\mathrm{det}} = 4\,\mu m$. Aeolian dust particles features a typical size ranging from submicron to about 20 µm in size with a mass median diameter of about 1.5 µm to 3 µm (Maher et al., 2010; Mahowald et al., 2014). In summary, we assumed the following order of primary particle sizes for the tracers: $d_{p,\mathrm{dust}} < d_{p,\mathrm{calc}} \leq d_{p,\mathrm{det}} < d_{p,\mathrm{frustule}}$.

The size distribution-limiting maximum aggregate diameter, $d_{\mathrm{max}}$, is variable in the model domain and depends on the critical particle Reynolds number $Re_{\mathrm{crit}}$. $Re_{\mathrm{crit}}$ and its potential dependency on aggregate properties is weakly constrained which lets us fix the value globally to $Re_{\mathrm{crit}} = 20$, a conservative value compared to measured maximum particle Reynolds number up to 32 by Alldredge and Gotschalk (1988).

   Opal dissolution and detritus remineralization are $Q_{10}$ temperature-dependent in M$^4$AGO. Typically, the $Q_{10}$ factor for

biological processes is in the range between 2 and 3. We here chose $Q_{10,\mathrm{POC}} = 2.1$ (similar to Mislan et al., 2014, who applied $Q_{10,\mathrm{POC}} = 2.0$). For opal, we tuned $Q_{10,\mathrm{opal}} = 2.6$ compared to 2.3 suggested by Bidle et al. (2002).

### 2.4.3 Model tuning and evaluation

The close connection between the parametrized processes of sinking and remineralization, the transfer efficiency and the climatological nutrient field hampers the clear distinction between tuning and evaluation data when comparing the model

results to literature values for transfer efficiency (Weber et al., 2016) and World Ocean Atlas data (Garcia et al., 2014a, b). The transfer efficiency, in combination with the general circulation pattern, affects the nutrient climatology on the long term. In turn, sinking velocity, remineralization and dissolution define the transfer efficiency on time scales of days to months. A direct comparison of modeled to observed sinking velocities and fluxes is challenging, as scale dissimilarities introduce uncertainty for comparisons between models and observations (Bisson et al., 2018). Furthermore, sediment trap data for POC and mineral

fluxes exhibit high uncertainties which complicates model comparisons and even make different parametrizations for vertical fluxes undistinguishable (Cael and Bisson, 2018). As a consequence, we remain with a general evaluation of our model results.

   Long simulations with high computational costs to reach steady state are required in the process of model tuning which prevents from intensive parameter variations. Since the adaptation of the sinking velocity versus the remineralization and dissolution rates, and thus the transfer efficiency, was within a few years, parameter variations aiming at a quantitative agreement




with the transfer efficiency of Weber et al. (2016) enabled a useful strategy to select for promising parameter sets. With respect to the primary particle characteristics, we kept the stickiness values, once chosen to our best knowledge, untouched and minimally varied the primary particle sizes within the range of literature values. We primarily focused on tuning the remineralization and dissolution rates of POC and opal, respectively. We choose this strategy, since reliable remineralization and dissolution

rate measurements are available to evaluate the tuned rates (see ranges in Tab. 1). We aimed at keeping global mean values of primary production, export production of POC, opal and $CaCO_3$ as well as their fluxes to sediment within estimated literature ranges. This let us to minimally vary the fraction of maximum $CaCO_3$ production, $c_{max}$, in $M^4AGO$ compared to the standard run. We here compare and evaluate the $M^4GO$ run with respect to: i) where possible, the standard run, ii) the regional transfer efficiency as derived by Weber et al. (2016), iii) World Ocean Atlas data from the World Ocean Database (Boyer et al., 2013;

Garcia et al., 2014b) and iv) independently to sediment trap-sampled POC and biogenic mineral fluxes compiled by Mouw et al. (2016a, b). The Mouw et al. (2016a, b) data were time- and depth-weighted to receive monthly climatological values for the respective grid boxes in MPIOM, where the sediment trap records were taken. Model results are presented as yearly mean of the last simulated year, unless stated otherwise.

## 3  Results & Discussion

In the following, we evaluate the global net primary production, the export of POC to the mesopelagic zone and associated pattern of biogenic mineral fluxes (Sect. 3.1). In $M^4AGO$, the pattern of POC and associated minerals determine the aggregate properties, which we explicate in Sect. 3.2. In Sect. 3.3, we present the global pattern of transfer efficiency. In Sect. 3.4 and 3.5, we examine the contributions of remineralization rates, sinking velocity and aggregate properties to the transfer efficiency pattern. Thereafter, we evaluate the modeled rain ratios (Sect. 3.6) and the biogeochemical tracer distributions (Sect. 3.7). In

Sect. 3.8, we discuss the consequence of the transfer efficiency pattern on regional $CO_2$ fluxes. Subsequently, we examine the sensitivity of the transfer efficiency to selected model parameters (Sect. 3.9), and conclude with a critical review of underlying assumptions of $M^4AGO$ (Sect. 3.10).

### 3.1  Spatial distribution of POC export fluxes and associated biogenic minerals

The global pattern of the depth-integrated primary production is dominated by global circulation and thus nutrient transport.

The global pattern of annual mean integrated primary production therefore remains similar between the standard and the $M^4AGO$ run (Fig. 4). Globally integrated, the annual net primary production is $\approx 55.3\,\mathrm{Gt\,C\,yr^{-1}}$ in $M^4AGO$ compared to $\approx 44.7\,\mathrm{Gt\,C\,yr^{-1}}$ in the standard run.

The ratio between carbon export out of the euphotic zone at depth $z_0 = 100\,\mathrm{m}$ and the net primary production, NPP,

$$p-\mathrm{ratio}(z) = \frac{\mathrm{POC\ flux\ at\ depth}\,z = z_0}{\mathrm{NPP}} \tag{32}$$

provides an estimate, of how efficient the export is with respect to the net primary production. Globally, about $5.56\,\mathrm{Gt\,C\,yr^{-1}}$ and $6.28\,\mathrm{Gt\,C\,yr^{-1}}$ are exported out of the euphotic zone in $M^4AGO$ and the standard run, respectively. In the standard run,





**Figure 4.** Yearly mean integrated primary production in a) the standard, and b) the M⁴AGO run. The export efficiency ($p-$ratio) in c) the standard, and d) the M⁴AGO run.

the subtropical gyres exhibit $p-$ratios of more than 0.2 and the high latitudes feature lower export efficiency (Fig. 4 c). In the M⁴AGO run, the equatorial Pacific exhibits the lowest export efficiencies, features maximum values of about 0.14-0.16 in the subtropical gyres and about 0.20 in the Arctic region (Fig. 4 d). The M⁴AGO run thus possesses smaller latitudinal variability of the $p-$ratio compared to the standard run.

5    In comparison to previous estimates on global primary production, both model runs are well within the range of $30\,\mathrm{Gt}\,\mathrm{yr}^{-1}$ to $70\,\mathrm{Gt}\,\mathrm{C}\,\mathrm{yr}^{-1}$ and show similar pattern of NPP (Carr et al., 2006). The higher NPP in M⁴AGO is due to the enhanced remineralization rates in surface waters which also leads to the lower export efficiencies in the equatorial and subtropical regions. Estimates of the export efficiency from either satellite, *in situ* observations, or models lead to partly contrasting pattern (e. g. Lutz et al., 2002; DeVries and Weber, 2017; Henson et al., 2011, 2012; Buesseler, 1998; Neuer et al., 2002; Siegel
10   et al., 2014; Cram et al., 2018). In contrast to our two model runs, highest $p-$ratios are suggested to be found in the North Pacific and Antarctic Ocean (e. g. Dunne et al., 2005; Henson et al., 2011; DeVries and Weber, 2017). In the tropical and





subtropical regions, the $M^4$AGO run reduces the bias with respect to previously found low export efficiencies of about 1 % to 10 % (Buesseler, 1998; Neuer et al., 2002).



**Figure 5.** Yearly mean flux ratios of opal ($SiO_2$) to POC in a) the standard, and b) the $M^4$AGO run. Yearly mean flux ratios of $CaCO_3$ to POC in c) the standard, and d) the $M^4$AGO run.

The exported POC is accompanied by biogenic minerals and dust. The export flux ratios between opal and detritus exhibit a clear latitudinal pattern, with high values in the high latitudes and upwelling regions compared to the subtropical gyres (Fig. 5 a, b). Both model simulations show similar pattern of opal to detritus ratio fluxes. Higher $CaCO_3$ to detritus flux ratios are confined to equatorial and subtropical regions where silicate depletion favours calcification (Fig. 5 c, d). $M^4$AGO exhibits a higher $CaCO_3$ to detritus mass flux ratio in the western tropical and subtropical Pacific than the standard run. The higher remineralization in the surface waters in this region reduces the amount of detritus that can coalesce with $CaCO_3$. The spatial distribution of sinking tracers and their ratios prime the marine aggregate properties in $M^4$AGO.

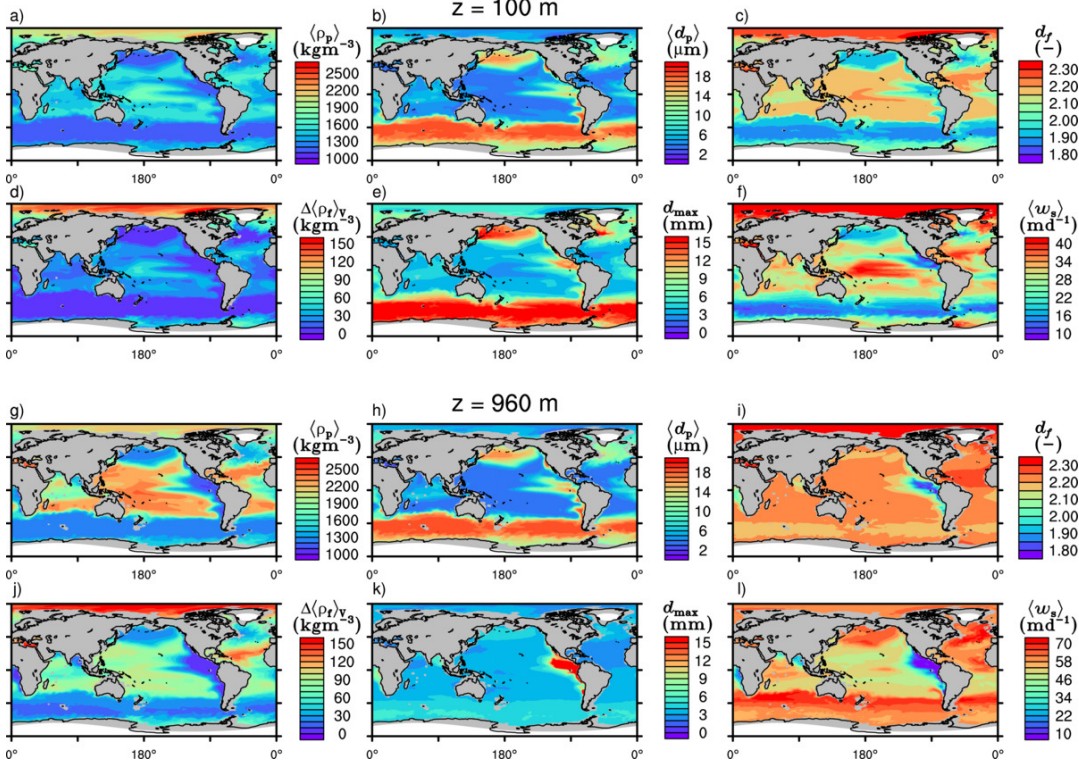

**Figure 6.** Modeled marine aggregate properties at a)-f) 100 m and g)-l) 960 m depth. Mathematical symbols are: $\langle \rho_{\mathrm{p}} \rangle$: mean primary particle density, $\langle d_{\mathrm{p}} \rangle$: mean primary particle diameter; $d_f$: microstructure (fractal dimension); $\Delta \langle \rho_{\mathrm{f}} \rangle_{\mathrm{V}}$: volume-weighted mean excess aggregate density; $d_{\mathrm{max}}$: maximum aggregate diameter; $\langle w_{\mathrm{s}} \rangle$: concentration-weighted mean sinking velocity of aggregates. For the mean stickiness, $\langle \alpha \rangle$, volume-weighted mean porosity, $\langle \phi \rangle_{\mathrm{V}}$, and the number distribution slope, $b$, see Fig. A1.

## 3.2 Spatial distribution of marine aggregate properties

The spatial patterns of detritus and mineral fluxes are reflected in the distribution of aggregate properties (cmp. pattern in Fig. 5 b, d to 6 a-f). The primed characteristics further evolve while aggregates descend through the water column and become remineralized. Hence, the information of the tracer distribution in the euphotic zone propagates into the mesopelagic zone.

5 The mean primary particle density, $\langle \rho_p \rangle$, ranges at export depth (100 m) from about 1100 kg m$^{-3}$ in diatom-dominated regions to 1850 kg m$^{-3}$ in the western Pacific where CaCO$_3$ to detritus export ratios are high (Fig. 5 b, d and 6 a). In the Arctic, some regions harbour $\langle \rho_p \rangle$ of max. 2600 kg m$^{-3}$ where aggregates in our model are dominated by dust particles (Fig. 6 a). Particularly in regions of high CaCO$_3$ to detritus export ratios $\langle \rho_p \rangle$ increases with depth (Fig. 6 a,g and 7 a,d). Accordingly, the volume-weighted mean excess aggregate density, $\Delta \langle \rho_f \rangle_V$, exhibits the same pattern as $\langle \rho_p \rangle$, while it ranges from about 2 kg m$^{-3}$

10 in diatom- to 35 kg m$^{-3}$ in calcifier-dominated regions (Fig. 6 d). Note that we chose $\Delta \langle \rho_f \rangle_V$ to account for the increasing porosity with size, Eq. (6), and thus decreasing aggregate excess density with size, Eq. (5). As a mean value, $\Delta \langle \rho_f \rangle_V$ thus





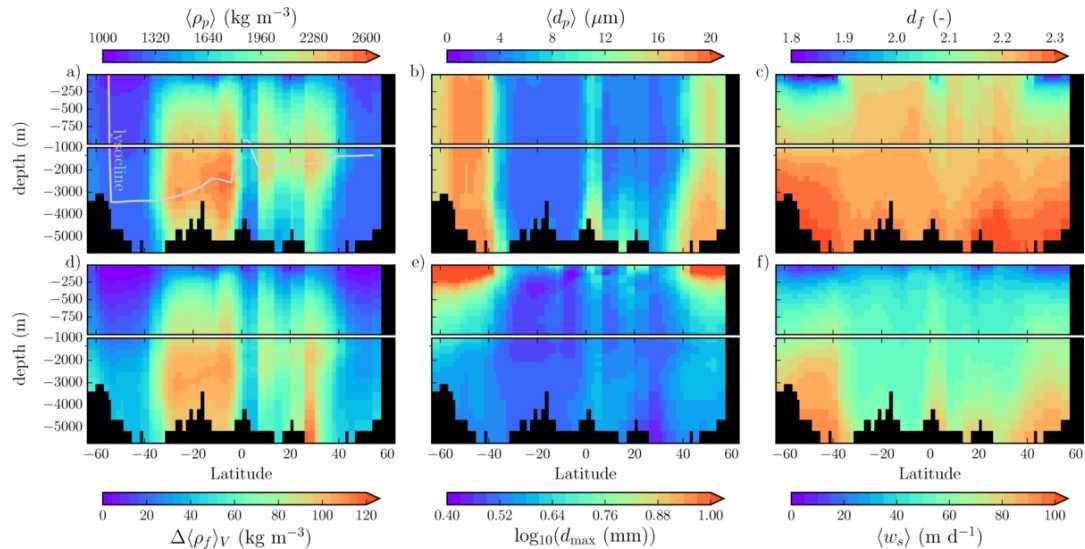

**Figure 7.** Modeled marine aggregate properties on WOA transect P16. For symbol descriptions, see caption of Fig. 6.

underestimates excess density for small aggregates while it overestimates excess density for large aggregates. Generally both, $\langle \rho_p \rangle$ and $\Delta \langle \rho_f \rangle_V$, tend to increase with depth (Fig. 7a,d). In the Pacific, $CaCO_3$ as ballasting mineral becomes dissolved below the lysocline and modeled $\langle \rho_p \rangle$ decreases again in the deep ocean (Fig. 7 a,d). The mean primary particle size, $\langle d_p \rangle$, ranges between the attributed minimum and maximum primary particle size of tracers, 2 µm to 20 µm, respectively. Mean

primary particle size shows an opposing pattern to $\langle \rho_p \rangle$ (Fig. 6 a,b,g,h and 7 a,b), since regions are either dominated by small, dense coccoliths or large, less dense diatom frustules. The global pattern of $\langle d_p \rangle$, primed through export fluxes of detritus and minerals, varies only a little throughout the water column (Fig. 7 b) since we assumed invariance of primary particle size to remineralization, dissolution and other processes.

Our simulated $d_{max}$ are largest in the surface waters of the Southern Ocean and upwelling regions, where TEP-rich aggre-

gates prevail. In calcifier-dominated regions, the maximum aggregate size is small. Generally, $d_{max}$ tends to decrease with depth (Fig. 7 e).

The microstructure of marine aggregates, modeled as fractal dimension, $d_f$, shows a pronounced spatial distribution. At 100 m depth, $d_f$ ranges from about 1.7 in upwelling regions and the Southern Ocean to about 2.2 in the western equatorial Pacific and features maximum values of 2.38 in Arctic regions (Fig. 6 c). With increasing depth in the mesopelagic zone,

aggregates tend to experience a rapid compaction as $d_f$ increases (Fig. 7 c). Ongoing POM remineralization during aggregates descend shifts the aggregate composition towards mineral components which feature lower stickiness in our model. Thus the fractal dimension increases which mimics compaction of aggregates. The global pattern tends to homogenize with depth at about 1000 m, where modeled aggregates feature $d_f$ of about 2.2 (Fig. 6 i). An exception are upwelling regions, where $d_f$ remains low since detritus is slowly remineralized anaerobically in the associated oxygen minimum zones (OMZs). Below

1000 m, $d_f$ only increases slowly with depth (Fig. 7 c).




Particle properties and molecular dynamic viscosity determine the concentration-weighted mean sinking velocity of aggregates, $\langle w_s \rangle$ (Fig. 6 f,l), for particle sizes ranging from few micrometers to millimeters. At the export depth, $\langle w_s \rangle$ ranges from about $10 \, \text{m} \, \text{d}^{-1}$ in the Southern Ocean and North Pacific region to about $35 \, \text{m} \, \text{d}^{-1}$ in the western equatorial Pacific and reaches maximum values of $\sim 48 \, \text{m} \, \text{d}^{-1}$ in dust-dominated Arctic regions (Fig. 6 f,l). Upwelling-influenced surface waters tend to show

smaller $\langle w_s \rangle$ than the western equatorial Pacific. Generally, $\langle w_s \rangle$ appears to increase rapidly with depth within the mesopelagic zone (Fig. 7 f). At about $1000 \, \text{m}$, the latitudinal pattern changes and the high latitudes, in the Southern Ocean at around $45 \, ^\circ \text{S}$, exhibit the highest $\langle w_s \rangle$ of about $65 \, \text{m} \, \text{d}^{-1}$. Along the equator, outside the OMZs, $\langle w_s \rangle$ exhibits similarly high values of about $55 \, \text{m} \, \text{d}^{-1}$ (Fig. 7 f). Inside the OMZs, where remineralization is slower than in oxygenated waters, the detritus residence time is longer and leads to lower $\langle w_s \rangle$ (Fig. 6l). The subtropical regions, dominated by calcifiers, show a rather homogeneous mean

sinking velocity ($\langle w_s \rangle \approx 50 \, \text{m} \, \text{d}^{-1}$) throughout the water column apart from the first few hundred meters and near bottom regions. In turn, diatom-dominated waters feature a significantly increasing $\langle w_s \rangle$ with depth which reaches values of up to approximately $80 \, \text{m} \, \text{d}^{-1}$ (Fig. 7 f). Diatom-dominated aggregates in surface waters feature a high buoyancy through TEPs and a loose structure which diminishes with continuous remineralization during aggregates descent.

The aggregate properties entering the M$^4$AGO scheme are all directly or indirectly measurable. The comparison of simu-

lated and measured aggregate properties is, however, difficult as M$^4$AGO depicts mean values of aggregate populations that *in situ* encompass heterogeneous composition among size spectra. In addition, measurements are often limited to particular aggregate characteristics while others remain unconstrained within the same data set. We therefore compare M$^4$AGO to field and laboratory measurements which examine subsets of the simulated aggregate characteristics.

The modeled aggregate excess densities in diatom-dominated regions compare well to former field and laboratory measure-

ments, where marine aggregates showed excess densities of about 0 to $\sim 10 \, \text{kg} \, \text{m}^{-3}$ (Alldredge and Gotschalk, 1988; Ploug et al., 2008; Iversen and Ploug, 2013). In calcifier-dominated regions, aggregate excess densities are about $\sim 35 \, \text{kg} \, \text{m}^{-3}$, which is in the upper range of measured values of $2.1 \, \text{kg} \, \text{m}^{-3}$ to $41 \, \text{kg} \, \text{m}^{-3}$ (Engel et al., 2009). The increased excess density of $CaCO_3$-dominated aggregates in about $1000 \, \text{m}$ depth compares with about $100 \, \text{kg} \, \text{m}^{-3}$ to $150 \, \text{kg} \, \text{m}^{-3}$ well with observed fecal pellets egested by coccolith-consuming zooplankton (White et al., 2018). These fecal pellets also show similar mean sinking

velocities as our modeled $\langle w_s \rangle \approx 50 \, \text{m} \, \text{d}^{-1}$. The change of the excess density is linked to the increasing $d_f$ of aggregates with depth that is in qualitative agreement with present conceptual understanding (Mari et al., 2017). The increasing $d_f$ depicts the expected continuous repacking of aggregates and the zooplankton-mediated compaction in fecal pellets. The latter particularly takes place in the upper few hundred meters of the ocean and is a major pathway of coccolith transport (De La Rocha and Passow, 2007; Honjo, 1976).

The general difference in typical size between diatom-rich aggregates and $CaCO_3$ shell-enriched aggregates compares well to observations that also showed smaller $CaCO_3$-dominated aggregates (Biermann and Engel, 2010). The decreasing $d_{\max}$ with depth is in qualitative agreement with observed vertically decreasing aggregate mean diameters (De La Rocha and Passow, 2007). Decreasing maximum aggregate sizes with depth and reduced organic matter content also agree qualitatively well with experiments of Hamm (2002) and Passow and De La Rocha (2006). Both studies showed a significant decrease in aggregate size





with increasing mineral components. In their experimental setups, it remains elusive, if a certain threshold of carrying capacity of POM was reached (Passow and De La Rocha, 2006) or if the balance between adhesive forces within the aggregates and the sinking-induced shear forces (Adler, 1979; Brakalov, 1987) was shifted towards smaller aggregates. It is likely, that both effects act at the same time since natural polymers possess stronger adhesive surface properties than biogenic minerals (Eisma, 1986). In M$^4$AGO, only the compaction towards higher fractal dimension through lower internal binding forces, expressed as stickiness of the primary particles, is represented. While compaction likely enhances the internal number of binding links in aggregates, we neglect this effect versus the lower primary particle binding forces on the overall susceptibility to shear stress and thus kept $Re_{\mathrm{crit}}$ globally constant.

In summary, the resulting mean sinking velocity in M$^4$AGO is of same order of magnitude as found in observations (Alldredge and Gotschalk, 1988; White et al., 2018; Villa-Alfagame et al., 2016). We note, however, that mean sinking velocities estimated from observations potentially overestimate $\langle w_s \rangle$ as they i) are methodologically constrained to particles larger than a lower detection limit, which is typically much larger than primary particle size, and ii) depend on often uncertain size to mass relationships. We emphasize further that our modeled $\langle w_s \rangle$ embraces numerous slowly sinking aggregates of primary particle size ($w_s \approx O(1\,\mathrm{m\,d^{-1}})$) as well as rare, but large, fast sinking aggregates ($w_s \approx O(1000\,\mathrm{m\,d^{-1}})$). This range hence encompasses observations for single cells and coccoliths up to large marine snow aggregates and fecal pellets (Miklasz and Denny, 2010; Alldredge and Gotschalk, 1988; Biermann and Engel, 2010). Generally, the spatio-temporal variability of $\langle w_s \rangle$ in M$^4$AGO differs significantly from the simple underlying assumption of a linear increasing sinking velocity with depth in the standard run. M$^4$AGO resembles the strongly increasing sinking velocity found in the mesopelagic zone in observations (e. g. Villa-Alfagame et al., 2016) and a modeling approach (DeVries et al., 2014).

## 3.3  Global pattern of transfer efficiency

The transfer efficiency of POC from 100 m to depth $z$ in the ocean

$$T_{\mathrm{eff,POC},z} = \frac{\text{POC flux at depth } z}{\text{POC flux at depth } 100\,\mathrm{m}} \quad \text{for}: z > z_0 \tag{33}$$

provides an estimate on the fraction of exported POC that reaches a particular depth and is determined by sinking velocity and remineralization. In the Martin curve, Eq. (1), the slope constant, $\beta$, prescribes the transfer efficiency

$$T_{\mathrm{eff,POC},z}(\mathrm{Martin}) = \left( \frac{z}{z_0} \right)^{-\beta} \tag{34}$$

to a particular depth and leads to an almost homogeneous global transfer efficiency of $T_{\mathrm{eff,POC},z}(\mathrm{Martin}) \approx 0.12$ to about 1000 m depth in our standard run (Fig. 8a). The lower remineralization rates in sub- or even anoxic OMZs compared to oxygenated regions lead to higher transfer efficiencies, visible in the equatorial eastern Pacific Ocean, upwelling regions off Peru and Africa as well as the northern Indian Ocean. Apart from these regions, the standard run features only little variability as expected from relationship between the Martin curve slope parameter and transfer efficiency. The remaining variability is related to ocean currents and spatially variable turbulent mixing. By contrast, the transfer efficiency in M$^4$AGO exhibits a distinct global pattern and possesses higher efficiency in high latitude and upwelling regions compared to the subtropical gyres where



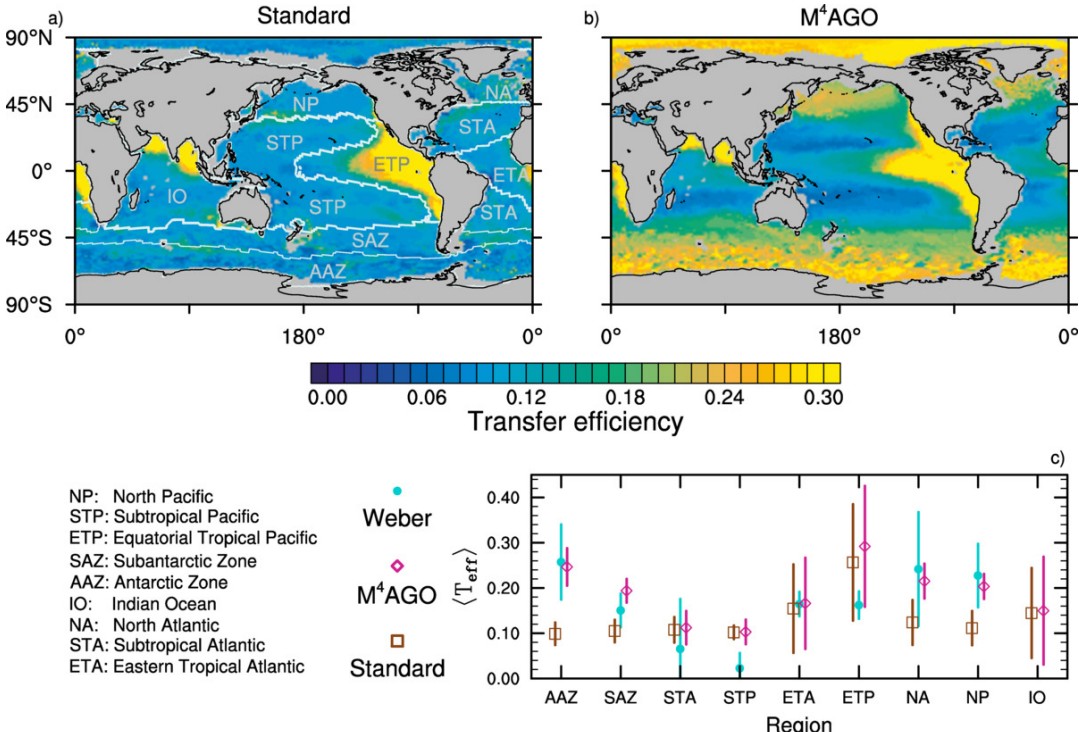

**Figure 8.** Annual mean transfer efficiency for POC in a) Standard and b) M$^4$AGO from export depth (100 m) to about 1000 m (960 m). In c) the mean transfer efficiency in regions, as defined in a), are compared to the reconstructed transfer efficiency by Weber et al. (2016). Error bars for Weber et al. represent uncertainty for the reconstruction of the regional transfer efficiency. For the model results, error bars indicate the spatial standard deviation. For the resulting effective $\beta'$ in M$^4$AGO, see Fig. B1. For a seasonal evolution of the transfer efficiency in M$^4$AGO, see Fig. C1.

low $T_{\mathrm{eff,POC},z}$ appear. Similar to the standard run, the OMZ regions feature high transfer efficiencies in M$^4$AGO. Since local Martin curves provide meaningful information on the attenuation of POC fluxes with depth, we analyzed the effective slope parameter, $\beta'$ for both, the standard and the M$^4$AGO run. We fitted the Martin curve, Eq. (1), to modeled POC fluxes below $z_0$ to estimate $\beta'$. As expected, the standard run shows little spatial variability apart from the OMZs and features a global,
5  area-weighted mean slope of $\langle\beta'\rangle_{\mathrm{A,Martin}} \approx 0.82$. In agreement with the transfer efficiency pattern, M$^4$AGO possess a strong latitudinal pattern of the effective slope $\beta'$ that varies between $\beta' \approx 0.59 - 0.67$ in high latitudes (Antarctic Zone, AAZ; North Pacific, NP; Subantarctic Zone, SAZ), $\approx 0.30$ to $0.60$ in OMZ regions and max. $\beta' \approx 1.31$ in the subtropical Pacific gyres (Fig. B1).

Recently, Weber et al. (2016) reconstructed the global transfer efficiency pattern by diagnosing particulate organic phosphate
10  fluxes. Reconstructing the transfer efficiency via inverse modeling from phosphate concentrations circumnavigates the obstacle of sparse direct observations of fluxes and allows for a more reliable constrain on POC transfer efficiency (Usbeck et al., 2003; Weber et al., 2016). The comparison of the standard and M$^4$AGO run reveal the inherent inability of the Martin approach





to capture the latitudinal variability (Fig. 8 a-c). M[4]AGO agrees qualitatively and quantitatively well with the reconstructed transfer efficiency pattern of Weber et al. (2016). The overestimation of the transfer efficiency in the equatorial tropical Pacific (ETP) by both, the standard and M[4]AGO run, is due to the models overestimation of OMZs extensions (Bopp et al., 2013). The large OMZ causes diminished remineralization and, hence, reduced attenuation of POC fluxes. In general, however, the

increased transfer efficiency associated to OMZs is in agreement with observations that suggest lower flux attenuation, and hence, small $\beta$ (Roullier et al., 2014; Löscher et al., 2016; Le Moigne et al., 2017). Lower remineralization in OMZs keeps the POC to ballasting minerals ratio higher as compared to oxygenated waters. Thus, the lower remineralization rates are potentially accompanied by lower sinking velocities which provides a positive feedback loop on the OMZ evolution. The larger the vertical extend of the OMZ becomes, the longer the retention time becomes through higher POM aggregate content

and lower sinking velocities. Eventually, OMZ evolution is balanced by oxygen supply through mixing and transport processes. The global pattern of $\beta'$ in M[4]AGO agrees well with the geographical range and pattern found by Buesseler and Boyd (2009) and suggested by Marsay et al. (2015). However, M[4]AGO shows lower maximum values of $\beta' \approx 1.31$ compared to $\beta' \approx 1.9$ suggested by Marsay et al. (2015). Globally averaged, M[4]AGO exhibits an effective slope parameter of $\langle\beta'\rangle_{\mathrm{A,M^4AGO}} \approx 0.75$ which is lower than the slope of $\beta' = 0.86$ originally published by Martin et al. (1987) based on local observations. Noticeably,

however, the trend of smaller to larger $\beta'$ values from near- to offshore Pacific US coast is in agreement with the elusive trend found by Martin et al. (1987). Overall, M[4]AGO clearly improves the representation of the POC transfer efficiency pattern compared to the standard Martin approach.

### 3.4  Contributions of $\langle w_s \rangle$ and temperature-dependent remineralization to the transfer efficiency pattern

The attenuation of POC fluxes is primarily regulated by sinking velocity and total remineralization rate of POC. Depth-

dependent, sheared lateral transport and vertical water motion can additionally affect the vertical distribution of particulate matter and thus vertical fluxes. We neglect these processes in the following, since i) the timescale of sinking from one layer to the next layer below is typically shorter than the lateral transport at grid resolutions used in our model runs, and ii) sinking velocity is faster than typical vertical motions represented by models with this grid resolution. The remineralization length scale (RLS), $z^*_{\mathrm{POC}}$ is given by the local ratio of $\langle w_s \rangle$ to remineralization ($R_{\mathrm{POC,remin}}$),

$$z^*_{\mathrm{POC}} = \frac{\langle w_s \rangle}{R_{\mathrm{POC,remin}}} \qquad (35)$$

and defines the vertical distance, in which POC would decay to half of its initial value, the POC $e$-folding depth (Cram et al., 2018). RLS can be locally calculated and enables i) to explore the local effects of remineralization and sinking velocity on the attenuation of POC fluxes and ii) to better understand the depth-integrated information provided by the transfer efficiency or Martins effective slope parameter, $\beta'$.

The RLS in surface waters and upper mesopelagic zone of subtropical and equatorial regions are by more than a factor of two shorter in the M[4]AGO run than in the standard run (Fig. 9 a). This higher turnover causes the lower $p$-ratio in M[4]AGO compared to the standard run in these regions (Fig 4 b). In the mesopelagic zone, the RLSs in M[4]AGO is similar or pronounced longer and decreases again in deeper regions compared to the standard run (Fig. 9 a). The longer RLS in the mesopelagic zone



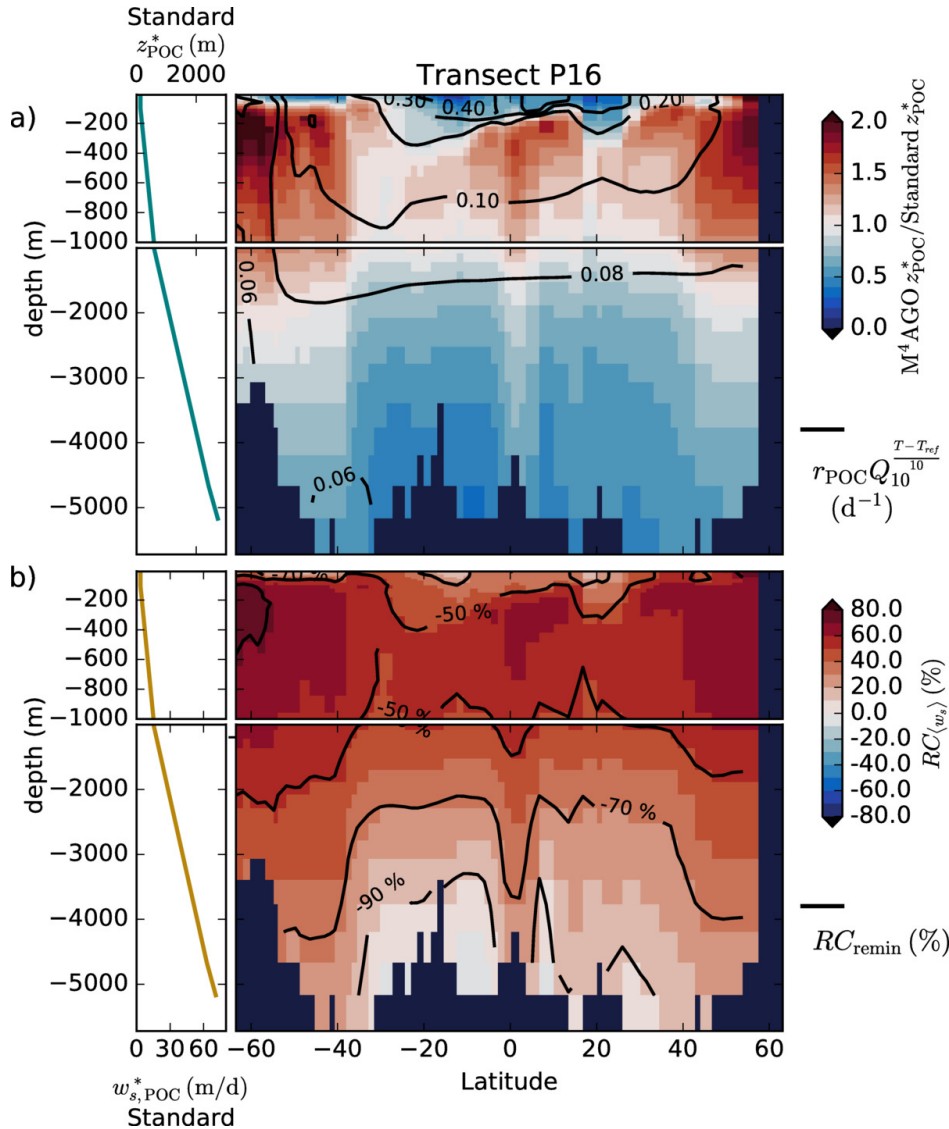

**Figure 9.** a) Remineralization length scale ratio of $M^4AGO$ to the standard model version at the World Ocean Atlas transect P16. We here focus on showcasing the temperature effect on remineralization and calculated $z_{POC}^*$ without oxygen limitation of remineralization which cancels out for equal $O_2$ concentrations. Values smaller than one imply stronger POC flux attenuation in $M^4AGO$ than in the standard run. For reference, the RLS in the standard run is given at top left. The standard RLS is increasing due to increasing sinking velocity with depth (shown bottom left). Contour lines provide the temperature-dependent remineralization rates in $M^4AGO$. In the standard run, the aerobic rate is globally constant ($0.026\,\mathrm{d}^{-1}$). b) Relative contributions of sinking and remineralization to difference between standard and $M^4AGO$ run. Contour lines provide the relative contribution of remineralization.





of the high latitude ocean are the reason for the higher transfer efficiency of M$^4$AGO compared to the standard run. In order to analyze which of the two processes, sinking or remineralization, is of primary importance for the change in the RLS and thus the transfer efficiency, we define their relative contributions to the difference between M$^4$AGO and the standard run as

$$RC_{P_i,\mathbf{x}}[\%] = \frac{\partial_{P_i} z^*_{\mathrm{POC}} \cdot \Delta P_{i,\mathbf{x}}}{\sum_i |\partial_{P_i} z^*_{\mathrm{POC}} \cdot \Delta P_{i,\mathbf{x}}|} \cdot 100 \tag{36}$$

where the processes, $P_i = \{\langle w_s \rangle, R_{\mathrm{POC,remin}}\}$, refer to Eq. (35), $\partial_{P_i}$ is the partial derivative of $z^*_{\mathrm{POC}}$ with respect to the process $P_i$ (applying the standard run rates), and $\Delta P_{i,\mathbf{x}}$ is the difference between the value in the M$^4$AGO run and the standard run at spatial point $\mathbf{x}$. M$^4$AGO possesses generally higher remineralization rates than the standard run which would increase the flux attenuation compared to the standard run. The temperature-dependent remineralization in M$^4$AGO shows lower rates in the cold waters of the high latitudes than in the equatorial and subtropical regions (Fig. 9 a). Within M$^4$AGO, this pattern enhances
the RLSs, and thus transfer efficiency, in the high latitudes compared to the equatorial regions, which is in agreement with Marsay et al. (2015); Cram et al. (2018). Compared to the standard run, higher $\langle w_s \rangle$ overcompensates the effect of intensified remineralization below the thermocline and leads to longer RLS (Fig. 9 b) in the mesopelagic zone, which is particularly true in diatom-dominated regions. In CaCO$_3$-dominated subtropical gyres, $\langle w_s \rangle$ falls below the sinking velocity of the standard model in regions deeper than 3000 m and thus contributes further to the stronger flux attenuation compared to the standard run. The
higher $\langle w_s \rangle$ in M$^4$AGO turns out to dominate over remineralization and increases the RLS, and hence the transfer efficiency, in the mesopelagic zone of the high latitudes. In summary, temperature-dependence of remineralization in M$^4$AGO induce a latitudinal pattern of longer RLSs, and thus higher transfer efficiency, in high latitudes that is further amplified by high sinking velocities of diatom-dominated aggregates in the mesopelagic zone.

### 3.5   Impact of mineral size and ballasting effect on sinking velocity

The analysis of RLS changes compared to the standard run emphasizes the role of sinking velocity for longer RLS in the mesopelagic zone in high latitudes and thus for the enhanced transfer efficiency. We therefore aim at a better understanding of the underlying factors that control the mean sinking velocity. We suggest in the following that the size of primary particles might be as important as density of the ballasting material for defining the sinking velocity of aggregates.

M$^4$AGO allows for assessing the contributions of aggregate properties and molecular viscosity on $\langle w_s \rangle$. We define the
relative contributions of the modelled particle properties and the molecular viscosity that control $\langle w_s \rangle$ at particular depth $z$ based on a first order approach

$$R_{X_i,z}[\%] = \frac{\partial_{X_i} \langle w_s(\bar{X}_{i,z}) \rangle \cdot \Delta X_{i,z}}{\sum_i |\partial_{X_i} \langle w_s(\bar{X}_{i,z}) \rangle \cdot \Delta X_{i,z}|} \cdot 100 \tag{37}$$

where $X_i = \{\langle \rho_p \rangle, \langle d_p \rangle, \mu, d_{\max}, b, d_f\}$, $\Delta X_{i,z} = X_{i,z} - \bar{X}_{i,z}$, and $\sum R_{X_i,z} \neq 100$, but $\sum |R_{X_i,z}| = 100$, and $\bar{X}_{i,z}$ as the global mean of the contributing property at depth $z$. The relative contributions provide an information on the main driv-
ing factors, expressed as percentage, for the local $\langle w_s \rangle$ as compared to global average aggregates. For example, what is the percentage-wise contribution of denser primary particles than global average to the local sinking velocity. By neglecting the





higher order terms, we provide only qualitative insights into the role of the different aggregate properties and molecular viscosity on $\langle w_s \rangle$.

CaCO$_3$ acts as a strong positive, $\langle w_s \rangle$-increasing factor for sinking velocities in the equatorial and subtropical regions due to its high primary particle density (Fig. 10 a). As an exogenous factor, the low molecular viscosity in warm regions contributes positively to sinking velocity (Fig. 10 d). Additionally, in our model, $d_f$ increases particle excess density and thus $\langle w_s \rangle$ in CaCO$_3$-rich areas (Fig. 10 c). In the Southern ocean, North Pacific, North Atlantic, and upwelling regions, particularly the large $\langle d_p \rangle$ in diatom-dominated regions enhances $\langle w_s \rangle$ (Fig. 10 b). According to our model, $d_{\max}$ of aggregates and the number distribution slope, $b$, contribute positively to $\langle w_s \rangle$ in the high latitudes, except for the Arctic ocean, where mineral material strongly affects the aggregate properties. The pattern of highly variable endogenous and exogenous controls on $\langle w_s \rangle$ emerges particularly in the upper ocean, while in deeper regions of about 1000 m depth, the aggregate properties become more and more homogeneous, except for $\langle d_p \rangle$ and $\langle \rho_p \rangle$ and in OMZs (Fig. 10, lower two panels). With homogenization of most aggregate properties with depth, the contrasting relative contributions of dense mean primary particles and large mean primary particles to $\langle w_s \rangle$ become even more pronounced. Denser, small CaCO$_3$ particles contribute to $\langle w_s \rangle$ in the subtropical, oligotrophic regions of the oceans, while comparably less dense, but larger opal frustules lead to high $\langle w_s \rangle$ in nutrient-rich upwelling and high latitude regions. Even though our formulation for $\langle w_s \rangle$ is highly non-linear, we expect the qualitative pattern of the relative contributions to $\langle w_s \rangle$, particularly through $\langle d_p \rangle$ and $\langle \rho_p \rangle$, to be coherent. In sum, we therefore emphasize the potential role of primary particle size, in particular that of diatom frustules, for determining $\langle w_s \rangle$, and thus POC fluxes.

The effects of microstructure on *in situ* $\langle w_s \rangle$ are not well studied and $d_f$ of aggregates is weakly constrained. It therefore warrants further investigation, especially if a spatially variable $d_f$ effect occurs on *in situ* $\langle w_s \rangle$ in the upper ocean. Vertically varying $d_f$ has been suggested (Mari et al., 2017) and given the microstructure general importance for sinking velocity, a deeper understanding of the factors and processes influencing $d_f$ is hence highly desirable.

Microstructure $d_f$ directly prescribes the aggregate number distribution slope, $b$, in M$^4$AGO (see Fig. A1 c, f for a map of $b$). Observations show a higher variability of $b \approx 2$ to $5$ (Guidi et al., 2009) than in M$^4$AGO, where $b$ varies between $\approx 3.19 - 3.76$ and thus likely underestimates the spatial variability of the relative contribution to $\langle w_s \rangle$. At present, M$^4$AGO is limited to the steady state size distribution that represents the characteristic processes of aggregation and fragmentation in the system. An explicit modeling of the dynamics of the aggregate size spectrum would be required to cover the variability of measured $b$ which would further enhance the variability of $\langle w_s \rangle$. We briefly discuss this current model limitation in Sec. 3.10.

Previous studies support our finding that CaCO$_3$ act as a strong ballasting agent (e. g. Francois et al., 2002; Balch et al., 2010; Cram et al., 2018). By contrast, opal density of the hollow silicate structures have less effect on $\langle \rho_p \rangle$ (see Fig. 7 a). Instead, silicate frustule size of diatoms significantly affects $\langle w_s \rangle$ (see Fig. 10 b, h and 7 b). This contrasts the assumption of opal acting solely as ballasting material (Cram et al., 2018) and is congruent with Francois et al. (2002) who noticed that factors other than particle density likely play a role. Deciphering the size effect of primary particles on *in situ* $\langle w_s \rangle$ and fluxes might be particularly challenging in regions with a diverse size structure of phytoplankton community. Oligotrophic regions typically harbor a narrower size distribution of phytoplankton (see e. g. size ranges and standing stocks in Kostadinov et al., 2016), which may produce more homogeneous aggregates and thus more predictable POC fluxes as found by Guidi et al. (2016) who



**Figure 10.** Qualitative first order relative contributions of marine aggregate properties to the change of local mean sinking velocity compared to a global mean aggregate spectrum at depth $z$. a)-f): 100 m, g)-l) 960 m.





correlated POC fluxes to the oligotrophic phytoplankton community. In addition, interannual variability of the dominant size of primary producers have been suggested to drive the interannual change in export fluxes (Boyd and Newton, 1995). In contrast to oligotrophic phytoplankton communities, diatom-dominated communities feature a higher size diversity (Tréguer et al., 2018) and different morphologies, which both affect the sinking velocity of aggregates (Laurenceau-Cornec et al., 2015; Bach

et al., 2016). Phytoplankton community size structure and morphology thus introduces higher variability to sinking velocity which complicates attribution of cell size and morphology effects on POC fluxes. The higher size and morphology variability in diatom-dominated phytoplankton communities, together with variable cellular silicate to carbon ratios (Brzezinski, 1985), likely explains the poor correlations of opal to POC fluxes (e.g. found by Francois et al., 2002). Similarly, the correlation between POC fluxes and, by number and size, more variable foraminiferal $CaCO_3$ is weaker than for coccolith fluxes (De La

Rocha and Passow, 2007). We therefore suggest to factor in, or even focus on, frustule sizes and morphology as explanatory variables for POC fluxes when carrying out mineral ballasting and rain ratio studies in diatom-dominated regions.

### 3.6   Regional fluxes & rain ratios

Biogenic minerals, dust particles, and detritus are tied together in $M^4AGO$ and define the composition, microstructure and thus sinking velocity of aggregates. In turn, the tracers are independently remineralized or dissolved. Both processes affect $\langle w_s \rangle$ and

thus the e-folding depth of tracers. For example, the remineralization of POC increases $\langle w_s \rangle$ of diatom-dominated aggregates and thus increases the dissolution length scale of opal, which we refer to as 'opal RLS'. The combined sinking thus i) affects the RLS of POC and opal (and $CaCO_3$ when dissolution takes place) and ii) couples the timing of mineral and POC fluxes at depth.

The Martin curve concept can be applied to represent the effective attenuation of opal fluxes. The effective Martin curve

slope for opal, $\beta'_{opal}$, exhibits similar regional variability in the $M^4AGO$ run as in the standard run (Fig. 11). However, the Equatorial Tropical Pacific (ETP) region shows higher spatial variability. The low POC remineralization in OMZs leads to small $\langle w_s \rangle$ of aggregates and decreases the opal RLS and thus increases $\beta'_{opal}$. Similar to the POC remineralization length scales, opal fluxes exhibit shorter opal RLS in the surface waters, while they exceed the standard RLS in the mesopelagic zone and below (not shown). Particularly in the deep waters of the high latitudes, the opal RLSs are longer than in the standard run.

This is due to the aggregates higher sinking velocity than the $25\,\mathrm{m\,d^{-1}}$ opal sinking speed in the standard run and partially to the reformulation of temperature-dependent opal dissolution. In the subtropical regions, opal is remineralized faster below $2000\,\mathrm{m}$ and fluxes are generally small.

In sum, $M^4AGO$ estimates globally $\sim 1.03\,\mathrm{Gt}$ Si per year reaches the seafloor which is in good agreement with present estimates of about $1\,\mathrm{Gt}$ Si per year (Tréguer, 2002). Generally, the modelled $\beta'_{opal}$ is well within the range of observations

with estimates of $\beta'_{opal} = 0.22 \pm 0.53$ (Boyd et al., 2017). In $M^4AGO$, dissolution of opal has the implication that ballasting mineral ratios between opal and $CaCO_3$ shift towards higher importance of $CaCO_3$ with depth. Since the dissolution of opal is temperature-dependent, the preservation efficiency for opal to $CaCO_3$ ratios is thus lowest in regions with strong vertical temperature gradients.





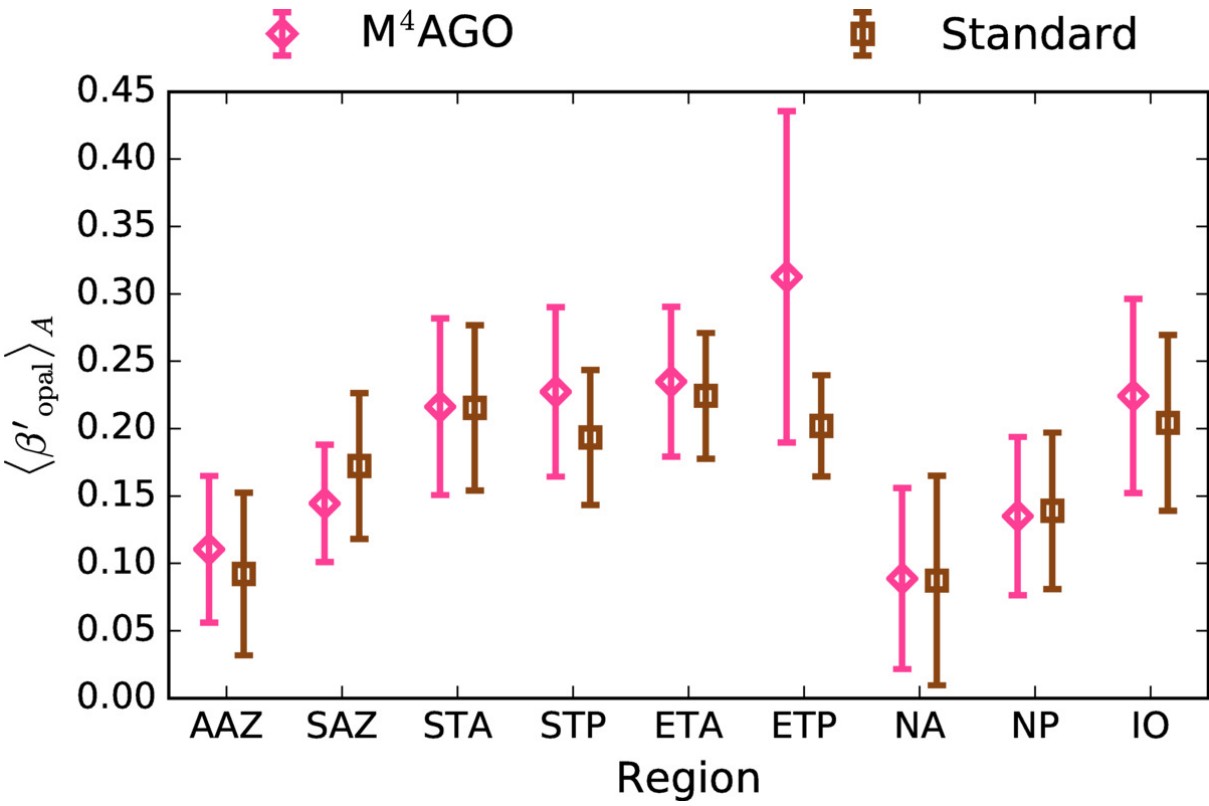

**Figure 11.** Area-weighted mean effective power function slopes for opal fluxes $\langle \beta'_{\mathrm{opal}} \rangle_{\mathrm{A}}$ in the regions defined in Fig. 8. Calculated for opal fluxes below 100 m. Error bars show regional standard deviation.

The direct coupling of POC and biogenic mineral fluxes through marine aggregates potentially enhances the models ability to represent rain ratios in space and time. We therefore aggregated the sediment trap data set of Mouw et al. (2016a, b) with 15792 individual POC flux measurements at 673 unique locations to a monthly climatology of POC, particulate inorganic carbon (PIC) and silicate fluxes. We accounted for the time spans of sediment trap deployment ranging from hours to years by

5    time-weighting with a minimum weight of one day per month in cases of few hours measurement, and for each covered month the full monthly weight in cases of a yearly measurement. We compared the climatologies to modeled monthly mean flux ratios at the stations at their respective depths. The M$^4$AGO run represents the POC/PIC flux ratio equally well as the standard run (not shown). For POC/Si fluxes, the scatter around the 1:1 line is reduced in M$^4$AGO compared to the standard run (Fig. 12).

However, M$^4$AGO introduces a bias in regions deeper than $\sim$4000 m towards smaller POC/Si ratios. The RLSs in the deep

10   ocean are hence too short in M$^4$AGO. While the scatter of the POC/Si ratio reduces in M$^4$AGO, the overall variability of the flux ratios becomes compressed compared to the measured variability. This compression might be caused by several factors. First, we assume a tight connection of aggregate components and ignore heterogeneous composition among a local particle size spectra which would potentially cause different sinking velocities among the different components. Second, HAMOCC

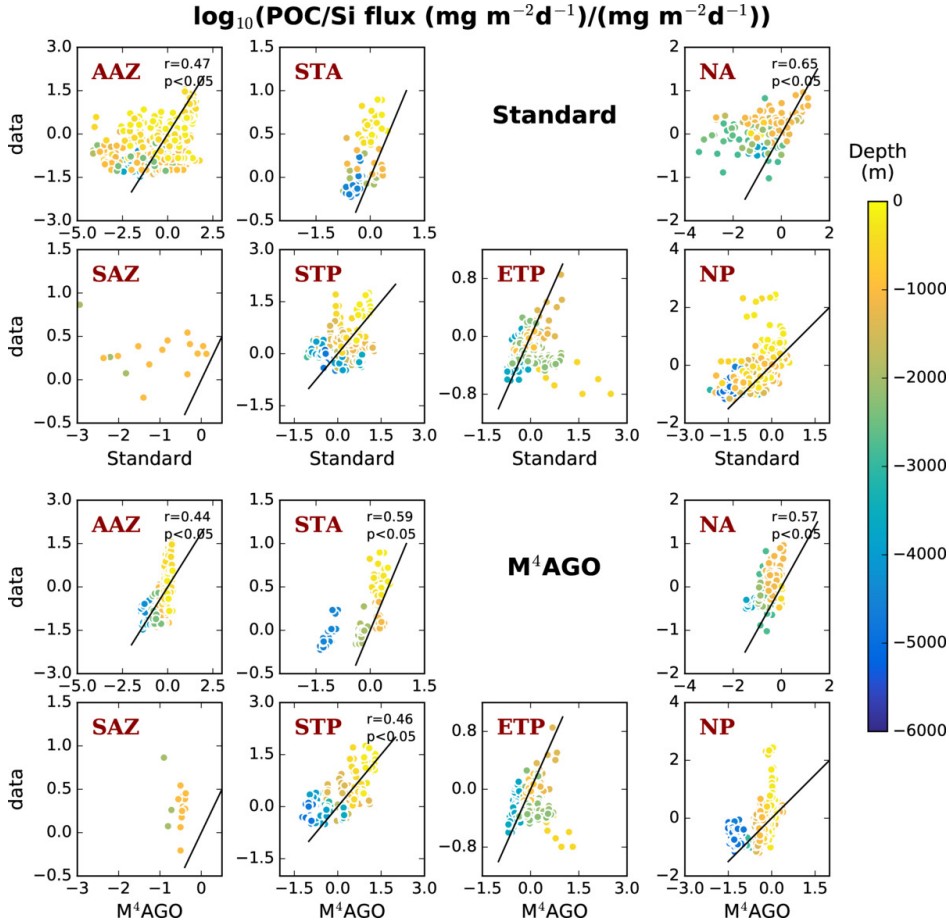

**Figure 12.** POC/Si rain ratios in the standard run and M⁴AGO compared to the Mouw et al. (2016a, b) data set. The black line denotes the 1:1 line. Notice that the axes are in log and have different limits among regions, but are comparable between runs. No data are available for opal fluxes in the equatorial tropical Atlantic (ETA) where opal fluxes are small (refer to Fig. 5 a, b). Correlation coefficient, r, and significance value, p, are given, if r> 0.4.

ignores changing diatom silification caused by temperature (see Ragueneau et al., 2006, and references therein) and by seasonal changes in nutrient availability (Assmy et al., 2013). Generally, deficits exist in both the model runs and sediment trap data. The lack of resolving small scale variability through eddies and their role in shaping the phytoplankton community, as well as general timing, internal variability and spatial current shifts, limits the global models ability to represent local features.

5  Measurement limitations of sediment traps have the potential to additionally increase the mismatch (see e. g. Usbeck et al., 2003, and references therein).





### 3.7 Evaluation of biogeochemical tracer distributions

We evaluate the simulated spatial distribution of nutrients, oxygen, and alkalinity by comparison to gridded observation climatologies. Silicate, phosphate, nitrate, and oxygen are compared to the World Ocean Atlas (WOA; Boyer et al., 2013; Garcia et al., 2014b). Alkalinity, $A_T$, is compared to the Global Ocean Data Analysis Project (GLODAPv2) climatology (Lauvset et al., 2016). We present the results in Taylor diagrams (Taylor, 2001), which aggregate correlation, root-mean-square-deviation (RMSD), and standard deviation of simulated and observed variables into distances between model results and the reference observations (Fig. 13 a-d). Here, we use spatial grid-cell-wise statistics derived from temporally averaged (100 year mean) fields for which we interpolated the observational data to the model grid. To be able to show all parameters in one plot we derive normalized standard deviations and RMSD. We evaluate the Atlantic and the Pacific Ocean separately to account for their different hydrographical features. For example, the Atlantic ocean is characterized by ventilation through deep water formation in the high latitudes, a feature that does not exist in the Pacific. Furthermore, we select four depth levels: surface waters (6 m depth), the two depths that determine the transfer efficiency, 100 m and approximately 1000 m, and an intermediate depth in the mesopelagic zone, 362 m.

Generally, the two model runs equally well represent nutrients, silicate and oxygen distributions at these depth (Fig.13 a-d), i.e. differences to observations are larger than among the two models. This is expected, as tracer distributions in the model are predominantly determined by the flow field and both simulations use identical physical model conditions. The Taylor diagram captures matches of spatial pattern such as location of fronts, extension of gyres and the location of water masses in an aggregated form. The ocean model represents such features realistically (Jungclaus et al., 2013), but we cannot expect a perfect match of circulation pattern with *in situ* conditions for multiple reasons. The climatological, simplistic atmospheric forcing damps in particular the interannual variability of ocean currents and can contribute to spatial shifts of e. g. frontal regions with respect to the observed climatology. In addition, the coarse horizontal resolution of the model and atmospheric forcing affects features such as upwelling strength (Milinski et al., 2016) and location. In turn, the spatio-temporal interpolation of data, necessary through scarcity of observations, limits the resolution of nutrient variability and introduces uncertainty, particularly in deeper regions, where the density of observations is lower than in surface waters. As a consequence, a mismatch between modelled biogeochemical tracers and the climatological mean of observations is expected.

Tracers undergoing a less complex biogeochemical cycling, such as phosphate or silicate, reflect the quality of the flow field more directly than tracers such as nitrate, oxygen, or alkalinity (see also England and Maier-Reimer, 2001). The latter ones are therefore generally more prone to biogeochemical models reformulation and larger differences can be expected between M[4]AGO and the standard run. In addition, M[4]AGO links the cycles of phosphate and silicate closer to nitrate and oxygen through common sinking of particulate matter. Phosphate and silicate therefore experience an additional biogeochemical cycles imprint in M[4]AGO compared to the standard run. In surface waters of the Pacific, this led e. g. to a slight improvement of the phosphate correlation with WOA observations, while it increased the normalized standard deviation compared to the standard run (Fig. 13 a). The improvement is related to the better representation of surface phosphate concentrations off Peru and North America coast (see Fig. 13 e-g).


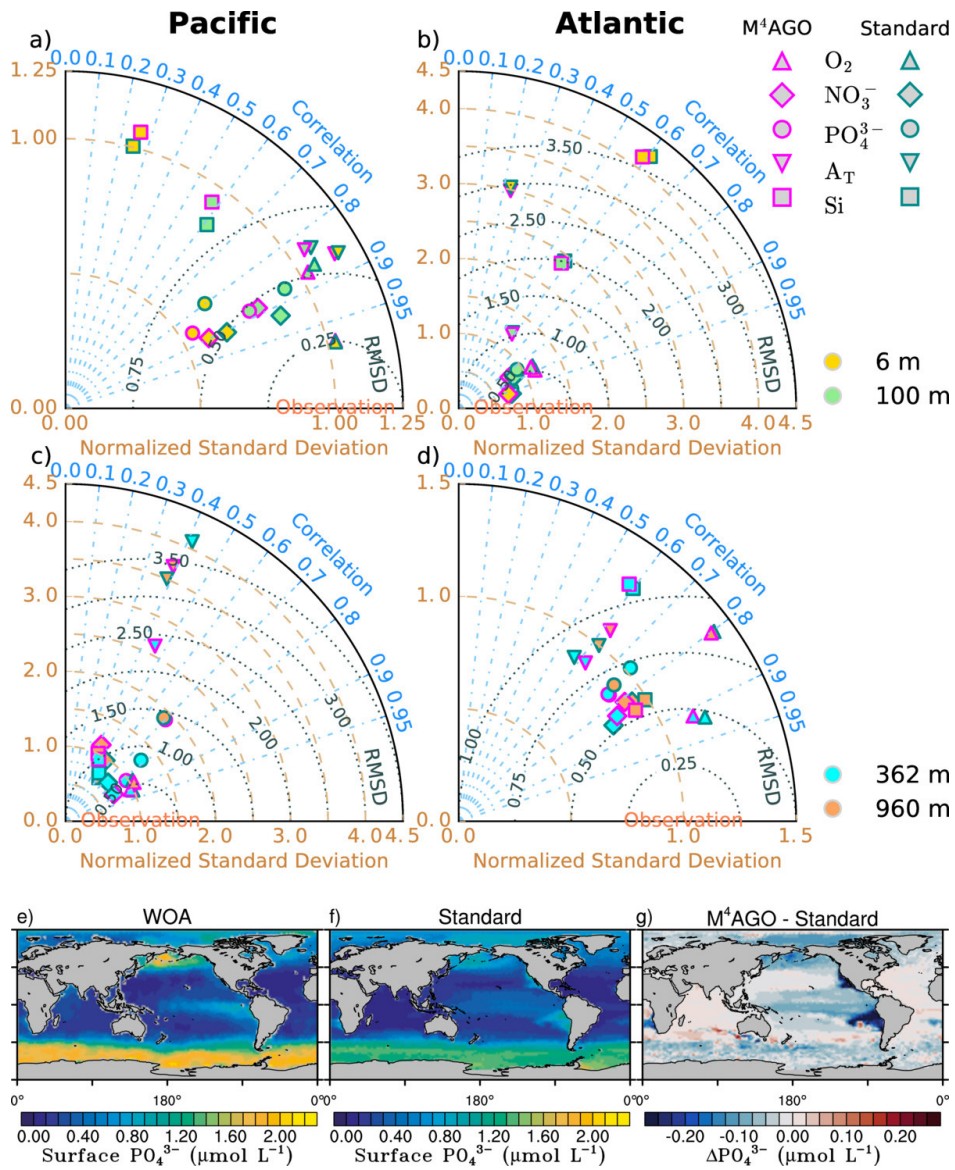

**Figure 13.** a)-d) Taylor diagram for tracers in comparison to the World Ocean Atlas data (Boyer et al., 2013; Garcia et al., 2014b) and, in the case of total alkalinity, $A_T$, to GLODAPv2 data (Lauvset et al., 2016). a) and c): Pacific, b) and d) Atlantic for upper panel 6 m and 100 m. Lower panel 362 m and 960 m. e) surface phosphate concentration in WOA f) in the standard run and g) difference plot between $M^4$AGO and the standard run.

In both simulations, biogeochemical tracer distributions in the Pacific are strongly influenced by the OMZ in the eastern boundary upwelling region. An exception is oxygen in surface waters which is primarily determined by gas exchange processes. As a common feature of all state-of-the-art global ocean biogeochemistry models, OMZs are too large and result from





nutrient trapping through a sluggish circulation and associated insufficient supply of oxygen (Dietze and Loeptien, 2013). The sluggish circulation and mixing is likely associated with underrepresented equatorial currents, particularly the equatorial intermediate current system and equatorial deep jets (Brandt et al., 2012; Shigemitsu et al., 2017). The OMZ shape in the eastern Pacific ocean, though, changes between the runs. In M$^4$AGO, the water column above 800 m is more and below less
oxygenated than in the standard run. The change in OMZ shape imprints on $A_T$, as anaerobic and aerobic remineraliziation processes change alkalinity in a different manner, i.e. denitrification and sulfate reduction increase alkalinity, whereas aerob remineralization decreases alkalinity. While $A_T$ and silicate are generally worst compared to observations, M$^4$AGO improves the representation of $A_T$ in the Pacific at 360 m compared to the standard run. This is well visible through the higher correlation ($\sim 0.45$ compared to 0.4) and lower RMSD of $\sim 2.3$ compared to $\sim 3.75$ in the standard run (Fig. 13 c, 362 m depth).
The increased RMSD of silicate in surface waters and 100 m in M$^4$AGO are associated to the eastern equatorial and upwelling regions in the Pacific. The lower remineralization in OMZs not only decreases sinking velocity of detritus, but also increases the retention time of the tightly coupled opal in M$^4$AGO and thus enhances the opal dissolution and silicate concentration. In general, the tighter coupling of silicate to nutrients through common sinking in aggregates leads to a higher sensitivity of silicate to ocean circulation and temperature deficiencies in M$^4$AGO than in the standard run. The high RMSD and low correlation
for silicate compared to observations, particularly in the euphotic zone, can have additional implications for M$^4$AGO, since the silicate distribution in the euphotic zone directly affects opal production and thus the sinking velocity of aggregates, transfer efficiency and nutrient distributions. The tighter coupling of silicate to other biogeochemical cycles in M$^4$AGO therefore warrants further future investigation.

### 3.8   Regional CO$_2$ uptake

In a 100-year climatological steady state, both model runs show the southern hemisphere ocean acting as a net source of CO$_2$ to the atmosphere while the northern hemisphere acts as a net sink (Fig. 14 a). Consequently, a net oceanic CO$_2$ transport across the equator from the northern to the southern hemisphere exists. In the simulation with M$^4$AGO, a stronger CO$_2$ uptake compared to the standard run occurs in the region between 60 °S and 45 °S which coincides with deeper transfer of POM reflected by an increased transfer efficiency (Fig. 8). In both model runs, the subtropical regions outgas CO$_2$ to the atmosphere.
As visible from the decreasing difference between the cumulative zonal CO$_2$ fluxes, M$^4$AGO exhibits a stronger outgassing in the subtropical region, particularly in the northern hemisphere (Fig.14 a). Even though a small residual of zonally integrated fluxes of about $0.02\,\mathrm{Gt\,C\,yr^{-1}}$ between the two model runs remains, the small residual CO$_2$ flux imbalance of $0.07\,\mathrm{Gt\,C\,yr^{-1}}$ and $0.05\,\mathrm{Gt\,C\,yr^{-1}}$ in M$^4$AGO and the standard run, respectively, indicate well spun-up model runs.

   To study the effect of the changed transfer efficiency in M$^4$GO on CO$_2$ uptake without indirect climate effects, we perform
model runs in which we linearly increase the atmospheric CO$_2$ concentration with a yearly increment from $\sim$285 ppm to 400 ppm within 150 years. The transient forcing does not feed back onto the physical climate state. We therefore emphasize that no climate feedback on e. g. ocean temperature, stratification, and thus primary production and remineralization occurs in these model runs.



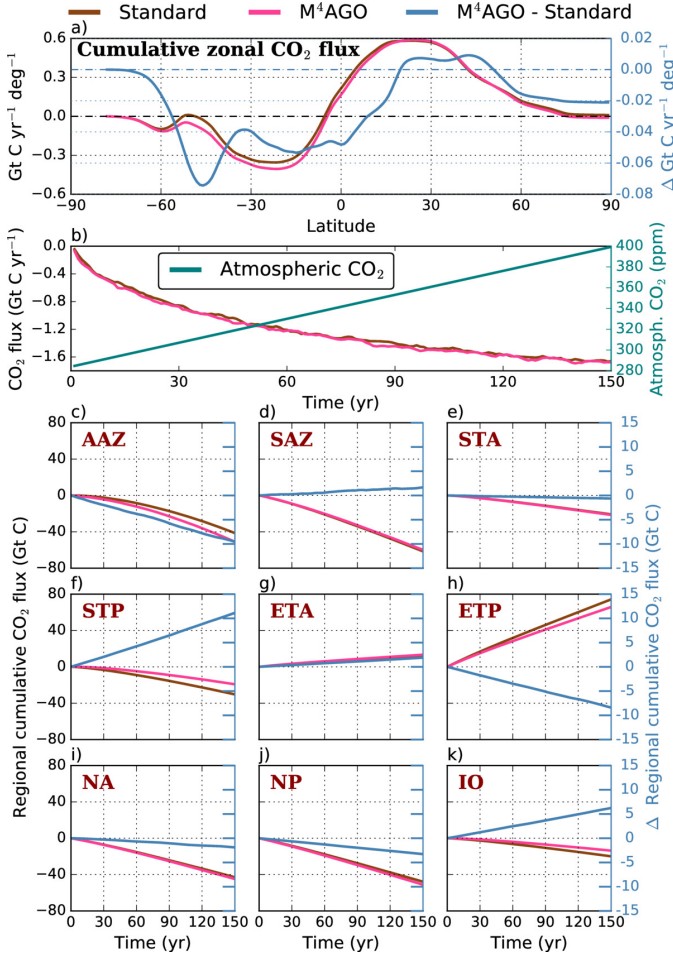

**Figure 14.** a) Climatological cumulative zonal $CO_2$ flux in the standard and the $M^4$AGO run. Generally, negative fluxes represent net-$CO_2$ uptake by the ocean. Note the second y-axis for the difference between the two runs (also in c) - k)). b) Global annual net $CO_2$ flux under linearly increasing atmospheric $CO_2$ concentration. c) - k) Regional cumulative $CO_2$ fluxes under increasing atmospheric $CO_2$ (regions as defined in Fig. 8).

During the atmospheric $CO_2$ increase, both model runs show similar global annual oceanic $CO_2$ uptake (Fig. 14 b). About 181-185 Gt C are taken up by the global ocean throughout the 150 year period. The discrepancy of $\approx 4$ Gt C between the two runs arises to a large extent from the residual imbalance of $0.02$ Gt C yr$^{-1}$ uptake, cumulated over 150 years.

The general trends in the regional cumulative fluxes remain the same between the two model runs (Fig. 14 c-k) and most regions act as net sinks of $CO_2$. Only the upwelling regions of the equatorial tropical Atlantic (ETA) and Pacific (ETP) are net sources of $CO_2$ to the atmosphere at the end of the 150 yr period. The magnitude of the regional cumulative fluxes, however, varies among the two model runs.



Since we can rule out temperature-driven effects on the differences in $CO_2$ uptake when comparing the two climatologically forced runs, a different state and/or changes in $A_T$ and dissolved inorganic carbon (DIC) concentration determine the partial pressure of $CO_2$, $pCO_2$, and thus $CO_2$ fluxes. In our runs, major changes in $A_T$ and DIC can locally occur due to a change in, first, the $CaCO_3$ to detritus rain ratio as a result of varying $CaCO_3$ and POM production and remineralization, or, second, e. g.

an increase of the POC transfer efficiency, which reflects a transfer of carbon to deeper waters where $CO_2$ is withdrawn from immediate exchange with the atmosphere. In addition, such different signals can be transported by the flow field and remote effects can appear. Quantitatively, differences of regional cumulative $CO_2$ fluxes larger than $5\,\mathrm{Gt\,C}$ appear in the Antarctic Zone (AAZ), the Subtropical Pacific (STP), the Equatorial Tropical Pacific (ETP), and the Indic Ocean (IO) (Fig. 14 c-k). In the upwelling and high latitude regions, the $pCO_2$ is lower in $M^4AGO$ than in the standard run, which translates to higher

oceanic uptake (in AAZ, NP) or less outgassing (in ETP). Qualitatively, this coincides well with the higher transfer efficiencies in these regions (cmp. to Fig. 8). In the STP region, where high $CaCO_3$ to detritus rain ratios occur (refer to Fig. 5), $A_T$ is about 0.2 % lower in $M^4AGO$ than in the standard run, which enhances $pCO_2$ by about 2 % and thus decreases the $CO_2$ uptake.

In summary, under linearly increasing $CO_2$ in a non-interactive mode, the global $CO_2$ fluxes remain similar for the standard and the $M^4AGO$ run. However, regional $CO_2$ fluxes change, which is potentially linked to the changing pattern of transfer

efficiency and the $CaCO_3$ to detritus rain ratio. A detailed study on the feedbacks between the transfer efficiency, represented by $M^4AGO$, and a transient climate on oceanic $CO_2$ uptake is out of the scope of this manuscript and will be part of future investigations.

## 3.9 Sensitivity analysis

With $M^4AGO$, the processes of temperature-dependent remineralization and a number of new model parameters to represent

variable sinking velocity of marine aggregates were introduced to HAMOCC. We carried out three sensitivity experiments to provide insights into HAMOCCs response to the uncertainty of selected parameters (see below for the criteria). As target variables, we chose i) the transfer efficiency, ii) phosphate as essential nutrient for primary production, and iii) silicate as nutrient and circulation-reflecting agent.

We suggested the size of primary particles, particularly that of opal frustules, as an important factor for regulating $\langle w_s \rangle$

and thus the transfer efficiency. We thus study the effect of primary particle size in the sensitivity experiment $S(d_{p,\mathrm{frustule}})$ exemplarily for diatom frustules because: i) diatom size is highly variable, ii) HAMOCC does not explicitly represent algae size classes and iii) algae body size is expected to decrease with rising water temperatures (Daufresne et al., 2009). We decrease their size by 50 % from $d_{p,\mathrm{frustule}} = 20\,\mu m$ to $d_{p,\mathrm{frustule}} = 10\,\mu m$. In the second sensitivity experiment, $S(d_f)$, the varying fractal dimension, ranging between 1.6 and 2.4 in the $M^4AGO$ run, is set to a constant, $d_f = 2$, to eliminate its variable effect

on $b$ and $\langle w_s \rangle$. In the third experiment, $S(K_{P,\mathrm{diaz}})$, we showcase the sensitivity of phosphate concentration in the subtropical gyres to diazotrophs standing stock which we noticed during the process of tuning the $M^4AGO$ run. Here, we increase the half-saturation constant for phosphate uptake by diazotrophs from $0.05\,\mu mol\,L^{-1}$ to $0.10\,\mu mol\,L^{-1}$. All sensitivity runs are in quasi-steady state with shown surface properties.



**Figure 15.** Sensitivity of a)-c) surface phosphate, d)-f) silicate concentrations, and g) POC transfer efficiency to $d_{p,\text{frustule}}$ ($S(d_{p,\text{frustule}})$, modified from 20 µm to 10 µm), the half-saturation constant for phosphate uptake by diazotrophs ($S(K_{P,\text{diaz}})$, modified from 0.05 µmol L$^{-1}$ to 0.1 µmol L$^{-1}$), and $d_f$ ($S(d_f)$, from variable to constant $d_f = 2$). Error bars in g) represent regional standard deviation and for Weber et al. the uncertainty for the reconstruction of the regional transfer efficiency.

Decreasing the opal frustule size by 50 % compared to the M$^4$AGO run reduces $\langle w_s \rangle$ and thus the RLS in diatom-dominated regions. As a consequence, the transfer efficiency in silicifier-dominated regions is low (Fig. 15 g). Accordingly, opal dissolves closer to surface waters and the silicate concentration increases with respect to the M$^4$AGO run in silicifier-dominated regions (Fig. 15 d). In the subtropical gyres, where silicate is diminished, it further decreases in $S(d_{p,\text{frustule}})$. The higher remineral-





ization in the euphotic zone in upwelling regions increases the phosphate concentrations downstream in the subtropical gyres (Fig. 15 a). In sum, the sensitivity to $d_{p,\text{frustule}}$ underpins the potential importance of primary particle size for $\langle w_s \rangle$ and thus for biogeochemical cycling.

Generally, eco-physiological responses of primary producers are typically neglected in ESM-type models such as HAMOCC. Under the premise that the size of primary particles is of importance to represent $\langle w_s \rangle$ as depicted by M$^4$AGO, changes of the phytoplankton size structure with ongoing ocean warming could affect RLSs, transfer efficieny, and thus the biological carbon pump. Indeed, body size decreases with increasing water temperature (Daufresne et al., 2009) which is suggested to shift eco-physiological regions polewards by rates of about 22 to 36 km per decade (Lefort et al., 2015, and references therein). $S(d_{p,\text{frustule}})$ shows that decreasing primary particle size reduces the transfer efficiency and thus likely weakens the biological carbon pump. Yet, the net-effect of such eco-physiological responses and adaptability of primary producers on transfer efficiency and $CO_2$ uptake under ongoing climate change remains elusive and thus demands for further future investigation.

In the sensitivity study $S(d_f)$, $d_f$ in surface waters is increased in diatom-dominated and reduced in calcifier-dominated waters compared to the M$^4$AGO run (cmp. to Fig 6 c). In the M$^4$AGO run, aggregates experience rapid compaction within the depth of about 150 m to 250 m in diatom-dominated regions, $d_f$ increases rapidly (see Fig. 7 c), and hence, $\langle w_s \rangle$. By contrast, $\langle w_s \rangle$ is only enhanced in the productive surface waters of the SAZ, upwelling regions and the associated OMZs due to the larger $d_f$ and the dependent smaller $b$ in $S(d_f)$. The lower $\langle w_s \rangle$ in large parts of the mesopelagic zone and below lead to a generally decreased transfer efficiency in $S(d_f)$ (Fig. 15 g). Consequently, the RLSs are shorter and enhance the silicate concentrations in surface waters in diatom-dominated regions (Fig. 15 e). In the subtropical gyres, the silicate diminishes even further, which is likely related to the enhanced bulk phytoplankton growth through higher phosphate concentrations. Phosphate is remineralized more rapidly than silicate and thus can be transported downstream the equatorial current into the subtropical gyres, where it is enhanced in $S(d_f)$ compared to M$^4$AGO (Fig. 15 b). As a consequence of the potential sensitivity of biogeochemical cycles on $d_f$, more knowledge of processes affecting the aggregate microstructure is required, among them e. g. compaction through repacking and zooplankton egestion of fecal pellets.

Diazotrophs in HAMOCC can grow independently of nitrate on phosphate. They are regionally confined to warm tropical and subtropical regions (Paulsen et al., 2017, 2018). Diazotrophs modulate the phosphate concentration in the subtropical gyres. The diazotrophs global primary production increases from about 2.2 Gt C yr$^{-1}$ in the standard run to $\approx 2.9$ Gt C yr$^{-1}$ in the M$^4$AGO run. A slower growth response to phosphate concentrations, enforced by raising the half-saturation constant from 0.05 µmol L$^{-1}$ to 0.10 µmol L$^{-1}$ in $S(K_{P,\text{diaz}})$, increases the phosphate concentrations in the subtropical gyres as to more than 150 % (Fig. 15 c). The primary production through diazotrophs reduces significantly from formerly global $\approx 2.9$ Gt C yr$^{-1}$ to $\approx 1.9$ Gt C yr$^{-1}$ in $S(K_{P,\text{diaz}})$, and regionally particularly in the equatorial Panama basin. By contrast, diazotrophs feature a primary production of more than 5 Gt C yr$^{-1}$ and 7 Gt C yr$^{-1}$ in $S(d_{p,\text{frustule}})$ and $S(d_f)$, respectively, due to the increased phosphate concentrations in the subtropical gyres. Diazotrophs in the model only produce organic matter. In $S(K_{P,\text{diaz}})$, the lower diazotrophs primary production in the in the equatorial upwelling regions (ETA and ETP) thus lead to a shift of the aggregate composition towards higher opal to detritus ratios which slightly increases the transfer efficiency and reduces the





available silicate (Fig. 15 f). Phosphate previously utilized by diazotrophs in the Panama basin now partially populates the downstream equatorial current and increases the subtropical gyres phosphate concentrations. In comparison to the standard run, this phenomenon is generally intensified in M$^4$AGO through the shallower remineralization and lower transfer efficiency in the subtropical gyres (Fig. 8 c). In conclusion, the representation and effects of diazotrophs in HAMOCC, particularly in

conjunction with M$^4$AGO, require further future evaluation.

### 3.10   Current limitations of M$^4$AGO

Developing M$^4$AGO, we followed a process-oriented approach and explicitly incorporated ballasting and microstrucure of aggregates of heterogeneous composition to calculate the mean sinking velocity. Introducing such complexity in ESMs typically comes at the cost of high computational efforts. This is a non-negligible factor for model development which we reduce to a

minimum with M$^4$AGO. Acknowledging the trade-off between increasing model complexity and computational limitations let us to deploy a number of simplifying assumptions that we critically review in the following.

In the euphotic zone of the oceans, changing phytoplankton community structure, phytoplankton growth, decay, and grazing though zooplankton leads to a dynamic supply of various small particles that potentially aggregate and eventually sink and become remineralized. Assuming homogeneous composition of aggregates throughout a dynamic steady state size distribution

is therefore only the first step to a model representation of marine snow, where local diversity of particles and the processes of aggregation and fragmentation are explicitly resolved. By assuming a dynamic steady state for the size distribution, we only represent the characteristic processes within the system and underestimate the variability of the number distribution slope which ranges between $b \approx 3.19 - 3.76$ compared to measurements, where $b \approx 2$ to $5$ (Guidi et al., 2009). As a consequence, M$^4$AGO probably underestimates the variability of sinking velocity, particularly in the upper ocean, where the size distribution

dynamically evolves.

In M$^4$AGO, TEPs are simplistically considered. In agreement with Mari et al. (2017), TEPs act as microstructure-loosening and buoyancy-adding for diatom-dominated aggregates. For CaCO$_3$, we only considered the coccolith size which decreases sinking velocity compared to an aggregate composed of intact coccospheres. For simplicity, we assumed that aggregated TEP and intact cells sink as fast as coccoliths and detritus. To decipher the role of TEPs for aggregate formation (Passow, 2002; Mari

et al., 2017), our model would require i) to represent TEPs and disintegrating coccospheres, and ii) to consider aggregation and fragmentation processes explicitly to depict the dynamic evolution of marine aggregate distributions. Such dynamic approach is necessary to e. g. study short term events of high POC export hypothesized to be driven by silicate depletion and TEP release by diatoms and their subsequent aggregation (Martin et al., 2011). Furthermore, an explicit representation of aggregation and fragmentation enables transient size distributions and would allow for a direct comparison to a growing number of particle

distribution slope measurements (e. g. Guidi et al., 2009).





## 4 Summary & Conclusions

We implemented the novel, *Microstructure, Multiscale, Mechanistic, Marine Aggregates in the Global Ocean* (M⁴AGO)
scheme in HAMOCC to improve the representation of the biological carbon pump in an Earth System Model framework.
M⁴AGO accounts for the heterogeneity and microstructure of aggregates and thus clearly defines measurable statistic aggre-

gate properties in HAMOCC. M⁴AGO links the nutrient and silicate cycle closer together by incorporating opal and other
ballasting minerals in aggregate formation and sinking. This lets us to introduce a consistent $Q_{10}$ temperature-dependent dis-
solution and aerobic remineralization of opal and POC, respectively.

  In contrast to the standard HAMOCC version, M⁴AGO well represents and provides a mechanistic understanding for the
recently published global transfer efficiency pattern. We identify primary particle size, particularly of diatom frustules, and

the compaction of aggregates with depth as strong driving factors for sinking velocity that, in combination with temperature-
dependent remineralization, co-determines the high POC transfer efficiency in high latitudes and upwelling regions. Our model
results support previous findings that $CaCO_3$ with its high density acts as ballasting mineral in calcifier-dominated regions of
the ocean. The changed transfer efficiency pattern in combination with the $CaCO_3$ to detritus rain ratio alters regional $CO_2$
fluxes, while the global uptake remains the same as in the standard run when atmospheric $CO_2$ is linearly increased without

climate feedbacks.

  Highest uncertainties in parametrizing M⁴AGO are with respect to the weakly constrained primary particle surface proper-
ties and their likely effect on the related microstructure of aggregates. Since sinking velocity and transfer efficiency are highly
sensitive to microstructure of aggregates, gaining insights into the controlling factors and processes for microstructure is de-
sirable. Future model development for the representation of marine aggregates would highly benefit from sub-aggregate scale

measurements of microstructure, adhesive surface properties, and primary particle composition.

  Our findings and the underlying model concept suggest a number of implications. First, size of aggregate constituents, par-
ticularly of diatom frustules, as potential factor for high sinking velocity suggest to widen the perspective of mineral ballast
studies towards a size-and-ballast hypothesis. Further, it requires to factor in the different temperature-dependent remineraliza-
tion and dissolution rates that aggregates experience during their descend. Accounting for cell size and morphology will aid

to better assess the role of the phytoplankton community size structures on POC fluxes, particularly in nutrient-rich upwelling
regions, where a wide, variable size spectrum of diatoms prevail. Second, the indirect temperature effect on phytoplankton cell
size (e. g. Daufresne et al., 2009) poses the challenging task to resolve the temperature-adapting cell size structure of the phy-
toplankton community in global carbon cycle models to depict the potential effect on sinking velocity and thus the biological
feedback on rising $CO_2$ in the atmosphere.

In conclusion, M⁴AGO provides well-defined aggregate properties in an ESM framework and can thus serve as a testbed
to upscale aggregate-associated processes to potential global impacts on biogeochemical cycles, and, in particular, on the
biological carbon pump.





*Code and data availability.* Primary data and code for this study is stored and made available through the Max-Planck-Gesellschaft Publications Repository: https://pure.mpg.de

## Appendix A: Additional aggregate properties

In addition to the aggregate properties presented in Fig. 6, $M^4AGO$ involves the aggregate mean stickiness and the number distribution slope that are tightly connected to each other via $d_f$ (Fig. A1). In addition, the porosity of aggregates, Eq. (6), can

be deduced from aggregate size, primary particle size, and $d_f$. Porosity is frequently calculated for aggregates (Alldredge and Gotschalk, 1988; Ploug et al., 2008). Its potential dependency on the microstructure and primary particle size is, however, seldom covered. We therefore calculated the mean volume-weighted porosity of aggregates, $\langle \phi \rangle_V$, to provide a perspective on how aggregate porosity varies with the aggregate properties shown in Fig. 6 in the global ocean (Fig. A1).

As defined by the attributed primary particle stickiness, aggregates in diatom-dominated regions show the highest stickiness

values in surface waters since the virtual TEP particles linked to detritus increase the $\langle \alpha \rangle$ in $M^4AGO$ (Fig. A1). At the bottom of the mesopelagic zone, $\langle \alpha \rangle$ is almost homogeneous apart from the OMZ regions, where the lower anaerobic remineralization retain higher detritus concentration that lead to higher $\langle \alpha \rangle$ in conjunction with diatom frustules. The number distribution shows an inverted picture compared to $\langle \alpha \rangle$ with smallest decay slope, $b$, in diatom-dominated and OMZ regions and strongest decline in calcifier-dominated surface waters (Fig. A1 c,f). In $M^4AGO$, $\langle \phi \rangle_V$ is tightly connected to $\langle \alpha \rangle$ in the euphotic zone

(Fig. A1 b). With increasing depth, aggregates get compacted, $d_f$ increases, and the maximum size of aggregates decrease. Both lead to lower $\langle \phi \rangle_V$ particularly in the diatom-dominated regions. In OMZs, however, $\langle \phi \rangle_V$ is undiminished high. Generally, the exceptional behavior of aggregate properties in OMZs due to implicitly modelled TEPs is likely overestimated. TEPs possess higher remineralization rate than detritus (Mari et al., 2017), which likely reduces the TEPs occurrence in deep OMZs and thus their influence on aggregate properties.





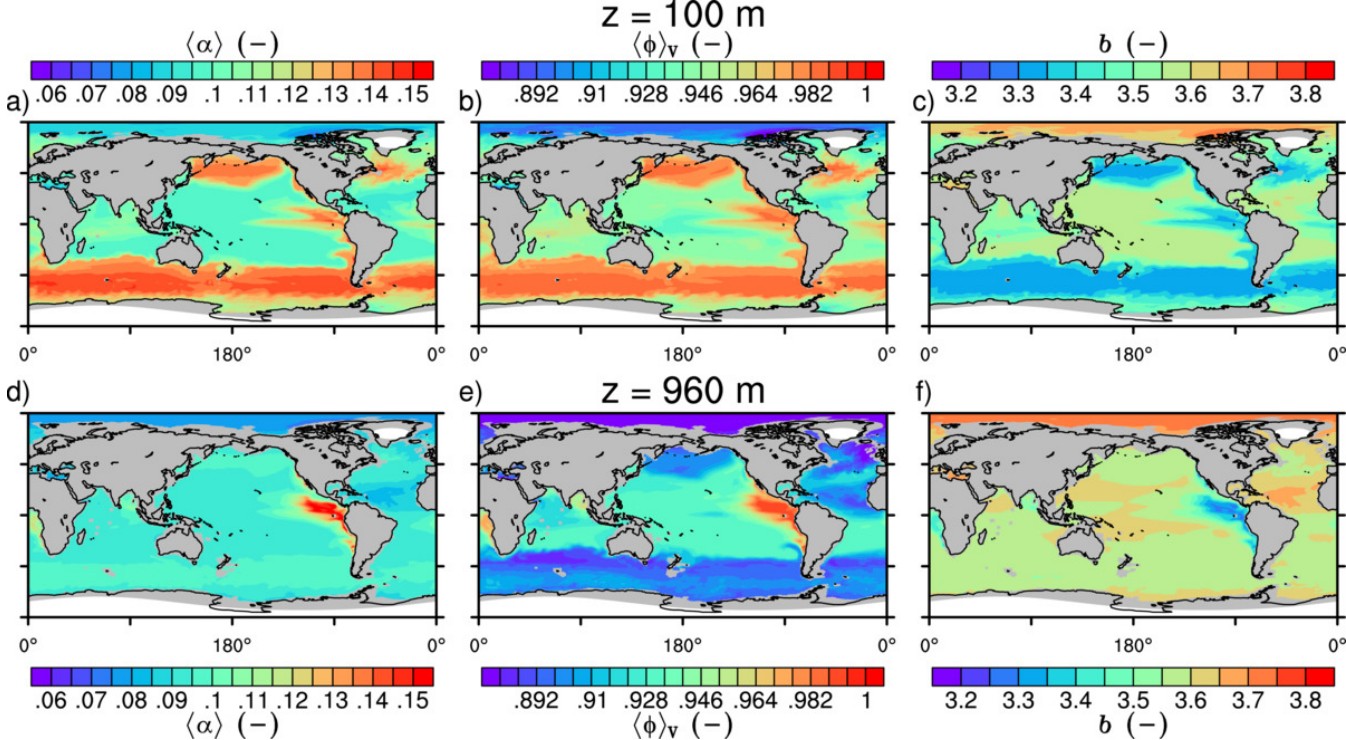

**Figure A1.** Mean stickiness $\langle\alpha\rangle$, Eq. (22), volume-weighted porosity, $\langle\phi\rangle_V$, and the number distribution slope, $b$, Eq. (21).

## Appendix B: Effective Martins slope in M$^4$AGO

The effective Martin curve slope $\beta'$ provides a meaningful measure on the attenuation of POC fluxes. This made $\beta'$ to be a widely used measure to evaluate POC concentration and flux observations (e. g. Lutz et al., 2007; Lam et al., 2011; Marsay et al., 2015). In M$^4$AGO, we found a $\beta'$ pattern similar to the inverse of the transfer efficiency (Fig. B1 compared to Fig. 8, respectively). The smallest $\beta'$ are found in the models OMZ regions and the shallow Arctic shelf regions. In the North Pacific and North Atlantic, $\beta'$ features values of about 0.47 to 0.60 and reaches maximum values of about 1.31 in the subtropical gyres. Qualitatively, the pattern thus follows and underpins previously suggested POC flux attenuation pattern (Marsay et al., 2015; Weber et al., 2016; DeVries and Weber, 2017). By contrast, in our standard run, we found, apart from OMZs, a rather homogeneous $\beta'$ with a global value of $\langle\beta'\rangle_{A,Martin} \approx 0.82$ that is smaller than the prescribed value of $\beta = 1.00$. This discrepancy can be attributed to a number of processes. First, the global $\langle\beta'\rangle_{A,Martin}$ is reduced by the lower anaerobic remineralization in OMZs. Second, turbulent diffusion and vertical transport processes represented by MPI-OM in addition to sinking contribute to vertical POC concentration profiles and fluxes (Boyd et al., 2019). Third, the artificial numerical diffusion inherent to HAMOCCs implicit upstream scheme for particle sinking contributes to higher mass transport to depth than prescribed by $\beta = 1$. This inherent numerical diffusion is, however, implicitly accounted for during the process of tuning $\beta$ in the model.


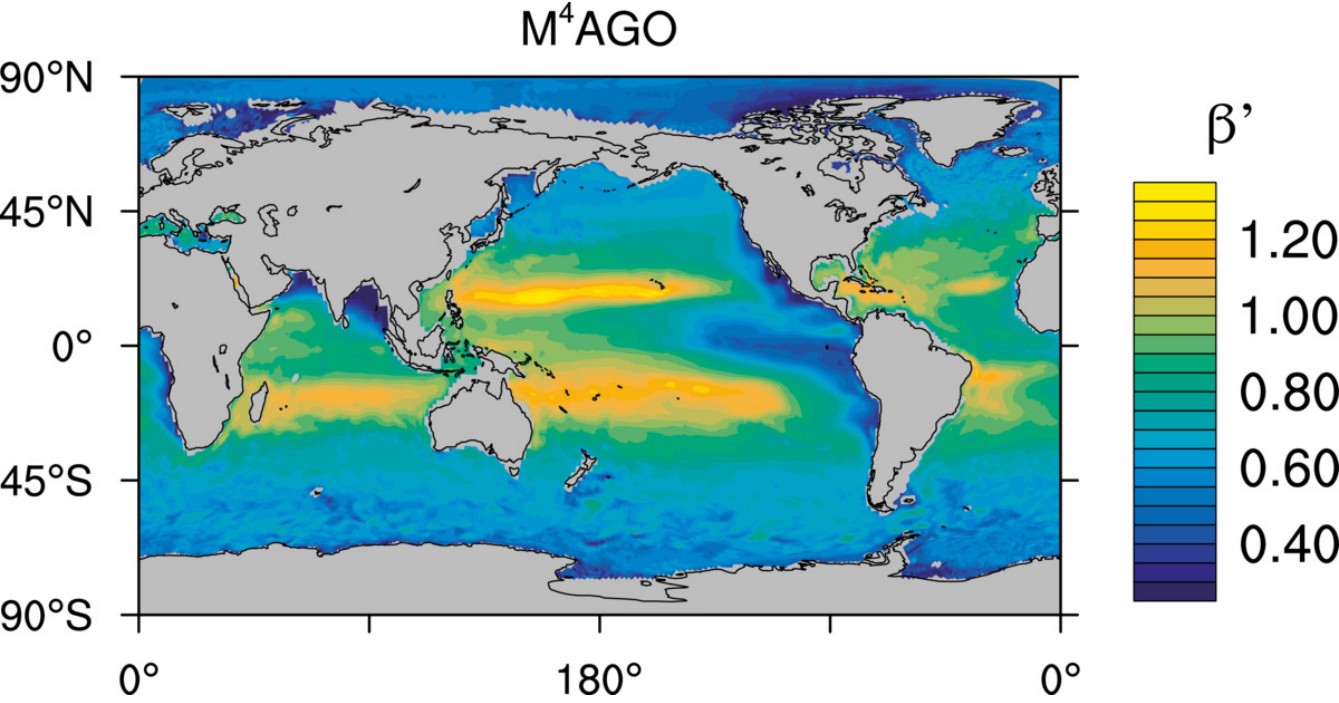

**Figure B1.** Effective Martin curve slope for POC in M$^4$AGO, $\beta'$.

## Appendix C: Seasonal transfer efficiency

The inversely identified transfer efficiency pattern by Weber et al. (2016) provides an estimate on time-integrated climatological POC fluxes. In regions of high seasonal variability in primary production, POC fluxes can undergo strong seasonal variation and so does transfer efficiency (Lutz et al., 2007). For example, if we assume an average sinking speed of about $25\,\mathrm{m\,d^{-1}}$, a pulsed

5   flux at 100 m reaches the depth of 1000 m about a month later and can strongly alter flux and concentration profiles (Lam et al., 2011; Giering et al., 2017). Accordingly, single measured POC concentration profiles can even show higher concentration at depth than in surface waters which results in negative $\beta'$ values and thus higher transfer efficiency than one. In turn, transfer efficiency can be extremely small at the beginning of a phytoplankton bloom. Seasonal transfer efficiency can thus heavily deviate from the climatological state. This seasonal behavior is reflected in M$^4$AGO which shows high transfer efficiency in

10   late autumn and early times of low primary production after the bloom in high latitudes (Fig. C1). Even though the standard run exhibits a similar qualitative pattern, the seasonal amplitude of the transfer efficiency in high latitudes is lower in M$^4$AGO compared to the standard run.

*Author contributions.* Joeran Maerz performed the M$^4$AGO model development, the HAMOCC model runs and wrote the manuscript. Katharina D. Six and Irene Stemmler significantly contributed in tuning the M$^4$AGO sinking scheme in HAMOCC and aided during imple-



**Figure C1.** Seasonal evolution of the transfer efficiency in M$^4$AGO. Note the difference of the colorbar values compared to Fig. 8.

mentation. Soeren Ahmerkamp contributed by significant discussions on aggregates microphysics. All authors of the manuscript critically discussed the presented results and contributed by providing valuable feedback during the manuscript compilation.

*Competing interests.* The authors declare that they have no conflict of interest.



*Acknowledgements.* The authors thank Thomas Weber for sharing the results on the transfer efficiency, and Adrian Burd for discussion on fractal dimension of aggregates. The authors thank Bo Liu for the internal review and comments on the manuscript. J. Maerz thanks U. Feudel for earlier discussions on the topic of marine aggregates. The *Multiscale Approach on the Role of Marine Aggregates (MARMA)* project is funded by the Max-Planck-Society (MPG). We acknowledge funding from European Union Horizon 2020 research and innovation programme under grant agreement No 641816 (CRESCENDO). This work contributes to the project PalMod of the German Federal Ministry of Education and Research (BMBF) as Research for Sustainability initiative (FONA). Thanks to Thyng et al. (2016) for providing the cmocean colormap. All simulations were performed at the German Climate Computing Center (DKRZ).



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
