# Peer review of "Microstructure and composition of marine aggregates as co-determinants for vertical particulate organic carbon transfer in the global ocean"

_Biogeosciences, 2019_

## Referee Comment (RC1) · Anonymous Referee #1 · 22 Nov 2019

**1   General comments**

The authors present a new sinking scheme for marine aggregates that takes into account selected important effects of aggregate microstructure (such as estimates of porosity, TEP content, and density based on the aggregate composition, which is derived from HAMOCC tracer concentrations) and of the resulting estimated aggregate size distribution. The authors achieve this without the use of an explicit aggregation model, and without introducing different particle size classes, thereby keeping the

scheme very affordable, affordable enough for long-term global carbon cycle modelling. Because several of the incorporated mechanisms that affect the sinking of particulate carbon in the ocean were previously neglected in global carbon cycle models, the presented work is a welcome contribution to the field and should be published.

While the presented sensitivity experiments with respect to selected parameters of the sinking scheme seem well-placed in the manuscript, I would suggest to reconsider wether the $CO_2$-sensitivity experiments would be better-placed in a separate manuscript, 1) given the length of the manuscript, 2) given that the title does at least not explicitly reflect those results, and 3) given some inconsistencies compared to atmosphere–ocean $CO_2$ flux observations described below that may be better addressed in more detail in a separate manuscript, specifically aiming at the role of aggregate and sinking speed changes in response to greenhouse gas emissions and climate change.

The manuscript provides a large amount of sinking-relevant background information that is interesting on its own, and necessary to understand the (incorporated or neglected) processes in the new sinking scheme. The description of the new sinking scheme itself is also very detailed, making the results reproducible – also with the help of the very well-documented supplementary material. This, combined with the presented extensive analysis and selected parameter sensitivity experiments, understandably leads to a rather long manuscript. However, I do believe that the manuscript can still be shortened and readability can be improved by clarifying / simplifying some formulations (see comments on selected sentences below).

Some additional minor comments to improve/clarify the manuscript prior to publication, as well as some typos are listed below.

[Figure]

**2  Minor comments**

*Abstract, line 10:* I would suggest to replace *"which has been recently constrained by"* by *"as recently constrained by"*, to clarify that this particular latitudinal pattern of POC transfer efficiency is reproduced. I think it would be appropriate to mention (here or at least later on page 3 around lines 6-12 or 29-31) that previous estimates of transfer efficiency showed an opposing latitudinal pattern (Henson et al. 2012).

*Abstract, lines 14-16:* Please rephrase. In standalone runs *with* rising carbon dioxide... M$^4$AGO only alters the *simulated* fluxes. Sentences could maybe also be shortened, e.g.: *Using M$^4$AGO in standalone runs with prescribed rising $CO_2$ concentrations (without climate feedback) leads to higher $CO_2$ uptake in the Southern Ocean, and to lower $CO_2$ uptake in the subtropical gyres compared to the standard run, while the global oceanic $CO_2$ uptake remains the same.*

*Abstract, lines 12-13:* Please rephrase / clarify. Are temperature effects contributing ("driving factor") to the simulated transfer efficiency pattern? Wouldn't at least the temperature effect on viscosity counteract the simulated pattern? Or does this refer to the newly introduced temperature-dependent remineralization of POC, which, if I understand correctly, least counteracts the high sinking speeds in the high latitudes (countours in Fig. 9b)?

*Page 2, line 17, "The sinking velocity of aggregates is primarily determined by their size."* I understand that aggregate size does matter, but is it really the main factor? Reference? Even very large aggregates can be rather buoyant (e.g., Riebesell 1992).

*Page 3, line 15:* Please replace "while ignoring" with "while neglecting" (the effects are still discussed).

*Page 3, lines 29-31:* As mentioned above, I would point out that Henson et al. (2012) suggested an opposing pattern. Would it be possible to reproduce also this opposing pattern with M$^4$AGO? I think a brief discussion of this issue would be interesting –

potentially regarding the presented sensitivity experiment with smaller diatom frustules showing much lower transfer efficiencies in high latitudes?

*Page 4, line 28:* Please consider including the equation for opal dissolution explicitly, also to better understand the given dissolution rates in Table 1. As far as I understand / looking at the HAMOCC code, the opal dissolution rate given in Table 1 corresponds to 7°C?

*Page 5, line 9-11:* Please rephrase; e.g., *... is eventually computed from a number distribution that is truncated at the minimum and maximum aggregate diameters ..., and expressions for the mass and sinking velocity of aggregates of a particular diameter:*

*Page 5, line 26:* It would be helpful to define the diameter *d* of an aggregate more accurately here. For example, is the diameter of an aggregate with $d_f$=1 (i.e., a chain) just given by its length?

*Page 6, line 7:* Please move reference to "well known" Stokes (1851) here.

*Page 6, line 26:* I am a little lost here. What is the motivation for this paragraph? What is *n*? And why is that equation only true for $n \neq n_p$?

*Page 6, line 27:* Do I understand correctly that *"same heterogeneity in a size spectrum"* means that all aggregates of a particular composition / in a particular grid cell are assumed to have the same heterogeneity/microstructure/$d_p$ (for all aggregate diameters d)?

*Page 7, line 9:* Please define $V_{p,i}$ more accurately; volume of the primary particles per unit volume of sea water?

*Page 7, line 6-8:* Maybe shorten to *"... tracer $C_i$, namely detritus, opal, calcite and dust."*

*Page 7, line 15:* Add *"Multiplication by the volume of the mean primary particle then yields..."*; helps the reader / I didn't see this at first.

*Page 8, line 3:* Please add reference to traditional scaling relationship.

*Page 9, line 1:* For consistency, if j=0..3 here, also add definitions of a/b$_{j=0}$ in last paragraph of page 8.

*Page 9, line 13:* It is not clear to me what *"dynamic steady state"* means.

*Page 9, line 20-21:* I am wondering why the assumption of Reynolds numbers between 0.1 and 10 here is okay, while the authors go through the extra trouble of deriving expressions for the sinking speed for even smaller and even larger Reynolds numbers in Section 2.2.2. Is there a reason for this?

*Page 10, line 6:* Please clarify / see above comment on "in a size spectrum": *... as one value across all aggregate sizes.*?

*Page 11, line 10:* Shouldn't it read: *When detritus from the frustules is remineralized, it is replaced...*?

*Page 12, line 2:* Please rephrase: *...thus decrease the fractal dimension of aggregates, and ii) ...*

*Page 15, Table 1:* Is it correct that the applied opal dissolution rate in the setup with M$^4$AGO is larger than that in the standard model setup? What is the resulting effect of the temperature-dependency here? Wouldn't the larger remineralization rate combined with the slower opal sinking speeds in the euphotic zone lead to very high opal production, and consequently to very low calcite production?

*Page 16, line 7:* I am not sure if I understand correctly: Is d$_{p,calc}$ chosen particularly small to avoid an overestimate of the volumetric density effect? If so, for clarity, *while accounting for* could be rephrased: *We set d$_{p,calc}$ ..., to account for ....*

*Page 16, line 24:* Sentence unclear to me; *...distinction between parameter tuning and model evaluation, when...* (?)

*Page 16, line 33:* Please rephrase: Since the adaptation of the sinking velocity *and thus*

*of the transfer efficiency to the remineralization and dissolution rates occurs within a few years, parameter variations aiming at a quantitative agreement with the transfer efficiency of Weber et al. were feasible.* (?)

*Page 17, lines 8-13:* Move to results section / next paragraph?

*Page 17, lines 12-13:* The annual mean of only one year seems rather short. Have you checked how sensitive your results are with respect to interannual variability due to, e.g., ENSO or deepwater formation variability?

*Page 18, lines 1-3:* Please rephrase / correct sentence structure (*"features"* can not refer to *"In the M⁴AGO run"*).

*Page 18, lines 3-4:* It is difficult to say from Fig. 4 wether the use of M⁴AGO really leads to smaller latitudinal variability, since the minima and the global mean p-ratio also seem lower than in the standard run. Maybe remove this statement or double-check?

*Page 20, line 8:* This is only shown in Fig. 7a, not in Fig. 7d.

*Page 21, Figure 7 (and page 26, Fig. 9):* It would be helpful to show (or at least describe) the location of WOA transect P16.

*Page 23, line 7:* Sentence unclear to me: ... we neglect this effect *versus*? Maybe delete *versus the lower primary particle binding forces*? What are those forces?

*Page 23, line 8:* "we neglect... and kept" For clarity, I would suggest to consistently stick to present-tense for the work performed for this study, and to past tense for previous results.

*Page 25, line 26:* Shouldn't it read *"...decay to 1/e of its initial value, ..."*?

*Page 27, line 9 / Figure 9:* The lower remineralization rates described here are hard to see in Figure 9b, also due to the missing label on the -30% (?) contour; maybe smaller contour intervals would help.

*Page 27, lines 29-30:* Maybe clearer: *The relative contributions provide information about the main driving factors for local sinking speed deviations from the global mean.*

*Page 28, lines 23-24:* Please rephrase / clarify sentence; e.g.: *$M^4AGO$ thus likely underestimates the spatial variability *and* relative contribution *of $b$ to $w_s$*.

*Page 30, lines 22-23:* Unclear sentence structure. Maybe: *Similar to POC fluxes, opal fluxes exhibit shorter RLSs in ..., while they exceed the standard RLSs in ...*

*Page 30, line 27:* "... fluxes are generally small." Is this true for both model versions? Not shown here, or is it?

*Page 30, lines 28-29:* Please add sedimentation flux in standard model for comparison.

*Page 31, line 7:* "The $M^4AGO$ run represents the PIC/POC fluxes equally well as the standard run." It would be interesting to know how well that is.

*Page 31, lines 8 and 11:* "... the scatter around the 1:1 line is reduced ..." (line 8) At least for some regions, e.g. for the Sub Antarctic Zone, the points are not really scattered around the 1:1 line. But I agree with the view that $M^4AGO$ reduces the variability in the POC/Si ratio (line 11). Isnt't this reduced variability / compression in the POC/Si fluxes expected, because the variability of the fluxes is only due to the variability of the POC/Si concentrations in $M^4AGO$ (POC and opal sink at the same speed), while in the standard model, varibility is also introduced due to differences between Si- and POC-sinking speeds?

*Page 32, Figure 12:* Do I understand correctly that each dot in the figure is a generated monthly mean data point, compared to the respective location and monthly mean of the last year in the model run?

*Page 33, line 34:* ... and the North American Westcoast (?)

*Page 35, lines 20-22 / page 36, Figure 14a:* If negative fluxes really do represent a net-$CO_2$ uptake by the ocean in Fig. 14a, as stated in the caption, the southern hemisphere

ocean acts as a net sink for atmospheric $CO_2$ (and not a source), and the northern hemisphere ocean acts as a net source (not a sink). Consequently, the oceanic $CO_2$ transport would be from south to north.

Irrespective of the sign / flux direction, these results are in stark contrast to $CO_2$ flux observations of net zonal mean outgassing at the Equator and net ocean $CO_2$ uptake in mid-latitudes (e.g., Figure 14 in Takahashi et al. 2019). Maybe this is just due to a plotting error in Figure 14?

I also am not sure if I understand the units in Figure 14a. Does the left axis show the net sea–air $CO_2$ flux accumulated over the respective $1°$ latitude band? If that is true, the values seem very large. I am guessing from Fig. 14a that the ocean $CO_2$ uptake accumulated in the southern hemisphere would then amount to around 0.2 GtC/yr/deg*60deg≈12 GtC/yr, which is an order of magnitude larger than the observed net uptake by the southern hemisphere ocean (south of $14°S$) of about 1.1 GtC/yr.

*Page 35, lines 22-24: "In the simulation with M⁴AGO, a stronger $CO_2$ uptake in the region ... coincides with ... increased transfer efficiency"* This is a very interesting point; does it still hold despite the (to my understanding) erroneous Figure 14a? To me it is surprising that the $CO_2$ fluxes do \*not\* differ more, despite the very different transfer efficiencies. Why do the $CO_2$ fluxes hardly differ south of, say, $55°S$, where the transfer efficiency difference is largest? Why is there hardly an effect in the Arctic Ocean?

*Page 39, line 7: Sentence structure. ... body size decreases with increasing water temperature. And increasing water temperature has been suggested to ...* (?)

*Page 39, line 10: Does "... such eco-physiological responses ..."* refer to the primary particle size change, or to other effects?

*Page 39, line 29:* ... increases the phosphate concentrations in the subtropical gyres

*by up to* 50% (?)

*Page 40, line 1-2:* Please rephrase (phosphate increases phosphate concentrations...), e.g. by: *...phosphate ... populates ... and reaches the subtropical gyres.*

*Page 41, lines 21-29:* This paragraph, describing the main implications of this study, is not formulated very clearly. *"Our findings ... suggest a number of implications."* Number=2, according to later "first" and "second"? Please rephrase second sentence. E.g., *First, the finding that the size ... is a potential contributor to high sinking speeds suggests that the ballast hypothesis needs to be extended to a size-and-ballast hypothesis.* What does *"it requires"* in line 23 refer to?

*Page 43, line 9:* As far as I understand $\beta$ is not prescribed in the standard run, but only the sinking speed, i.e., $\beta$ still depends on the remineralization (which varies with temperature and oxygen concentrations). How do you get to the value of $\beta$=1?

**3 Typos**

*Page 2, line 32:* Prim*ary* / fundamental(?) determining factor*s*?

*Page 3, line 29:* ...benefits from *an* order of...

*Page 4, line 23:* Bar over $w_s$ meaning global mean / annual mean?

*Page 4, line 27: The* opal dissolution rate...

*Page 10, line 27:* ... enhance *the* sinking velocity...

*Page 18, line 7:* ... which also lead (plural)

*Page 18, line 8:* ... from either satellite *data*, in situ observations, or models lead to partly contrasting pattern*s* (add "data" and plural "s")

*Page 19, line 5:* Both model simulations show *a* similar pattern (add "a")

*Page 22, line 11:* Use *"By contrast"* rather than *"In turn"*?

*Page 22, line 13:* ... during *the* aggregates*'s* descent. (?)

*Page 23, line 3:* It is likely that (no comma)

*Page 23, line 17:* linear*ly* increasing

*Page 23, line 30:* ...from *the* relationship... and *the* transfer efficiency.

*Page 24, line 5:* M$^4$AGO posses*es*...

*Page 24, line 11:* ...allows *to more reliably constrain POC transfer efficiency*

*Page 25, line 5:* exten*t*

*Page 25, line 30:* ... in surface waters and *the* upper mesopelagic zone...

*Page 25, lines 32-33:* ... the RLSs... *are* similar or *slightly* (?) longer, or *smaller* again in ... The longer RLS*s* in the mesopelagic zone... (plural)

*Page 27, line 16:* In summary, *the* temperature-dependence...induce*s*...

*Page 27, line 28:* ...X$_{i,z}$ *is* (not as)

*Page 28, line 20:* ... given the general importance of the microstructure....

*Page 35, line 29:* ... in M$^4$*A*GO on *the* CO$_2$ uptake ...

*Page 39, line 34:* one *in the* too much; thus lead*s* to (singular)

*Page 40, line 13:* grazing th*r*ough zooplankton

*Page 43, line 7:* underpins *the* previously...

*Page 50, line 33:* initials for Núñez-Riboni

*Page 53, line 4:* Aiko Voigt (not Vogt?)

*Page 53, line 19:* please check reference / entry missing?

*Page 54, line 27:* Ocean-Atmosphere

*Page 55, line 11: please check / C. R. Geoscience?*

**References**

Henson, S. A., Sanders, R., and Madsen, E.: Global patterns in efficiency of particulate organic carbon export and transfer to the deep ocean, Global Biogeochemical Cycles, 26(1), http://doi.org/10.1029/2011GB004099, 2012.

Riebesell, U.:The formation of large marine snow and its sustained residence in surface waters, Limnology and Oceanography, 37(1), pp. 63–76, http://doi.org/10.4319/lo.1992.37.1.0063, 1992.

---

## Referee Comment (RC2) · Anonymous Referee #2 · 5 Dec 2019

Review of Maerz et al: Microstructure and composition of marine aggregates as co-determinants for vertical particulate organic carbon transfer in the global ocean

The authors present a new scheme for the calculation of the mean sinking velocity of marine aggregates as a function of the aggregate composition and the fractal dimension. This scheme is reported to be cost-efficient and hence useful in large-scale ocean models. The model is described in detail and carefully evaluated. The authors report a substantial improvement in the simulation of the latitudinal pattern of POC transfer efficiency.

[Figure]

This is an impressive effort and worth of publication. I have some specific comments that should be addressed before publication.

Writing style: Sentences are very long and not always clear. This is particularly true for the introduction and model description.

General comments:

- I miss a comparison with the stochastic, Lagrangian model of sinking biogenic aggregates in the ocean (SLAMS) by Jokulsdottir and Archer. Jokulsdottir, T. and Archer, D.: A stochastic, Lagrangian model of sinking biogenic aggregates in the ocean (SLAMS 1.0): model formulation, validation and sensitivity, Geosci. Model Dev., 9, 1455–1476, https://doi.org/10.5194/gmd-9-1455-2016, 2016.

- Please make the model code publicly available. It is not in the repository that you mention.

- explain ALL abbreviations and symbols used in the figures in each and every figure caption.

- I am quite worried about the high buoyancy of diatom-dominated aggregates through the TEP formulation. This needs more justification. Do you here assume that all organic carbon has the same density as TEP? That would explain your low density of diatom-dominated aggregates. Is there sufficient evidence for such behavior?

Abstract:

Line 14: too much information given: delete rising $CO_2$ and without $CO_2$ climate feedback.

Introduction:

- P.2 Please give more references for your statements, especially in the first paragraph. No reference given between line 5 and 11.

- P. 2, line 17: "The sinking velocity of aggregates is primarily determined by their size". This needs a reference. I would argue it is density, e.g. Iversen and Robert, http://dx.doi.org/10.1016/j.marchem.2015.04.009 . The next sentence also needs a reference (line 19).

- P. 2, line 32: primer –> primary?

Model description

- It would be very helpful to have a table with all symbols used in the equations at the beginning of section 2.1

- P. 5, line 2, what is meant with "terminal sinking velocity"? I suggest to delete "terminal"

- Eq. 3: I can guess what is meant with dd, but it is easily misunderstandable.

- P. 5, line 28: What is meant by a "primary particle". How does that differ from "a particle"?

- P. 7, line 1: What is meant by "the total number of one primary particle type" ? The total number of one should be one. Do you mean "of particles of one particle type"?

- P. 8, line 5: "mean primary particle size, (. . .) which we apply as a lower integration bound". Please give a justification for this choice.

- P. 10, line 5: no reference to Engel et al 2004? Engel, A. , Thoms, S., Riebesell, U. , Rochelle-Newall, E. and Zondervan, I. (2004) Polysaccharide aggregation as a potential sink of marine dissolved organic carbon. Nature, 428 . pp. 929-932. DOI 10.1038/nature02453.

- Figure 2: explain abbreviations and symbols in each and every figure caption.

- P. 11, line 18: how is m_e and m_potential calculated? I can't follow whether the masses of opal and of TEP are taken into account correctly to calculate the density

of the diatom-aggregate. Do you here assume that all organic carbon has the same density as TEP? That would explain your low density of diatom-dominated aggregates. Please clarify.

- P. 13, line 1: mention that this forcing is based on ERA reanalysis (be specific) and avoid the abbreviation OMIP which you don't explain (or explain it)

- P. 13, line 3/4: are these global numbers? What are corresponding model parameters?

- Table 1: caption: "The value for the Martin curve...". Why is this single parameter given in the caption, please add it it to the list of parameters in the main body of the table.

- P. 16, line 4: "a minor role in biogenic fluxes". This statement needs a reference.

- P. 16, line 33/34: "adaptation .. within a few years." Is adaptation the right word here? Maybe "an equilibrium was established"? or its change after a few years was small..

Results

- P. 22, line 12-13: "diatom-dominated aggregates feature a high buoyancy through TEP." Is there any evidence for such behavior or is this a major model bug?

- P. 23, line 25: I assume $z_0$ is 100m, please clarify.

- Line 26: "to about 1000m" → at 1000 m.

- P. 28, line 29: this is not shown in Fig 7a, you only show mean density, not the effect of opal on density.

- Line 30: any indication in the literature and any scientific explanation why silicate frustule size affect the sinking speed if not by density?

- Fig 10: colorbar label: conribution → contribution(add 't')

- P. 35 and Figure 14a: what is the reason of showing cumulative $CO_2$ fluxes integrated

over latitude? Please show just the zonal means, that's much easier to understand and compare to data. The units should not include per degree if it is cumulative.

- Figure 14c-k: cumulative fluxes make more sense here. I'd prefer actual fluxes/time and then the difference between the two could be cumulative. Then, one y-axis might also be enough.

- P. 37, line 24: suggested → hypothesize (careful which tense you use). Also, please please back up this hypothesis with literature.

- P. 38: you have not shown silicate distribution – is that reasonable? You refer to low transfer-efficiency in silicifier-dominate region, but this is not the case in the Southern Ocean, nor do you see much of an impact in Figs 15 a and d in the Southern Ocean, which is THE region dominated by silicifiers. This needs more explanation.

---

## Author Comment (AC1) · 27 Jan 2020

**1  General remark**

The reply is structured as follows:

- Referee comment

  ⇒ Authors reply

    → Modification(s) in the manuscript. "old" → "new"

**2  Reply to Referee #1**

**2.1  General comments**

- The authors present a new sinking scheme for marine aggregates that takes into account selected important effects of aggregate microstructure (such as estimates of porosity, TEP content, and density based on the aggregate composition, which is derived from HAMOCC tracer concentrations) and of the resulting estimated aggregate size distribution. The authors achieve this without the use of an explicit aggregation model, and without introducing different particle size classes, thereby keeping the scheme very affordable, affordable enough for long-term global carbon cycle modelling. Because several of the incorporated mechanisms that affect the sinking of particulate carbon in the ocean were previously neglected in global carbon cycle models, the presented work is a welcome contribution to the field and should be published. While the presented sensitivity experiments with respect to selected parameters of the sinking scheme seem well-placed in the manuscript, I would suggest to reconsider wether the CO2 -sensitivity experiments would be better-placed in a separate manuscript, 1) given the length of the manuscript, 2) given that the title does at least not explicitly reflect those results, and 3) given some inconsistencies compared to atmosphere–ocean CO2 flux observations described below that may be better addressed in more detail in a separate manuscript, specifically aiming at the role of aggregate and sinking speed changes in response to greenhouse gas emissions and climate change. The manuscript provides a large amount of sinking-relevant background information that is interesting on its own, and necessary to understand the (incorporated or neglected) processes in the new sinking

scheme. The description of the new sinking scheme itself is also very detailed, making the results reproducible – also with the help of the very well-documented supplementary material. This, combined with the presented extensive analysis and selected parameter sensitivity experiments, understandably leads to a rather long manuscript. However, I do believe that the manuscript can still be shortened and readability can be improved by clarifying / simplifying some formulations (see comments on selected sentences below).

Some additional minor comments to improve/clarify the manuscript prior to publication, as well as some typos are listed below.

$\Rightarrow$ We thank the reviewer for her/his comprehensive, constructive and positive review. With respect to the section on atmosphere-ocean carbon dioxide fluxes (Sect. 3.8 Regional $CO_2$ uptake), we admit having used an erroneous y-label for Fig. 14 a. Indeed, it is $Gt\,C\,yr^{-1}$ for the cumulative zonal $CO_2$ fluxes. We apologize for the confusion. Apart from that, we are confident that the results are in agreement with present knowledge, which we comment on below. Since the $CO_2$ fluxes are of clear interest in an ESM framework and a benchmark for the development of such a comprehensive aggregate-representing model component, we decided to keep this present section. For the sake of clarity, we changed the section title.

$\rightarrow$ Changed the unit (see Fig.1).
Section Title: "Regional $CO_2$ uptake" $\rightarrow$ "Regional $CO_2$ fluxes"

**2.2 Minor comments**

- Abstract, line 10: I would suggest to replace "which has been recently constrained by" by "as recently constrained by", to clarify that this particular latitudinal pattern of POC transfer efficiency is reproduced. I think it would be appropriate to mention (here or at least later on page 3 around lines 6-12 or 29-31) that previous estimates of transfer efficiency showed an opposing latitudinal pattern (Henson et al. 2012).

  $\Rightarrow$ Thanks for your suggestions. We did the replacement accordingly. In addition, we now mention the opposing pattern on page 3, line 30, see also below.

[Figure]

Figure 1: Units from manuscript Fig. 14 a corrected ("Gt C yr$^{-1}$ deg$^{-1}$" → "Gt C yr$^{-1}$").

$\rightarrow$ "which has been recently constrained by" → "as recently constrained by",
"...more reliable than previous estimates (e. g. Henson et al., 2012; Marsay et al., 2015)" → "...more reliable than previous estimates with partly opposing latitudinal pattern (e. g. Henson et al., 2012; Marsay et al., 2015)"

- Abstract, lines 14-16: Please rephrase. In standalone runs with rising carbon dioxide... M$^4$AGO only alters the simulated fluxes. Sentences could maybe also be shortened, e.g.: Using M$^4$AGO in standalone runs with prescribed rising CO2 concentrations (with-out climate feedback) leads to higher CO2 uptake in the Southern Ocean, and to lower CO2 uptake in the subtropical gyres compared to the standard run, while the global oceanic CO2 uptake remains the same.

$\Rightarrow$ Rephrased.

$\rightarrow$ "In ocean standalone runs and rising carbon dioxide ($CO_2$) without $CO_2$ climate feedback, M$^4$AGO alters the regional ocean-atmosphere $CO_2$ fluxes compared to the standard model." $\rightarrow$ "Prescribing rising carbon dioxide ($CO_2$) concentrations in standalone runs (without climate feedback), M$^4$AGO alters the regional ocean atmosphere $CO_2$ fluxes compared to the standard model."

- Abstract, lines 12-13: Please rephrase / clarify. Are temperature effects contributing ("driving factor") to the simulated transfer efficiency pattern? Wouldn't at least the temperature effect on viscosity counteract the simulated pattern? Or does this refer to the newly introduced temperature-dependent remineralization of POC, which, if I understand correctly, least counteracts the high sinking speeds in the high latitudes (countours in Fig. 9b)?

$\Rightarrow$ Thanks. We referred here to the temperature effect on remineralization. We clarified it.

$\rightarrow$ "a driving factor" $\rightarrow$ "a driving factor for remineralization"

- Page 2, line 17, "The sinking velocity of aggregates is primarily determined by their size." I understand that aggregate size does matter, but is it really the main factor? Reference? Even very large aggregates can be rather buoyant (e.g., Riebesell 1992).

$\Rightarrow$ This is an interesting comment and we realize, also by the same comment of reviewer #2, that there seems to be much confusion about the controlling factors for sinking velocity, which deserves a publication on its own (being in progress). We want to emphasize here that we clearly state in the follow-up sentence that structure and composition of aggregates regulate the excess density and can thus have a high impact on sinking velocity (we now provide a reference for it). Nevertheless, we would here argue from the mathematical perspective. For simplicity and neglecting the changing drag coefficient for particles with higher Reynolds particle numbers,

let's consider the Stokes sinking velocity for low particle Reynolds numbers:

$$w_s(d, \rho_f, \ldots) = \frac{1}{18\,\mu}\,(\rho_f - \rho)\,g\,d^2 \tag{1}$$

where $d$ is the diameter, $\mu$ the molecular dynamic viscosity, $\rho_f$ the aggregate density, $\rho$ is the density of the ambient fluid, and $g$ is the gravitational acceleration constant. It is obvious that $w_s \propto (\rho_f - \rho)$ and $w_s \propto d^2$. Hence, sinking velocity only linearly increases with aggregate density, while it increases with a power law relationship of the diameter. This suggests that size is indeed the primary factor controlling sinking velocity. If we consider the fractal scaling relationship for excess density (Eq. (5) and (8) in our manuscript), this clarity becomes blurred, because the aggregate excess density is itself size-dependent. However, if we further consider that natural aggregate size ranges over more than an order of magnitude (from sizes of about $0.45 \cdot 10^{-6}$ m, which is operationally defined by typical filter pore sizes for POM filtration, to size of $O(10^{-2}$ m)), while aggregate excess density $(\rho_f - \rho)$ typically ranges only between zero (neutrally buoyant) and $O(100\,\mathrm{kg\,m^{-3}})$, it is obvious that size is the dominant factor (for non-neutrally buoyant aggregates), while, as we clearly state, excess density can entail high variability of sinking velocity. This is also, what e.g. Iversen & Robert (2015)[1] imply, when writing '2- to 3-fold higher **size-specific** sinking velocities' for mineral ballasted aggregates.

$\rightarrow$ We add the reference Iversen & Robert 2015 to the follow-up sentence: "... entail high variability of excess density and thus sinking speed of aggregates (Iversen & Robert, 2015)"

- Page 3, line 15: Please replace "while ignoring" with "while neglecting" (the effects are still discussed).

$\Rightarrow$ Thanks.

$\rightarrow$ Changed.
* * *
[1]Iversen & Robert 2015: *Ballasting effects of smectite on aggregate formation and export from a natural plankton community.* Marine Chemistry 175, 18 - 27.

- Page 3, lines 29-31: As mentioned above, I would point out that Henson et al. (2012) suggested an opposing pattern. Would it be possible to reproduce also this opposing pattern with M$^4$AGO? I think a brief discussion of this issue would be interesting – potentially regarding the presented sensitivity experiment with smaller diatom frustules showing much lower transfer efficiencies in high latitudes?

  ⇒ As stated above, we now mention the opposing latitudinal pattern explicitly. From our present knowledge about the model responses, the pattern proposed by Henson et al. 2012 could be likely reproduced by applying unreasonable parameter values. However, an investigation of this question would require multiple model simulations, which is a computationally costly task and out of the scope of our manuscript. We therefore only provide reasons to relate our model results to Weber et al. 2016 (an order of magnitude more phosphate than direct flux observations, which makes the transfer efficiency calculations of Weber et al. more reliable), and don't discuss the pattern proposed by Henson et al. intensively, which may be a future work.

    → Changed as described above. We now mention the opposing pattern.

- Page 4, line 28: Please consider including the equation for opal dissolution explicitly, also to better understand the given dissolution rates in Table 1. As far as I understand / looking at the HAMOCC code, the opal dissolution rate given in Table 1 corresponds to 7 °C?

  ⇒ The reviewer is right, and we agree that this information is useful. We hence provide the equation. However, we believe, it is better placed on p. 12, l18, where the new $Q_{10}$-dependent remineralization has been introduced. While doing so, we realized that we haven't introduced the symbols $r_{\mathrm{opal}}$, $T$, $T_{\mathrm{ref}}$ and $r_{\mathrm{POC}}$ which we additionally added.

    → Added after Eq. (30):
      "where $r_{\mathrm{opal}}$ is the opal dissolution rate at the reference water temperature $T_{\mathrm{ref,opal}}$ and $T$ is the ambient water temperature."

"In the standard version, we remain with the former linearly temperature-dependent opal dissolution (Ragueneau et al., 2000, Segschneider and Bendtsen, 2013)" $\rightarrow$ "In the standard version, we remain with the former linearly temperature-dependent opal dissolution $(\partial_t[\text{opal}] = -r_{\text{opal}}(0.1(T+3))[\text{opal}])$ (Ragueneau et al., 2000, Segschneider and Bendtsen, 2013)"

"...where $K_{O_2}$ is the half saturation constant in Michaelis-Menten kinetics" $\rightarrow$ "...where $K_{O_2}$ is the half saturation constant in Michaelis-Menten kinetics, and $r_{\text{POC}}$ is the remineralization rate at reference temperature $T_{\text{ref,POC}}$

- Page 5, line 9-11: Please rephrase; e.g., ... is eventually computed from a number distribution that is truncated at the minimum and maximum aggregate diameters ..., and expressions for the mass and sinking velocity of aggregates of a particular diameter:

  $\Rightarrow$ Thanks. Rephrased accordingly and accounted for the reviewers #2 comment on integration differential.

    $\rightarrow$ "...determined by a truncated number distribution, Eq. (2), through the minimum and maximum aggregates sizes, $d_{\text{min}}$ and $d_{\text{max}}$, respectively, the aggregate mass, $m(d)$, and the sinking velocity of single aggregates, $w_s(d)$" $\rightarrow$ "...computed from the number distribution, Eq. (2), that is truncated at the minimum and maximum aggregate sizes, $d_{\text{min}}$ and $d_{\text{max}}$, respectively, and expressions for the aggregate mass, $m(d)$, and the sinking velocity of aggregates, $w_s(d)$, of a particular diameter, $d$. Integration over the aggregate size spectrum yields $\langle w_s \rangle$,"

- Page 5, line 26: It would be helpful to define the diameter d of an aggregate more accurately here. For example, is the diameter of an aggregate with df =1 (i.e., a chain) just given by its length?

  $\Rightarrow$ Yes. Adding a sub-clause.

    $\rightarrow$ A $d_f = 1$ would depict a chain of aggregate constituents, where the length equals the aggregate diameter, ..."

- Page 6, line 7: Please move reference to "well known" Stokes (1851) here.

  ⇒ Ok. Done.

  → Moved the reference.

- Page 6, line 26: I am a little lost here. What is the motivation for this paragraph? What is n? And why is that equation only true for $n \neq n_p$?

  ⇒ We here derive the mean primary particle size, based on the encapsulated solid volumes of individual, poly-sized primary particles inside the fractal aggregate, to conserve the total solid volume and thus the porosity. This comes at the cost that the theoretical number of primary particles of mean primary particle diameter $\langle d_{\mathrm{p}} \rangle$ is not necessarily the same as the number of individual primary particles (which would only be the case for mono-sized primary particles). This, however, is negligible for the calculations that follow. We now give some more explanation that clarifies the issue.

    → "while $n \cdot \langle d_p \rangle^3 = \sum_i n_i d_{p,i}^3$ with $n \neq \sum_i n_i$, and thus the porosity of the aggregate is unimpaired." → "and thus the porosity of the aggregate is unimpaired, while the calculation does not presume equal number of mean, $n$, and individual primary particles, $\sum_i n_i$, (hence, $n \cdot \langle d_p \rangle^3 = \sum_i n_i d_{p,i}^3$ with $n \neq \sum_i n_i$ for poly-sized primary particles), which is negligible in the following as we don't consider $n$ any further."

- Page 6, line 27: Do I understand correctly that "same heterogeneity in a size spectrum" means that all aggregates of a particular composition / in a particular grid cell are assumed to have the same heterogeneity/microstructure/dp (for all aggregate diameters d)?

  ⇒ Yes.

    → Seems, as there is no change needed. Remained.

- Page 7, line 9: Please define Vp,i more accurately; volume of the primary particles per unit volume of sea water?

$\Rightarrow$ $V_{p,i} = \frac{1}{6}\,\pi\,d_{p,i}^3$ is the individual primary particle volume. We clarified it.

$\rightarrow$ "$V_{p,i}$" $\rightarrow$ "$V_{p,i} = \frac{1}{6}\,\pi\,d_{p,i}^3$"

- Page 7, line 6-8: Maybe shorten to "... tracer Ci, namely detritus, opal, calcite and dust."

  $\Rightarrow$ Thanks, but we remained with the extra sentence, since we believe it is an important information that shouldn't be given in a sub-clause.

  $\rightarrow$ Remained.

- Page 7, line 15: Add "Multiplication by the volume of the mean primary particle then yields..."; helps the reader / I didn't see this at first.

  $\Rightarrow$ Changed.

  $\rightarrow$ "the mass of a mean primary particle can be written as" $\rightarrow$ "multiplication by the volume of the mean primary particle then yields the mass of a mean primary particle"

- Page 8, line 3: Please add reference to traditional scaling relationship.

  $\Rightarrow$ Added the previously given references.

  $\rightarrow$ added: (Logan and Wilkinson, 1990; Kranenburg, 1994)

- Page 9, line 1: For consistency, if j=0..3 here, also add definitions of a/bj=0 in last paragraph of page 8.

  $\Rightarrow$ We added a sub-clause at the end of the whole paragraph, p.9, l.7, since the application of $a_{j=0}$ and $b_{j=0}$ is only useful in the context of how the lower integration boundary of the mean sinking velocity is defined. Further, we introduce $a_{j=0} = b_{j=0} = 1$, since dividing by zero in case of $a_{j=0} = b_{j=0} = 0$ is not defined.

  $\rightarrow$ is the maximum diameter of aggregate, and by applying $a_{j=0} = b_{j=0} = 1$, the lower integration boundary equals the mean primary particle diameter

- Page 9, line 13: It is not clear to me what "dynamic steady state" means.

  ⇒ We extended the sentence to specify that the dynamic steady state is between aggregation and fragmentation

    → "Instead of modeling the processes of aggregation and fragmentation explicitly or prescribing $b$, we assume dynamic steady state for the slope of the number distribution" → "Instead of modeling the processes of aggregation and fragmentation explicitly or prescribing $b$, we assume dynamic steady state between aggregation and fragmentation to describe the slope of the number distribution."

- Page 9, line 20-21: I am wondering why the assumption of Reynolds numbers between 0.1 and 10 here is okay, while the authors go through the extra trouble of deriving expressions for the sinking speed for even smaller and even larger Reynolds numbers in Section 2.2.2. Is there a reason for this?

  ⇒ Thanks for pointing out the lacking information. Since we are aware of and discuss the limitations of the dynamic steady state size distribution, we here avoided extra complexity where little benefit as compared to an explicit representation of the dynamic size distribution is expected (see Sect. 3.10: Current limitations of M⁴AGO, where we discuss the limitations of our current approach versus an explicit representation of a dynamic size distribution). We therefore simplify at this point and remain with text.
  With regards to the drag formulation: in a previous model version of M⁴AGO, we applied the simple Stokes sinking velocity ($c_D = 24/Re_p$) and disregarded the restriction to $Re_p < 0.1$, with similar well results for the transfer efficiency, but clearly underestimated the maximum diameter of aggregates and thus the represented size range of aggregates. For future applications of M⁴AGO, this can be of relevance. In order to point out this advantage, we add a sentence in the previous paragraph (p.8,l.24) and remove sub-clause on p.9, l.4

$\rightarrow$ "We approximate this representation by" $\rightarrow$ "This drag representation leads to smaller settling velocities for large aggregates than the classical Stokes drag $(c_D = 24/Re_p)$. Hence, aggregates can grow larger, until they reach the globally fixed critical $Re_p$ for fragmentation, $Re_{\mathrm{crit}}$, which leads to a more realistic representation of the size range of aggregates. We approximate the White drag representation by"
Removed: "where $Re_{\mathrm{crit}}$ is the globally fixed critical $Re_p$ for fragmentation."

- Page 10, line 6: Please clarify / see above comment on "in a size spectrum": ... as one value across all aggregate sizes.?

  $\Rightarrow$ Modified.

    $\rightarrow$ "as one value across a particle size spectra" $\rightarrow$ as one value across all aggregate sizes

- Page 11, line 10: Shouldn't it read: When detritus from the frustules is remineralized, it is replaced...?

  $\Rightarrow$ No, since there can be more detritus available than needed to fill the void.

    $\rightarrow$ Remained.

- Page 12, line 2: Please rephrase: ...thus decrease the fractal dimension of aggregates, and ii) ...

  $\Rightarrow$ Done.

    $\rightarrow$ "thus fractal dimension of aggregates is small" $\rightarrow$ "thus decrease the fractal dimension of aggregates"

- Page 15, Table 1: Is it correct that the applied opal dissolution rate in the setup with M$^4$AGO is larger than that in the standard model setup? What is the resulting effect of the temperature-dependency here? Wouldn't the larger remineralization rate combined with the slower opal sinking speeds in the euphotic zone lead to very high opal production, and consequently to very low calcite production?

⇒ The reviewer is right that the dissolution rate is larger in M$^4$AGO than in the standard model setup. However, the RLSs for opal (the ratio between sinking velocity and dissolution rate) eventually determine the silicate retention in the water column. As briefly discussed in Sect. 3.6: Regional fluxes & rain ratios, the opal RLSs are indeed shorter in surface waters, but longer in regions of the mesopelagic zone and below. In total, the attenuation of opal fluxes with depth remain similar in most regions (see Fig. 11). Hence, HAMOCC with M$^4$AGO doesn't show much difference in the global opal to CaCO$_3$ production ratio, which can also be seen in the flux ratios, shown in Fig. 5.

→ Remained with the present state of description.

- Page 16, line 7: I am not sure if I understand correctly: Is dp,calc chosen particularly small to avoid an overestimate of the volumetric density effect? If so, for clarity, while accounting for could be rephrased: We set dp,calc ..., to account for ....

⇒ Modified accordingly.

→ ", and hence, we set $d_{p,\mathrm{calc}} = 3\,\mathrm{\mu m}$, which is thus at the lower bound of the observed range while accounting for the volumetric density effect of non-spherical plate-like coccoliths." → "We set $d_{p,\mathrm{calc}} = 3\,\mathrm{\mu m}$, which is thus at the lower bound of the observed range, to account for the volumetric density effect of non-spherical plate-like coccoliths."

- Page 16, line 24: Sentence unclear to me; ...distinction between parameter tuning and model evaluation, when... (?)

⇒ We aimed at clarification and rephrased the sentence.

→ "The close connection between the parametrized processes of sinking and remineralization, the transfer efficiency and the climatological nutrient field hampers the clear distinction between tuning and evaluation data when comparing the model results to literature values for transfer efficiency" → "The newly parametrized

processes of sinking and remineralization directly affect the transfer efficiency, and thus the climatological nutrient fields. This close connectedness hampers the clear distinction between data employed for model tuning or for model evaluation, when comparing the model results to literature values for transfer efficiency."

- Page 16, line 33: Please rephrase: Since the adaptation of the sinking velocity and thus of the transfer efficiency to the remineralization and dissolution rates occurs within a few years, parameter variations aiming at a quantitative agreement with the transfer efficiency of Weber et al. were feasible. (?)

  ⇒ For clarity, we split the sentence.

    → "Since the adaptation of the sinking velocity versus the remineralization and dissolution rates, and thus the transfer efficiency, was within a few years, parameter variations aiming at a quantitative agreement with the transfer efficiency of Weber et al. (2016) enabled a useful strategy to select for promising parameter sets." → "We performed parameter variations aiming at a quantitative agreement with the transfer efficiency of Weber et al. (2016). Since the adjustment of the sinking velocity versus the remineralization and dissolution rates, and thus the transfer efficiency, occurs within a few years, this strategy was useful to select for promising parameter sets."

- Page 17, lines 8-13: Move to results section / next paragraph?

  ⇒ Since this part includes methodological aspects, we follow your suggestion to start a new paragraph in the methods section.

    → New paragraph started.

- Page 17, lines 12-13: The annual mean of only one year seems rather short. Have you checked how sensitive your results are with respect to interannual variability due to, e.g., ENSO or deepwater formation variability?

⇒ We are using a climatological forcing, which, by definition, does not resolve interannual events such as ENSO. However, internal variability of ocean circulation still happens and we here therefore provide the climatological 100 a mean of the transfer efficiency and its respective standard deviation (Fig. 2). As can be seen, the standard deviation is small for most ocean regions and exhibits higher values in the Antarctic polar regions. These are likely linked to the shifting of polar fronts through the internal variability of ocean circulation that imprints on the ocean biogeochemistry. However, the overall latitudinal mean pattern of the transfer efficiency resembles the one of Fig. 8 in the manuscript. In our manuscript, we are not concerned with the internal variability and focused on the general potential effects of aggregate composition and microstructure on POC fluxes as explanatory factors for the global pattern of transfer efficiency. We may revisit the aspect of internal variability in a follow up work and don't want to lengthen the manuscript further at this stage. We therefore remain with the present status.

→ Remained.

- Page 18, lines 1-3: Please rephrase / correct sentence structure ("features" can not refer to "In the M$^4$AGO run").

⇒ Thanks. Rephrased accordingly.

→ "In the M$^4$AGO run, the equatorial Pacific exhibits the lowest export efficiencies, features maximum values of about 0.14-0.16 in the subtropical gyres and about 0.20 in the Arctic region (Fig. 4 d)" → "In the M$^4$AGO run, the equatorial Pacific exhibits the lowest export efficiencies, the subtropical gyres feature maximum values of about 0.14-0.16, and the Arctic region about 0.20 (Fig. 4 d)"

- Page 18, lines 3-4: It is difficult to say from Fig. 4 wether the use of M$^4$AGO really leads to smaller latitudinal variability, since the minima and the global mean p-ratio also seem lower than in the standard run. Maybe remove this statement or double-check?

[Figure]

[Figure]

Figure 2: 100 a climatology of the transfer efficiency (left) and the standard deviation (right). Shifting of polar fronts due to internal variability of ocean circulation and the affected biogeochmistry are likely the cause for higher standard deviation of the transfer efficiency in the Antarctic polar region than in the rest of the ocean. The overall mean transfer efficiency pattern resembles the one shown in Fig. 8 in our manuscript.

⇒ We double-checked. The standard has even smaller minimum p-ratios than the M⁴AGO run. We therefore remain with the text.

→ Remained.

- Page 20, line 8: This is only shown in Fig. 7a, not in Fig. 7d.

⇒ Thanks.

→ Modified: "7 a,d" → "7 a"

- Page 21, Figure 7 (and page 26, Fig. 9): It would be helpful to show (or at least describe) the location of WOA transect P16.

⇒ We now describe the location in the caption. Caption changed:

→ "Modeled marine aggregate properties on WOA transect P16" → "Modeled marine aggregate properties on the Pacific WOA transect P16, which is located at about 150 °W"

- Page 23, line 7: Sentence unclear to me: ... we neglect this effect versus? Maybe delete versus the lower primary particle binding forces? What are those forces?

  ⇒ We aimed at clarification and rephrased the sentence.

    → "While compaction likely enhances the internal number of binding links in aggregates, we neglect this effect versus the lower primary particle binding forces on the overall susceptibility to shear stress and thus kept $Re_{\mathrm{crit}}$ globally constant." → "Compaction can coincide with an increasing number of binding links in aggregates, which can lower the overall susceptibility of aggregates to shear stress. In $M^4AGO$, we disregard this effect and keep $Re_{\mathrm{crit}}$ globally constant.

- Page 23, line 8: "we neglect... and kept" For clarity, I would suggest to consistently stick to present-tense for the work performed for this study, and to past tense for previous results.

  ⇒ We rephrased the sentence and now stick to present tense. See above.

    → As reformulated in the previous comment.

- Page 25, line 26: Shouldn't it read "...decay to $1/e$ of its initial value, ..."?

  ⇒ The reviewer is right. Thank you very much! We modified it accordingly

    → "half" → "$1/e$ $(\approx 37\,\%)$"

- Page 27, line 9 / Figure 9: The lower remineralization rates described here are hard to see in Figure 9b, also due to the missing label on the -30% (?) contour; maybe smaller contour intervals would help.

  ⇒ For clarity, we add the $30\,\%$ contour label, see Fig. 3.

    → Contour label added.

[Figure]

Figure 3: Added "-30 %" contour level label in b)

- Page 27, lines 29-30: Maybe clearer: The relative contributions provide information about the main driving factors for local sinking speed deviations from the global mean.

  ⇒ We modified the sentence.

  → "The relative contributions provide an information on the main driving factors, expressed as percentage, for the local $\langle w_s \rangle$ as compared to global average aggregates." → "The relative contributions provide information about the main driving factors for the local $\langle w_s \rangle$ as compared to $\langle w_s \rangle$ of global average aggregates."

- Page 28, lines 23-24: Please rephrase / clarify sentence; e.g.: M[4]AGO

thus likely underestimates the spatial variability and relative contribution of b to ws.

$\Rightarrow$ Thanks. Done.

$\rightarrow$ "likely underestimates the spatial variability of the relative contribution to $\langle w_s \rangle$" $\rightarrow$ "likely underestimates the spatial variability and relative contribution of $b$ to $\langle w_s \rangle$"

- Page 30, lines 22-23: Unclear sentence structure. Maybe: Similar to POC fluxes, opal fluxes exhibit shorter RLSs in ..., while they exceed the standard RLSs in ...

$\Rightarrow$ Modified accordingly.

$\rightarrow$ "Similar to the POC remineralization length scales, opal fluxes exhibit shorter opal RLS in the surface waters, while they exceed the standard RLS in the mesopelagic zone and below (not shown)." $\rightarrow$ "Similar to POC fluxes, opal fluxes exhibit shorter opal RLSs in the surface waters, while they exceed the standard RLSs in the mesopelagic zone and below (not shown)."

- Page 30, line 27: "... fluxes are generally small." Is this true for both model versions? Not shown here, or is it?

$\Rightarrow$ It's a result from low opal production in the subtropical gyres, visible in Fig. 5 a,b. We therefore give a reference to Fig. 5a,b

$\rightarrow$ added: "(see Fig. 5a,b)

- Page 30, lines 28-29: Please add sedimentation flux in standard model for comparison.

$\Rightarrow$ We now provide the global Si flux to sediment for the standard run as well.

$\rightarrow$ "$\sim 1.03\,\mathrm{Gt}$ Si per year" $\rightarrow$ "$\sim 1.03\,\mathrm{Gt}$ Si per year ($\sim 1.04\,\mathrm{Gt}$ Si per year in the standard run)"

[Figure]

Figure 4: As Fig.12 in the manuscript, but for POC/PIC ratio. Comparison of the standard run and the M$^4$AGO run to the Mouw et al (2016a,b) data set. Standard refers to upper two rows, M$^4$AGO to the lower two rows.

- Page 31, line 7: "The M$^4$AGO run represents the PIC/POC fluxes equally well as the standard run." It would be interesting to know how well that is.

  ⇒ We here provide the PIC/POC fluxes in the reply, see Fig. 4

  → Nothing to be changed.

- Page 31, lines 8 and 11: "... the scatter around the 1:1 line is reduced ..." (line 8) At least for some regions, e.g. for the Sub Antarctic Zone, the points are not really scattered around the 1:1 line. But I agree with the view that M$^4$AGO reduces the variability in the POC/Si ratio (line 11). Isnt't this reduced variability / compression in the POC/Si fluxes expected, because the variability of the fluxes is only due to the

variability of the POC/Si concentrations in M$^4$AGO (POC and opal sink at the same speed), while in the standard model, variability is also introduced due to differences between Si- and POC-sinking speeds?

⇒ We write at the end of the first paragraph of Sect. 3.6 that M$^4$AGO couples the timing of mineral and POC fluxes and took that also as motivation for the comparison to data, as lined out on p. 31, l.1 ff. We therefore agree that the compression likely stems from the joint sinking of mineral and POC components, which we also discuss on p. 31, l. 11 ff. We therefore remain with the present text.

→ Remained.

- Page 32, Figure 12: Do I understand correctly that each dot in the figure is a generated monthly mean data point, compared to the respective location and monthly mean of the last year in the model run?

  ⇒ We modified the caption for clarity.

  → "POC/Si rain ratios in the standard run and M$^4$AGO compared to the Mouw et al. (2016a,b) data set." → "Monthly POC/Si rain ratios in the standard run and in M$^4$AGO compared to the monthly climatological mean derived from the Mouw et al. (2016a,b) data set."

- Page 33, line 34: ... and the North American Westcoast (?)

  ⇒ Yes. Thanks for improving clarity.

  → "North America coast" → "North American West Coast"

- Page 35, lines 20-22 / page 36, Figure 14a: If negative fluxes really do represent a net-CO2 uptake by the ocean in Fig. 14a, as stated in the caption, the southern hemisphere ocean acts as a net sink for atmospheric CO2 (and not a source), and the northern hemisphere ocean acts as a net source (not a sink). Consequently, the oceanic CO2 transport would be from south to north.

$\Rightarrow$ We are generally very sorry for having caused confusion about the units and thus the interpretation of Fig. 14a. It should be $\mathrm{Gt\,C\,yr^{-1}}$. For easier interpretation, we add the starting point of the cumulative flux calculation (from south to north). Given that the cumulative fluxes show a positive sign at the equator, the southern hemisphere is a net-source of $CO_2$. We remain with the general statement and provide additional information on the cumulation procedure in the caption of Fig. 14.

$\rightarrow$ "Climatological cumulative zonal $CO_2$ flux in the standard and the $M^4AGO$ run" $\rightarrow$ "Climatological cumulative zonal $CO_2$ flux in the standard and the $M^4AGO$ run (from south to north)"

- Irrespective of the sign / flux direction, these results are in stark contrast to CO2 flux observations of net zonal mean outgassing at the Equator and net ocean CO2 uptake in mid-latitudes (e.g., Figure 14 in Takahashi et al. 2019). Maybe this is just due to a plotting error in Figure 14?

$\Rightarrow$ The plot is in agreement with this general pattern. The cumulative sum shows a positive trend towards the equatorial region (and beyond until about $15\,°\mathrm{N}$), which indicates outgassing in the tropics. We, however, have to admit, that the text gave a different impression, which we change. Many thanks for pointing this out!

$\rightarrow$ p.35, l. 24,26: deleted "sub" of "subtropical" $\rightarrow$ "tropical"

- I also am not sure if I understand the units in Figure 14a. Does the left axis show the net sea–air CO2 flux accumulated over the respective $1°$ latitude band? If that is true, the values seem very large. I am guessing from Fig. 14a that the ocean CO2 uptake accumulated in the southern hemisphere would then amount to around 0.2 GtC/yr/deg·60deg≈12 GtC/yr, which is an order of magnitude larger than the observed net uptake by the southern hemisphere ocean (south of 14°S) of about 1.1 GtC/yr.

$\Rightarrow$ We are again sorry for the wrong unit, which we correct for. Generally, the uptake in the southern hemisphere is lower than 1.1 GtC/yr, since Fig. 14a corresponds to pre-industrial conditions, as stated at the beginning of Sec. 3.8. We add a sentence to provide references for the qualitative and quantitative agreement.

$\rightarrow$ Added the sentence: "In general, the latitudinal zonal $CO_2$ fluxes and the cross-equatorial southward oceanic $CO_2$ transport agree qualitatively and quantitatively well with former forward-integrated models for pre-industrial conditions (e. g Sarmiento et al., 2000; Gloor et al., 2003; Mikaloff Fletcher et al., 2007)."

- Page 35, lines 22-24: "In the simulation with M4AGO, a stronger CO2 uptake in the region ... coincides with ... increased transfer efficiency" This is a very interesting point; does it still hold despite the (to my understanding) erroneous Figure 14a? To me it is surprising that the CO2 fluxes do \*not\* differ more, despite the very different transfer efficiencies. Why do the CO2 fluxes hardly differ south of, say, 55°S, where the transfer efficiency difference is largest? Why is there hardly an effect in the Arctic Ocean?

  $\Rightarrow$ We are again sorry for the misleading, wrong axis label. At first glance, the strong differences in transfer efficiency contradicts the little changes in atmosphere-ocean $CO_2$ fluxes in some regions. However, transfer efficiency is not equal to actual POC fluxes. Transfer efficiency only describes, which fraction of exported POC is transferred to certain depth (in our manuscript calculated for about $1000\,\mathrm{m}$). For example, in the Arctic ocean, we find a high transfer efficiency of the exported material, but actual POC fluxes are small. Thus the high transfer efficiency in the Arctic has hardly any impact on the atmosphere-ocean $CO_2$ fluxes. We now provide a brief explanation for it. With regards to the region south of $55\,°S$, we hope that with clarification of the units in Fig. 14 a it becomes clear that indeed, a clear effect of the increased transfer efficiency in M$^4$AGO compared to the standard run is visible. There is a stronger uptake in the AAZ region in M$^4$AGO (see the difference between M$^4$AGO and the standard run), as also visible in the rising $CO_2$ experiments (cmp. Fig. 14 c). This is also described in the text (see p. 37, l.7ff: "Quantitatively, differences of regional cumulative $CO_2$ fluxes larger than $5\,\mathrm{Gt\,C}$ appear in the Antarctic Zone (AAZ),") We rephrase the sentence and link the transfer efficiency and the primary production pattern.

    $\rightarrow$ "Qualitatively, this coincides well with the higher transfer efficiencies in these regions (cmp. to Fig. 8)." $\rightarrow$ "Qualitatively,

this coincides well with the primary production, respective export and the higher transfer efficiencies in these regions (cmp. to Fig. 4 and 8). In regions of higher transfer efficiency, but similar $CO_2$ fluxes as compared to the standard run, either POC export fluxes are small (e. g. in the Arctic Ocean) or physical processes such as mixing or upwelling dominate over biologically induced $CO_2$ fluxes (e. g. in the SAZ)."

- Page 39, line 7: Sentence structure. ... body size decreases with increasing water temperature. And increasing water temperature has been suggested to ... (?)

  ⇒ Modified.

    → "...which is suggested..." → ". The increasing water temperature has been suggested..."

- Page 39, line 10: Does "... such eco-physiological responses ..." refer to the primary particle size change, or to other effects?

  ⇒ Yes, such eco-physiological responses refer to the change of primary particle size.

    → Remained with the sentence.

- Page 39, line 29: ... increases the phosphate concentrations in the subtropical gyres by up to 50% (?)

  ⇒ No, it's more than $150\%$. The sentence is fine.

    → Remained.

- Page 40, line 1-2: Please rephrase (phosphate increases phosphate concentrations...), e.g. by: ...phosphate ... populates ... and reaches the subtropical gyres.

  ⇒ Modified.

$\rightarrow$ "Phosphate previously utilized by diazotrophs in the Panama basin now partially populates the downstream equatorial current and increases the subtropical gyres phosphate concentrations." $\rightarrow$ "Phosphate previously utilized by diazotrophs in the Panama basin now partially populates the downstream equatorial current and reaches the subtropical gyres."

- Page 41, lines 21-29: This paragraph, describing the main implications of this study, is not formulated very clearly. "Our findings ... suggest a number of implications." Number=2, according to later "first" and "second"? Please rephrase second sentence. E.g., First, the finding that the size ... is a potential contributor to high sinking speeds suggests that the ballast hypothesis needs to be extended to a size-and-ballast hypothesis. What does "it requires" in line 23 refer to?

  $\Rightarrow$ We rephrased the sentences accordingly.

    $\rightarrow$ "First, size of aggregate constituents, particularly of diatom frustules, as potential factor for high sinking velocity suggest to widen the perspective of mineral ballast studies towards a size-and-ballast hypothesis." $\rightarrow$ "First, the finding that the size of aggregate constituents, particularly of diatom frustules, act as potential factor for high sinking velocities, suggests to widen the perspective of mineral ballast studies towards a size-and-ballast hypothesis."
    "it requires" $\rightarrow$ "such extended size-and-ballast hypothesis requires"

- Page 43, line 9: As far as I understand $\beta$ is not prescribed in the standard run, but only the sinking speed, i.e., $\beta$ still depends on the remineralization (which varies with temperature and oxygen concentrations). How do you get to the value of $\beta=1$?

  $\Rightarrow$ In the referred standard run, remineralization is not temperature-dependent, but oxygen-dependent. Thus, $\beta$ is implicitly prescribed by the gradient of sinking velocity $(\partial_z \bar{w}_s)$ and the remineralization rate $R_{POC}$ in oxygenated waters. Following Kriest and Oschlies (2008), the vertical mass concentration exponent is defined

by $R_{\mathrm{POC}}/\partial_z \bar{w}_s + 1 = 2$, (their r/a+1 in Eq. (5)), which is prescribed in HAMOCC. Hence, it translates to a preset $\beta = 1$, which, by internal processes such as reduced $R_{\mathrm{POC}}$, upwelling, but also numerical diffusion, then results in an effective Martin slope $\langle \beta' \rangle$ that is smaller than one. For clarity, we now provide information about the assumption on the oxygenation state.

$\rightarrow$ "that is smaller than the prescribed value of $\beta = 1.00$" $\rightarrow$ "that is smaller than the prescribed value of $\beta = 1.00$ in oxygen-saturated waters"

**2.3 Typos**

- Page 2, line 32: Primary / fundamental(?) determining factors?

  $\Rightarrow$ Thanks. Changed.

  $\rightarrow$ "primer" $\rightarrow$ "primary"

- Page 3, line 29: ...benefits from an order of...

  $\Rightarrow$ Thanks. Added.

  $\rightarrow$ added: "an"

- Page 4, line 23: Bar over ws meaning global mean / annual mean?

  $\Rightarrow$ It's also the mass concentration-weighted mean sinking velocity, but different from the spatio-temporally variable one in M$^4$AGO. So, we added the information.

  $\rightarrow$ "Below $z_0$, we assume a linearly increasing mean sinking velocity with depth." $\rightarrow$ "Below $z_0$, we assume a linearly increasing mass concentration-weighted mean sinking velocity with depth."

- Page 4, line 27: The opal dissolution rate...

  $\Rightarrow$ Modified.

  $\rightarrow$ "Opal" $\rightarrow$ "The opal"

- Page 10, line 27: ... enhance the sinking velocity...

  ⇒ Modified.

    → added: "the"

- Page 18, line 7: ... which also lead (plural)

  ⇒ Thanks. Changed.

    → "lead"

- Page 18, line 8: ... from either satellite data, in situ observations, or models lead to partly contrasting patterns (add "data" and plural "s")

  ⇒ Changed.

    → added "data" and plural "s" in pattern"s"

- Page 19, line 5: Both model simulations show a similar pattern (add "a")

  ⇒ Changed.

    → "Both model simulations show similar pattern of opal to detritus ratio fluxes" → "Both model simulations show a similar pattern of opal to detritus flux ratios"

- Page 22, line 11: Use "By contrast" rather than "In turn"?

  ⇒ Changed.

    → "In turn" → "By contrast"

- Page 22, line 13: ... during the aggregates's descent. (?)

  ⇒ Modified.

    → "aggregates" → "their"

- Page 23, line 3: It is likely that (no comma)

$\Rightarrow$ Changed.

$\rightarrow$ removed ","

- Page 23, line 17: linearly increasing

  $\Rightarrow$ Changed.

  $\rightarrow$ "linear" $\rightarrow$ "linearly"

- Page 23, line 30: ...from the relationship... and the transfer efficiency.

  $\Rightarrow$ Changed.

  $\rightarrow$ "from relationship between the Martin curve slope parameter and transfer efficiency" $\rightarrow$ "from the relationship between the Martin curve slope parameter and the transfer efficiency"

- Page 24, line 5: M$^4$AGO posseses...

  $\Rightarrow$ Changed.

  $\rightarrow$ "possess" $\rightarrow$ "possesses"

- Page 24, line 11: ...allows to more reliably constrain POC transfer efficiency

  $\Rightarrow$ Allows for is correct.

  $\rightarrow$ Remained.

- Page 25, line 5: extent

  $\Rightarrow$ Changed.

  $\rightarrow$ "extend" $\rightarrow$ "extent"

- Page 25, line 30: ... in surface waters and the upper mesopelagic zone...

  $\Rightarrow$ Changed.

$\rightarrow$ added: "the"

- Page 25, lines 32-33: ... the RLSs... are similar or slightly (?) longer, or smaller again in ... The longer RLSs in the mesopelagic zone... (plural)

  $\Rightarrow$ Modified in parts.

  $\rightarrow$ "In the mesopelagic zone, the RLSs in M$^4$AGO is similar or pronounced longer and decreases" $\rightarrow$ "In the mesopelagic zone, the RLSs in M$^4$AGO are similar or pronounced longer and decrease" "The longer RLS" $\rightarrow$ "The longer RLSs"

- Page 27, line 16: In summary, the temperature-dependence...induces...

  $\Rightarrow$ Changed.

  $\rightarrow$ "temperature-dependence of remineralization in M$^4$AGO induce" $\rightarrow$ "the temperature-dependence of remineralization in M$^4$AGO induces"

- Page 27, line 28: ...$X_{i,z}$ is (not as)

  $\Rightarrow$ Changed.

  $\rightarrow$ "as" $\rightarrow$ "is"

- Page 28, line 20: ... given the general importance of the microstructure....

  $\Rightarrow$ Changed.

  $\rightarrow$ "given the microstructure general importance" $\rightarrow$ "given the general importance of the microstructure"

- Page 35, line 29: ... in M$^4$AGO on *the* CO2 uptake ...

  $\Rightarrow$ Changed.

  $\rightarrow$ added: "the"

- Page 39, line 34: one *in the* too much; thus leads to (singular)

⇒ Thanks. Changed.

→ "primary production in the in the equatorial upwelling regions (ETA and ETP) thus lead" → "primary production in the equatorial upwelling regions (ETA and ETP) thus leads"

- Page 40, line 13: grazing through zooplankton

⇒ Changed.

→ "though" → "through"

- Page 43, line 7: underpins the previously...

⇒ Changed.

→ added: "the"

- Page 50, line 33: initials for Núñez-Riboni

⇒ Bibliography updated. Thanks.

→ Done.

- Page 53, line 4: Aiko Voigt (not Vogt?)

⇒ Yes. Thanks.

→ Done.

- Page 53, line 19: please check reference / entry missing?

⇒ Done.

→ added: "PANGAEA"

- Page 55, line 11: please check / C. R. Geoscience

⇒ Yes. It's the official abbreviation for 'Comptes Rendus Geoscience'

→ modified to: "Comptes Rendus Geoscience"

---

## Author Comment (AC2) · 27 Jan 2020

**1 General remark**

The reply is structured as follows:

- Referee comment

  ⇒ Authors reply

    → Modification(s) in the manuscript. "old" ↛ "new"

**2 Reply to Referee #2**

**2.1 General summary**

- Review of Maerz et al: *Microstructure and composition of marine aggregates as co-determinants for vertical particulate organic carbon transfer in the global ocean.* The authors present a new scheme for the calculation of the mean sinking velocity of marine aggregates as a function of the aggregate composition and the fractal dimension. This scheme is reported to be cost-efficient and hence useful in large-scale ocean models. The model is described in detail and carefully evaluated. The authors report a substantial improvement in the simulation of the latitudinal pattern of POC transfer efficiency. This is an impressive effort and worth of publication. I have some specific comments that should be addressed before publication. Writing style: Sentences are very long and not always clear. This is particularly true for the introduction and model description.

  ⇒ We thank the reviewer for her/his constructive and positive review. We tried to respond to every single comment and hope to improved the clarity of the manuscript.

    → –

**2.2 General comments**

- I miss a comparison with the stochastic, Lagrangian model of sinking biogenic aggregates in the ocean (SLAMS) by Jokulsdottir and Archer. Jokulsdottir, T. and Archer, D.: A stochastic, Lagrangian model of

sinking biogenic aggregates in the ocean (SLAMS 1.0): model formulation, validation and sensitivity, Geosci. Model Dev., 9, 1455–1476, https://doi.org/10.5194/gmd-9-1455-2016, 2016.

⇒ Indeed, the model of Jokulsdottir and Archer (2016) is an interesting new model approach. Unfortunately, their Lagrangian model is currently limited to a conceptual 1-D application and is computationally likely too costly for the incorporation in an Earth System Model framework. However, it might provide valuable insights for further development and tuning of M$^4$AGO or similar models. We pick up on it and briefly discuss it in Sect. 3.10 Current limitations of M$^4$AGO.

→ Added on p. 40, l. 28 (after ... "subsequent aggregation (Martin et al., 2011)."): "More detailed models, such as e.g. the 1-D Lagrangian approach of Jokulsdottir and Archer (2016), can likely provide more insights into aggregate dynamics and can help to further improve the aggregate representation in ESM frameworks."

• Please make the model code publicly available. It is not in the repository that you mention.

⇒ The model code was, is and will be publicly available on request. However, we acknowledge that the provided information how and where to request it, was insufficient. We therefore now provide more information i) in the manuscript and ii) in the MPGPuRe repository, how to inquire it. It requires to agree to the MPI-ESM license agreement and registering at the MPI-ESM-Forum. Unfortunately, the license obliges users to this method for accessing the code due to third party rights on the code.

→ "Primary data and code for this study is stored and made available through the Max-Planck-Gesellschaft Publications Repository: https://pure.mpg.de." → "Primary data and code for this study is stored and made available through the Max-Planck-Gesellschaft Publications Repository: https://pure.mpg.de. The respective MPI-OM and HAMOCC model code (revision numbers: r4981 and r5003) is available on request after agreeing

to the MPI-ESM license agreement and registering at the MPI-ESM-Forum
(https://www.mpimet.mpg.de/en/science/models/licenses/)."

- explain ALL abbreviations and symbols used in the figures in each and every figure caption.

  ⇒ We carefully went through the manuscript and added the abbreviations and symbols, where we believe, it's necessary. If not otherwise stated in the modifications below, no modifications were performed for the caption.

    → Added in caption of Fig. 1:
      "$V_{p,i}$ is the primary particle volume, $\rho_{p,i}$ is the primary particle density, and $d_{p,i}$ is the primary particle diameter of primary particle type $i$. $m(d)$ is the mass of an aggregate of diameter $d$. $\langle d_p \rangle$ and $\langle \rho_p \rangle$ represent mean primary particle diameter and density, respectively."

      Added in caption of Fig. 2:
      "$l$ denotes the thickness of the opal shell with volume $V_{\mathrm{opal}}$, $V_{\mathrm{aq}}$ and $V_{POM}$ are the encapsulated volumes of water and POM, respectively. $d_{p,\mathrm{frustule}}$ is the diameter of the diatom frustule."

      Added in caption of Fig. 3: "$d_{\max}$"

      Refrained from adding the symbols explicitly in caption of Fig. 7, since they are given nearby in caption of Fig. 6 and we refer to them explicitly.

      Modifications caption of Fig. 9:
      "$z^*_{\mathrm{POC}}$" → "the remineralization length scales, $z^*_{\mathrm{POC}}$,"
      "sinking velocity" → "sinking velocity, $w^*_{s,\mathrm{POC}}$,"
      "temperature-dependent remineralization rates" → "$Q_{10}$ factor temperature-dependent remineralization rates with $r_{\mathrm{POC}}$ at reference temperature $T_{\mathrm{ref}}$"
      "Relative contributions of sinking and remineralization" → "Relative contributions of sinking, $RC_{\langle w_s \rangle}$, and remineralization, $RC_{\mathrm{remin}}$,"

Added in caption of Fig. 10:
"Mathematical symbols are: $\langle \rho_p \rangle$: mean primary particle density; $\langle d_p \rangle$: mean primary particle diameter; $d_f$: fractal dimension; $\mu$: dynamic molecular viscosity; $d_{\max}$: maximum aggregate diameter; $b$: aggregate number distribution slope."

Modified caption of Fig. 13:
"tracers $\rightarrow$ "tracers (oxygen, $O_2$, nitrate, $NO_3^-$, phosphate, $PO_4^{3-}$, silicate, Si)"

Modifications in caption of Fig. 15:
"POC transfer efficiency to $d_{p,\text{frustule}}$" $\rightarrow$ "POC transfer efficiency to diatom frustule size, $d_{p,\text{frustule}}$"
"and $d_f$" $\rightarrow$ "and fractal dimension, $d_f$"

- I am quite worried about the high buoyancy of diatom-dominated aggregates through the TEP formulation. This needs more justification. Do you here assume that all organic carbon has the same density as TEP? That would explain your low density of diatom-dominated aggregates. Is there sufficient evidence for such behavior?

  $\Rightarrow$ We acknowledge that our formulation is simplified. We will make the simplification more clear. However, we don't assume that all detritus has the same density as TEPs. We assume that TEPs lower the density of the diatom frustule and parametrize the effect with respect to the model-defined freshness of detritus. In our model framework, we thus focus on the qualitative effect of TEP that has been previously suggested (Mari et al., 2017) and, as we learned, also applied by Jokulsdottir et al. 2016. There are a number of observational and experimental studies that support the general behaviour of TEPs as buoyancy adding agent in marine aggregates. For example, the following experimental and observational studies point to low TEP density or show low buoyancy of aggregates, similar to our modeled diatom-dominated aggregates with mean excess densities of $\Delta \langle \rho_f \rangle_V \approx 2 \, \text{kg} \, \text{m}^{-3}$

  – Azetsu-Scott and Passow, 2004[2]: $\rho_{\text{TEP}} \approx 700 - 840 \, \text{kg} \, \text{m}^{-3}$.
* * *
[2] Azetsu-Scott, K. and Passow, U.: Ascending marine particles: Significance of trans-

> Thus TEPs are lighter than sea water and add buoyancy, if incorporated in aggregates.

- Alldredge & Gottschalk 1988[3]: excess density of median marine aggregates: $0.14\,\mathrm{kg\,m^{-3}}$, and for diatom aggregates (their Fig. 2 d): $\approx 10^{-5} - 10^{-4}\,\mathrm{g\,cm^{-3}} = 10^{-2} - 10^{-1}\,\mathrm{kg\,m^{-3}}$ for large, porous diatom-dominated aggregates ($O(1\,\mathrm{cm})$).

- Laurenceau-Cornec et al. 2019[4]: excess density of diatom-dominated aggregates including TEPs (partly also including mineral components): $\approx 0.3 - 5.7\,\mathrm{kg\,m^{-3}}$

Some of the studies don't explicitly mention TEP (since they were not explicitly described at the time of the study), but nowadays TEPs are ubiquitously found in the global ocean and thus possess a likely explanation for the low aggregate excess densities. We believe that the confusion about our modeled diatom-dominated aggregates arises primarily due to the fact that most often diatom-dominated aggregates are reported in the form of marine snow, and hence large aggregates that have high sinking velocities of often more than $> 50\,\mathrm{m\,d^{-1}}$. Our model agrees well with sinking velocities of these individual aggregates, when considering only the respective sizes (and not the mean sinking velocity of the full size spectrum). For example, here using Stokes sinking velocity for simplicity, thus slightly overestimating $w_s$ compared to the White drag parametrization:

$$w_s(d = 1\,\mathrm{mm} = 0.001\,\mathrm{m}) = \frac{1}{18\,\mu}\Delta\rho_f\,g\,d^2$$

$$= \frac{1}{18 \cdot 0.0015\,\frac{\mathrm{kg}}{\mathrm{m\,s}}} \cdot 2\,\frac{\mathrm{kg}}{\mathrm{m^3}} \cdot 9.81\,\frac{\mathrm{kg}}{\mathrm{m}} \cdot (0.001\,\mathrm{m})^2 \cdot 86400\,\frac{\mathrm{s}}{\mathrm{d}}$$

$$\approx 62.8\,\mathrm{m\,d^{-1}}$$

parent exopolymer particles (TEP) in the upper ocean, Limnol. Oceanogr., 49, 741–748, 2004

[3]Alldredge, A. L. and Gottschalk, C.: In situ settling behaviour of marine snow, Limnol. Oceanogr., 33, 339-351, 1988

[4]Laurenceau-Cornec, E.C, Moigne, F.A.C., Gallinari, M, Moriceau, B., Toullec, J., Iversen, M.I., Engel, A., De La Rocha, C.L.: New guidelines for the application of Stokes' models to the sinking velocity of marine aggregates, Limnol. Oceanogr., 2019, doi:10.1002/lno.11388

The due to Stokes slightly overestimated value of $\approx 62.8\,\mathrm{m\,d^{-1}}$ is of the same order as e. g. the measurements by Alldredge & Gottschalk (1988), who found the settling velocities can be best described by $50 \cdot (d[\mathrm{in~mm}])^{0.26}$ (their Fig. 3 a) which would be $\approx 50\,\mathrm{m\,d^{-1}}$ for our example diameter of $1\,\mathrm{mm}$.

As a consequence of the reviewers comment, we now give the reference to Jokulsdottir and Archer (2016) and Mari et al. (2017) closer to the description of our diatom density description. Both author teams explicitly described the potential of TEPs to add buoyancy through the TEPs lower density than water ($\rho_{\mathrm{TEP}} \approx 700 - 840\,\mathrm{kg\,m^{-3}}$, Azetsu-Scott and Passow, 2004). In addition, we emphasize our simplification of TEP representation. We add the reference to Laurenceau-Cornec at the point, where we discuss the diatom-dominated aggregate excess density. To further clarify the fact that mean sinking velocity can differ significantly from reported large marine snow aggregates of sizes of typically $> 1\,\mathrm{mm}$, we add a note in the caption of Fig. 6. Further, we add a sentence on p. 22 l.2, where we present the sinking velocities.

$\rightarrow$ "The density of diatoms becomes" $\rightarrow$ "To account for the additional buoyancy through TEPs (Jokulsdottir and Archer, 2016; Mari et al., 2017), we here simplify and assume that the frustule density is lowered by TEPs in dependency on the freshness of detritus. Eventually, the diatom density, $\rho_{\mathrm{diatom}}$, becomes"

Added in caption of Fig. 6 (after concentration-weighted mean sinking velocity of aggregates):
"Note that $\langle w_s \rangle$ comprises the full range of many micrometer-sized to rare, large aggregates with low ($O(1\,\mathrm{m\,d^{-1}})$) and high ($O(> 100\,\mathrm{m\,d^{-1}})$) sinking velocities, respectively."

Added reference to: "Laurenceau-Cornec et al., 2019" on p. 22,l.20/21

"Particle properties and molecular dynamic viscosity determine the concentration-weighted mean sinking velocity of aggregates, $\langle w_s \rangle$ (Fig. 6 f,l), for particle sizes ranging from few micrometers

to millimeters." $\rightarrow$ "Particle properties and molecular dynamic viscosity determine the concentration-weighted mean sinking velocity of aggregates, $\langle w_s \rangle$ (Fig. 6 f,l). For $\langle w_s \rangle$, M$^4$AGO considers particle sizes ranging from few micrometers to millimeters and thus the full size spectrum, where sinking velocities of $O(1\,\mathrm{m\,d^{-1}})$ to $O(> 100\,\mathrm{m\,d^{-1}})$ are represented. $\langle w_s \rangle$ thus can significantly differ from reported sinking velocities for large individual aggregates."

**2.3   Abstract**

- Line 14: too much information given: delete rising CO2 and without CO2 climate feedback.

  $\Rightarrow$ We rephrased the sentence and hope, that it is now better readable.

    $\rightarrow$ "In ocean standalone runs and rising carbon dioxide (CO$_2$) without CO$_2$ climate feedback, M$^4$AGO alters the regional ocean-atmosphere CO$_2$ fluxes compared to the standard model." $\rightarrow$ "Prescribing rising carbon dioxide (CO$_2$) concentrations in standalone runs (without climate feedback), M$^4$AGO alters the regional ocean atmosphere CO$_2$ fluxes compared to the standard model."

**2.4   Introduction**

- P.2 Please give more references for your statements, especially in the first paragraph. No reference given between line 5 and 11.

  $\Rightarrow$ Given the general principles stated in the paragraph, we now provide a reference to the excellent book of Williams and Follows 2011: Ocean Dynamics and the Carbon Cycle: Principles and Mechanisms.

    $\rightarrow$ added reference: "(Williams and Follows, 2011)"

- P. 2, line 17: "The sinking velocity of aggregates is primarily determined by their size". This needs a reference. I would argue it is density, e.g. Iversen and Robert,http://dx.doi.org/10.1016/j.marchem.2015.04.009 . The next sentence also needs a reference (line 19)

$\Rightarrow$ A similar comment has been done by reviewer #1. We answered his/her comment and provide the answer here:

This is an interesting comment and we realize, also by the same comment of reviewer #2, that there seems to be much confusion about the controlling factors for sinking velocity, which deserves a publication on its own (being in progress). We want to emphasize here that we clearly state in the follow-up sentence that structure and composition of aggregates regulate the excess density and can thus have a high impact on sinking velocity (we now provide a reference for it). Nevertheless, we would here argue from the mathematical perspective. For simplicity and neglecting the changing drag coefficient for particles with higher Reynolds particle numbers, let's consider the Stokes sinking velocity for low particle Reynolds numbers:

$$w_s(d, \rho_f, \ldots) = \frac{1}{18\,\mu}\,(\rho_f - \rho)\,g\,d^2 \qquad (2)$$

where $d$ is the diameter, $\mu$ the molecular dynamic viscosity, $\rho_f$ the aggregate density, $\rho$ is the density of the ambient fluid, and $g$ is the gravitational acceleration constant. It is obvious that $w_s \propto (\rho_f - \rho)$ and $w_s \propto d^2$. Hence, sinking velocity only linearly increases with aggregate density, while it increases with a power law relationship of the diameter. This suggests that size is indeed the primary factor controlling sinking velocity. If we consider the fractal scaling relationship for excess density (Eq. (5) and (8) in our manuscript), this clarity becomes blurred, because the aggregate excess density is itself size-dependent. However, if we further consider that natural aggregate size ranges over more than an order of magnitude (from sizes of about $0.45 \cdot 10^{-6}$ m, which is operationally defined by typical filter pore sizes for POM filtration, to size of $O(10^{-2}$ m)), while aggregate excess density $(\rho_f - \rho)$ typically ranges only between zero (neutrally buoyant) and $O(100\,\text{kg m}^{-3})$, it is obvious that size is the dominant factor (for non-neutrally buoyant aggregates), while, as we clearly state, excess density can entail high variability of sinking velocity. This is also, what e.g. Iversen & Robert (2015)[5] imply, when writing '2- to 3-fold higher **size-specific** sinking velocities'
* * *
[5]Iversen & Robert 2015: *Ballasting effects of smectite on aggregate formation and export from a natural plankton community.* Marine Chemistry 175, 18 - 27.

for mineral ballasted aggregates.

$\rightarrow$ We add the reference Iversen & Robert 2015 to the follow-up sentence: "... entail high variability of excess density and thus sinking speed of aggregates (Iversen & Robert, 2015)"

- P. 2, line 32: primer $\rightarrow$ primary?

$\Rightarrow$ Changed.

$\rightarrow$ "primer" $\rightarrow$ "primary"

**2.5  Model description**

- It would be very helpful to have a table with all symbols used in the equations at the beginning of section 2.1

$\Rightarrow$ We will provide the table in the appendix.

$\rightarrow$ Adding table. At end of first paragraph of Sect. 2: Model description, p. 4, l.15, we add: "A table with the used mathematical symbols can be found in App. D, Tab. D1"

- P. 5, line 2, what is meant with "terminal sinking velocity"? I suggest to delete "terminal"

$\Rightarrow$ Terminal sinking velocity is the sinking velocity of any particle in steady state, when all involved forces balance. This is the classical assumption, when applying e. g. the Stokes formula for sinking velocity (or any other drag formulation in the provided Eq. (7)). By writing 'terminal sinking velocity', we make it clear that we are not attempting to solve the Maxey-Riley equation (M.R. Maxey & J.J. Riley 1983: Equation of motion for a small rigid sphere in a nonuniform flow. The Physics of Fluids 26, 883, https://doi.org/10.1063/1.864230). We thus remain with the present formulation.

$\rightarrow$ Remained with the formulation.

- Eq. 3: I can guess what is meant with dd, but it is easily misunderstandable.

$\Rightarrow$ In agreement with reviewer #1 and aiming at clarity, we rephrased this part and now make clear that $\langle w_s \rangle$ is derived from integration over the size spectrum.

$\rightarrow$ "The local concentration-weighted mean sinking velocity, $\langle w_s \rangle$, in M$^4$AGO is eventually determined by a truncated number distribution, Eq. (2), through the minimum and maximum aggregates sizes, $d_{\min}$ and $d_{\max}$, respectively, the aggregate mass, $m(d)$, and the sinking velocity of single aggregates, $w_s(d)$" $\rightarrow$ The local concentration-weighted mean sinking velocity, $\langle w_s \rangle$, in M$^4$AGO is eventually computed from the number distribution, Eq. (2), that is truncated at the minimum and maximum aggregate sizes, $d_{\min}$ and $d_{\max}$, respectively, and expressions for the aggregate mass, $m(d)$, and the sinking velocity of aggregates, $w_s(d)$, of a particular diameter, $d$. Integration over the aggregate size spectrum yields $\langle w_s \rangle$,"

- P. 5, line 28: What is meant by a "primary particle". How does that differ from "a particle"?

$\Rightarrow$ Primary particles are the modelled (smallest) entities of which an aggregate is composed of. As we state in p.5, line 28, we consider e.g. phytoplankton cells or coccolithophore shells as primary particles. We render the definition of primary particles more precisely on p. 6, l.17-19, when we introduce the theory for heterogeneously composed aggregates. There, we state:
"With M$^4$AGO, we represent aggregates composed of poly-dense, poly-sized primary particles under the assumption of a singled value fractal dimension throughout the aggregate size spectrum. This allows for representing heterogeneous primary particles such as diatom frustules, coccoliths, dust particles, and detritus as principal components of marine aggregates. ".
'Particle' is a more general term and can refer, depending on context, to a primary particle or an aggregate. As a consequence of the reviewer's question, we carefully checked throughout the manuscript, if we always refer correctly to primary particles.

$\rightarrow$ We remained with the text.

- P. 7, line 1: What is meant by "the total number of one primary particle type" ? The total number of one should be one. Do you mean "of particles of one particle type"?

  ⇒ Yes. We modified the sentence accordingly.

    → "between the total number of one primary particle type" → "between the total number of primary particles of one particle type"

- P. 8, line 5: "mean primary particle size, (. . .) which we apply as a lower integration bound". Please give a justification for this choice.

  ⇒ The fractal scaling law breaks for aggregates smaller than the smallest composing entity, i. e. the primary particle size. For $\lim d \to \langle d_p \rangle$, the aggregate is composed of one single primary particle. Hence, the lower limit must be $\langle d_p \rangle$. Anyhow, we now provide an additional reference to Kriest and Evans (1999).

    → Added reference: "and hence, $d_{\min} = \langle d_p \rangle$" → "and hence, $d_{\min} = \langle d_p \rangle$ (following Kriest and Evans, 1999)"

- P. 10, line 5: no reference to Engel et al 2004? Engel, A. , Thoms, S., Riebesell,U. , Rochelle-Newall, E. and Zondervan, I. (2004) Polysaccharide aggregation as a potential sink of marine dissolved organic carbon. Nature, 428 . pp. 929-932. DOI 10.1038/nature02453.

  ⇒ Thanks for the reference. Added it.

    → Added: "Engel et al., 2004"

- Figure 2: explain abbreviations and symbols in each and every figure caption.

  ⇒ Where needed, we now do it. See the remark in the general comments section of the same reviewer.

    → Particularly for Fig. 2, we added to the caption:
      "$l$ denotes the thickness of the opal shell with volume $V_{\text{opal}}$, $V_{\text{aq}}$ and $V_{POM}$ are the encapsulated volumes of water and POM, respectively. $d_{p,\text{frustule}}$ is the diameter of the diatom frustule."

- P. 11, line 18: how is $m_e$ and $m_{\text{potential}}$ calculated? I can't follow whether the masses of opal and of TEP are taken into account correctly to calculate the density of the diatom-aggregate. Do you here assume that all organic carbon has the same density as TEP? That would explain your low density of diatom-dominated aggregates. Please clarify.

  $\Rightarrow$ We do not assume that diatom frustule-associated detritus has the same density as TEP. We first calculate the frustule density according to Eq. (27), which is based on the density of opal, $\rho_{\text{opal}}$, and POM $\rho_{\text{POM}}$ (now corrected to $\rho_{\text{det}}$). We admit that we should have written $\rho_{\text{det}}$ instead of $\rho_{\text{POM}}$, which we change. For the eventual density of the diatom $\rho_{\text{diatom}}$, we assume that the amount of diatom frustule-associated detritus is proportionally linked to the presence of TEP, which makes the diatom frustule lighter. We apologize for not having provided the equations for $m_e$ and $m_{\text{potential}}$, which we now provide. Our modeled primary particle densities in silicifier-dominated waters like the Southern Ocean are about $1060 \, \text{kg m}^{-3}$ to $1100 \, \text{kg m}^{-3}$ (values based on re-checked monthly mean model output) are in a normal range of detritus. We are certain that the calculated excess density of rather fresh, porous diatom-dominated aggregates in surface waters is with about $2 \, \text{kg m}^{-3}$ in the range of previously measured excess densities of diatom aggregates. For example, a range of about 0 to $\sim 10 \, \text{kg m}^{-3}$ is given by Alldredge and Gottschalk, 1988, Ploug et al. 2008 and Iversen and Ploug 2013 (as cited in the manuscript). Please see also the reply to the general comments of the same reviewer. Further, the modeled excess density of diatom-dominated aggregates increases with ongoing remineralization during the decent through the mesopelagic zone (Fig. 7 d).
  We clearly state in Sect. 3.10 (Current limitations of M$^4$AGO) that TEPs are only simplistically considered. We make it more clear now in Sect. 2.2.5, p. 11, l.16. Given the potential role of TEPs in aggregate formation, we currently consider to include TEPs explicitly.

  $\rightarrow$ "$\rho_{\text{POM}}$" $\rightarrow$ "$\rho_{\text{det}}$"

    "defined as the mass ratio between the actual amount of detritus $m_e$ and the potential mass of detritus linked to opal production $m_{\text{potential}}$." $\rightarrow$ "defined as the mass ratio between the actual amount of detritus, $m_e = n_{\text{frustule}} V_{\text{POM}} \rho_{\text{det}}$ and the potential mass of detritus linked to diatom frustules $m_{\text{potential}} = n_{\text{frustule}} (V_{\text{POM}} + V_{\text{aq}}) \rho_{\text{det}}$."

"The density of diatoms becomes" $\rightarrow$ "To account for the additional buoyancy through TEPs (Jokulsdottir and Archer, 2016; Mari et al., 2017), we , we here simplify and assume that the frustule density is lowered by TEPs in dependency on the freshness of detritus. Eventually, the diatom density, $\rho_{\text{diatom}}$, becomes"

- P. 13, line 1: mention that this forcing is based on ERA reanalysis (be specific) and avoid the abbreviation OMIP which you don't explain (or explain it)

  $\Rightarrow$ We now provide more information on the OMIP forcing and avoid the word OMIP.

  $\rightarrow$ "We run both, the standard and the M$^4$AGO run, in a GR15/L40-OMIP setup. This translates to a horizontal resolution of about 1.5°, 40 uneven vertical layers with highest resolution in the first few hundred meters of the ocean. OMIP is a climatological daily atmospheric forcing (Röske, 2005)." $\rightarrow$ "We run both, the standard and the M$^4$AGO run, in a GR15/L40 setup with climatological forcing. This translates to a horizontal resolution of about 1.5°, 40 uneven vertical layers with highest resolution in the first few hundred meters of the ocean. The climatological atmospheric boundary conditions are derived from the second European Centre for Medium-Range Weather Forecasts (ECMWF) Re-Analysis project (ERA-40; Simmons and Gibson, 2000; Röske, 2005). The mean annual cycle of i.e. wind stress, heat and freshwater fluxes are resolved on a daily basis. The continental freshwater runoff is provided by means of a runoff model (Röske, 2005)."

- P. 13, line 3/4: are these global numbers? What are corresponding model parameters?

⇒ Yes, these are global number. We rephrased the sentence for clarity . As stated in the text, the values are the weathering rates for the substances: $CaCO_3$, dissolved organic phosphate, and silciate.

→ "The loss of POM, opal, and $CaCO_3$ due to sedimentation and subsequent burial was accounted for through homogeneously applied weathering rates which were adjusted accordingly, namely for the standard / $M^4AGO$ run: $CaCO_3 \approx 17.2 / 26.5\,T\,mol\,C\,yr^{-1}$, dissolved organic phosphorus $\approx 99.6 / 101.5\,G\,mol\,P\,yr^{-1}$, and silicate $\approx 3.2 / 2.3\,T\,mol\,Si\,yr^{-1}$" → "The loss of POM, opal, and $CaCO_3$ due to sedimentation and subsequent burial was accounted for through homogeneously applied weathering rates which were adjusted for the standard run (and the $M^4AGO$ run): Globally, we add $\approx 99.6\,(101.5)\,G\,mol\,P\,yr^{-1}$ as dissolved organic phosphorus, and $\approx 3.2\,(2.3)\,T\,mol\,Si\,yr^{-1}$. To compensate for the loss of $CaCO_3$, we add $\approx 17.2\,(26.5)\,T\,mol\,C\,yr^{-1}$ to surface dissolved inorganic carbon (DIC) and a corresponding amount to surface total alkalinity, $A_T$, as on DIC:$2A_T$."

- Table 1: caption: "The value for the Martin curve. . ." . Why is this single parameter given in the caption, please add it it to the list of parameters in the main body of the table.

  ⇒ Since the table is about the new $M^4AGO$ parameters, we completely remove the sentence. The value is still provided and discussed in the appendix.

  → Removed.

- P. 16, line 4: "a minor role in biogenic fluxes". This statement needs a reference.

  ⇒ We provide now references.

  → added: "(described by e. g. Berelson et al., 2007; Fischer and Karakaş, 2009, Fischer et al., 2016)"[6].

[6]Berelson, W. M., Balch, W. M., Najjar, R., Feely, R. A., Sabine, C., and Lee, K.: Relating estimates of $CaCO_3$ production, export, and dissolution in the water col-

- P. 16, line 33/34: "adaptation .. within a few years." Is adaptation the right word here? Maybe "an equilibrium was established"? or its change after a few years was small..

    ⇒ We rephrased the whole sentence and split in in two sentences for clarity. We now use the word 'adjustment' instead of 'adaptation'.

    → "Since the adaptation of the sinking velocity versus the remineralization and dissolution rates, and thus the transfer efficiency, was within a few years, parameter variations aiming at a quantitative agreement with the transfer efficiency of Weber et al. (2016) enabled a useful strategy to select for promising parameter sets." → "We performed parameter variations aiming at a quantitative agreement with the transfer efficiency of Weber et al. (2016). Since the adjustment of the sinking velocity versus the remineralization and dissolution rates, and thus the transfer efficiency, occurs within a few years, this strategy was useful to select for promising parameter sets."

**2.6   Results**

- P. 22, line 12-13: "diatom-dominated aggregates feature a high buoyancy through TEP." Is there any evidence for such behavior or is this a major model bug?

    ⇒ Yes, there is evidence for such behavior and we discuss the low excess densities of diatom-dominated aggregates in l. 19-21 on the same page. TEPs indeed feature high buoyancy (their density has

umn to measurements of $CaCO_3$ rain into sediment traps and dissolution on the sea floor: A revised global carbonate budget, Global Biogeochemical Cycles, 21, GB1024, https://doi.org/10.1029/2006GB0028

Fischer, G. and Karakaş G.: Sinking rates and ballast composition of particles in the Atlantic Ocean: implications for the organic carbon fluxes to the deep ocean , Biogeosciences, 6, 85–102, 2009

Fischer, G., Romero, O., Merkel, U., Donner, B., Iversen, M., Nowald, N., Ratmeyer, V., Ruhland, G., Klann, M., and Wefer, G.: Deep ocean mass fluxes in the coastal upwelling off Mauritania from 1988 to 2012: variability on seasonal to decadal timescales, Biogeosciences, 13, 3071 – 3090, 2016

been estimated to be about 700-840 $\mathrm{kg\,m^{-3}}$, Azetsu-Scott and Passow, 2004 as cited in the mansucript). The excess density of modelled, fresh aggregates in diatom-dominated regions is with about $2\,\mathrm{kg\,m^{-3}}$ in the observed range of about 0 to $\sim 10\,\mathrm{kg\,m^{-3}}$ is given by Alldredge and Gottschalk, 1988, Ploug et al. 2008 and Iversen and Ploug 2013 (as cited in the manuscript). We want to emphasize here again that the sinking velocity provided in Fig. 6 and 7 are mean sinking velocity. Observed sinking velocity of individual, large aggregates cannot be compared directly to the mean sinking velocity. Modeled large aggregates feature substantially higher sinking velocities than reflected by the mean sinking velocity. Large aggregates in our model thus have similar sinking velocity as compared to observed marine snow. See also the reply in the general part.

$\rightarrow$ Remained.

- P. 23, line 25: I assume z0 is 100m, please clarify.

  $\Rightarrow$ Yes, it is. It was introduced $z_0$ on p. 4, l. 21. We added $z_0$.

    $\rightarrow$ "The transfer efficiency of POC from $100\,\mathrm{m}$ to" $\rightarrow$ "The transfer efficiency of POC from $z_0 = 100\,\mathrm{m}$ to"

- Line 26: "to about 1000m" $\rightarrow$ at 1000 m.

  $\Rightarrow$ "at" would be wrong for two reasons: first, it is the transfer efficiency from $100\,\mathrm{m}$ to (about) $1000\,\mathrm{m}$, and second, we don't interpolate the fluxes to exactly $1000\,\mathrm{m}$, but use the fluxes across the model layer boundary at $960\,\mathrm{m}$ depth to calculate the transfer efficiency.

    $\rightarrow$ Remained.

- P. 28, line 29: this is not shown in Fig 7a, you only show mean density, not the effect of opal on density.

  $\Rightarrow$ We disagree. Even at deep ocean regions, where detritus is almost remineralized (and thus POM and TEP play only a minor role for

primary particle density), primary particle density of diatom frustules in silicifier-dominated regions (e. g. south of -40°) is significantly lower than primary particle density in calcifier-dominated regions (e. g. at about -20°). Hence, opal acts less than $CaCO_3$ as ballasting agent in our model.

$\rightarrow$ Remained.

- Line 30: any indication in the literature and any scientific explanation why silicate frustule size affect the sinking speed if not by density?

$\Rightarrow$ This is indeed a good question. Unfortunately, observational studies have so far heavily focused on aggregate size-to-sinking velocity relationships. Minor focus has been put on the solid hydrated density. Hence, even less emphasis has been put on the relation of primary particle sizes, microstructure and sinking velocity. However, as we discuss below line 30, studies by e. g. Laurenceau-Cornec et al. 2015 point to a relevance of cell size and morphology, and dominant size of primary producers have been suggested to drive interannual changes in export fluxes (Boyd and Newton, 1995, see the manuscript). From theoretical considerations, the size of primary particles can affect the porosity, Eq. (6), and hence the aggregate excess density (irrespective of the density of the single primary particles). This is described in the manuscript by Eq. (5). We again searched for relevant literature on experimental studies that clearly disentangle these aspects for marine aggregates. A very recent study by Laurenceau-Cornec et al. (2019)[7] provides a summary, how porosity can vary and can have a decisive role for determining the settling velocity. A new study by C. Flintrop is in preparation that will provide further insights into settling dynamics of marine aggregates. We now explicitly refer to the relevant equations to address, why primary particle size can affect the packaging and thus sinking velocity. We add an explanatory sentence and provide the reference to Laurenceau-Cornec et al. (2019).
* * *
[7]Laurenceau-Cornec, E. C., Le Moigne, F. A. C., Gallinari, M., Moriceau, B., Toullec, J., Iversen, M. H., Engel, A., and De La Rocha, C. L.: New guidelines for the application of Stokes' models to the sinking velocity of marine aggregates, Limnol. Oceanogr., https://doi.org/10.1002/lno.11388, 2019

→ Adding on p.28, l. 32 (after: ". . . likely play a role."): "As indicated by Eq. (5) and (6), primary particle size affects the excess density and porosity of aggregates, which have decisive effects on sinking velocity (Laurenceau-Cornec et al., 2019)."

- Fig 10: colorbar label: conribution → contribution(add 't')

⇒ Thanks. Changed.

→ added the "t", see Fig. 1

[Figure]

Figure 1: "conribution" → "contribution"

- P. 35 and Figure 14a: what is the reason of showing cumulative $CO_2$ fluxes integrated over latitude? Please show just the zonal means, that's much easier to understand and compare to data. The units should not include per degree if it is cumulative.

⇒ We apologize for the wrong units. However, cumulative fluxes give directly the net-transport across the equator and also provide the information that both model runs are in well spun-up states. We therefore remained with the cumulative fluxes and only corrected the units.

→ Units changed.

- Figure 14c-k: cumulative fluxes make more sense here. I'd prefer actual fluxes/time and then the difference between the two could be cumulative. Then, one y-axis might also be enough.

  ⇒ We agree with the reviewer that cumulative fluxes make sense here since the subfigures provide a direct insight in how much $CO_2$ is taken up in the course of the years. We thus remain with the figure as it is.

  → Remained.

- P. 37, line 24: suggested → hypothesize (careful which tense you use). Also, please please back up this hypothesis with literature.

  ⇒ We here refer to the suggestion made earlier in the manuscript (in Sect. 3.5). We therefore keep 'suggested', and now explicitly refer to the section. In Sect. 3.5, we discuss and cite a number of experimental studies that link sinking velocity of marine aggregates to morphology and size structure of the phytoplankton community. Together with the theoretical derivation of the aggregate excess density (Eq. (5)) and porosity (Eq. (6)), being dependent on primary particle size, these and our study provide reasonable indications that primary particle size (or -diatom- cell size, respectively) potentially affects the sinking velocity of marine aggregates. This led us to suggest primary particle size as an additional factor for aggregate sinking. We are, however, aware that full experimental evidence is, to our best knowledge, lacking and will hopefully be part of future research, where the role of microstructure and composition will be deciphered and disentangled.

  → "We suggested" → "Previously in Sect. 3.5, we suggested"

- P. 38: you have not shown silicate distribution – is that reasonable? You refer to low transfer-efficiency in silicifier-dominate region, but this is not the case in the Southern Ocean, nor do you see much of an impact in Figs 15 a and d in the Southern Ocean, which is THE region dominated by silicifiers. This needs more explanation.

⇒ We show in Fig. 5a,c the opal-to-POC flux ratio. As described in the methods part, Sect. 2 (first paragraph), HAMOCCs opal production is directly coupled to silicate availability. The regions of silicifiers are therefore well defined and previously described in the manuscript (and silicate is included in our general Taylor diagram-based analysis of nutrient fields). With respect to the low transfer efficiency, Fig. 15 g clearly shows a drop in transfer efficiency for the AAZ and SAZ regions, once the frustule size of diatoms, $d_{p,\text{frustule}}$ is decreased (cmp. M$^4$AGO run with $S(d_{p,\text{frustule}})$, where $\langle T_{\text{eff}} \rangle$ decreases from about 0.24 to about 0.14 for the Antarctic Zone (AAZ), and from about 0.19 to about 0.11 for the Subantarctic Zone (SAZ)). To make the difference more clear, we rephrase the sentence, where the sensitivity study is compared to the original M$^4$AGO run. We further add the specification, where the largest signal in silicate increase can be found.

→ "As a consequence, the transfer efficiency in silicifier-dominated regions is low (Fig. 15 g). Accordingly, opal dissolves closer to surface waters and the silicate concentration increases with respect to the M$^4$AGO run in silicifier-dominated regions (Fig. 15 d)."
⟶ "As a consequence, the transfer efficiency in silicifier-dominated regions is lower in $S(d_{p,\text{frustule}})$ than in M$^4$AGO (Fig. 15 g). Accordingly, opal dissolves closer to surface waters and the silicate concentration increases compared to the M$^4$AGO run in silicifier-dominated regions, particularly in and downstream of coastal upwelling regions (Fig. 15 d)."